# The microcephaly-associated transcriptional regulator AUTS2 cooperates with Polycomb complex PRC2 to produce upper-layer neurons in mice

Kazumi Shimaoka[1], Kei Hori[1], Satoshi Miyashita[1], Yukiko U Inoue [1], Nao K N Tabe[1,2], Asami Sakamoto[1], Ikuko Hasegawa[1], Kayo Nishitani[1], Kunihiko Yamashiro[1,8], Saki F Egusa[1], Shoji Tatsumoto[3], Yasuhiro Go [3,4,5], Manabu Abe [6], Kenji Sakimura[6], Takayoshi Inoue [1], Takuya Imamura [7] & Mikio Hoshino [1,2 ✉]

## Abstract

**AUTS2 syndrome is characterized by intellectual disability and microcephaly, and is often associated with autism spectrum disorder, but the underlying mechanisms, particularly concerning microcephaly, remain incompletely understood. Here, we analyze mice mutated for the transcriptional regulator AUTS2, which recapitulate microcephaly. Their brains exhibit reduced division of intermediate progenitor cells (IPCs), leading to fewer neurons and decreased thickness in the upper-layer cortex. Increased expression of the AUTS2 transcriptional target *Robo1* in the mutant animals suppresses IPC division, and transcriptomic and chromatin profiling shows that AUTS2 primarily represses transcription of genes like *Robo1* in IPCs. Regions around the transcriptional start sites of AUTS2 target genes are enriched for the repressive histone modification H3K27me3, which is reduced in *Auts2* mutants. Furthermore, we find that AUTS2 interacts with Polycomb complex PRC2, with which it cooperates to promote IPC division. These findings shed light on the microcephaly phenotype observed in the AUTS2 syndrome.**

**Keywords** AUTS2; Microcephaly; Neural Progenitors; Histone Modification; Polycomb
**Subject Categories** Chromatin, Transcription & Genomics; Molecular Biology of Disease; Neuroscience

## Introduction

*AUTS2* has been identified as a possible autism spectrum disorder (ASD) risk gene in a pair of monozygotic twins (Sultana et al, 2002). Structural variants and single nucleotide polymorphisms (SNPs) in the *AUTS2* locus have since been reported as associated with various psychiatric disorders (Oksenberg and Ahituv, 2013; Hori et al, 2021). Patients with AUTS2 syndrome carry heterozygous mutations and exhibit a high frequency of intellectual disability (ID) (98%), as well as attention deficit hyperactivity disorder (ADHD) and ASD (Sanchez-Jimeno et al, 2021; Biel et al, 2022; Beunders et al, 2016). In addition, several common pathological features have been reported frequently in AUTS2 syndrome, notably that 65% of the patients present with microcephaly (Sanchez-Jimeno et al, 2021; Beunders et al, 2016; Beunders et al, 2013). Recently, magnetic resonance imaging (MRI) analysis has reported cortical volume loss, cerebellar hypoplasia, and corpus callosum (CC) hypoplasia in patients with AUTS2 syndrome (Fair et al, 2023; Liu et al, 2021). *Auts2*-deficient mice show behavioral abnormalities that mimic some human symptoms, such as impaired memory, learning, and social behavior (Hori et al, 2020; Hori et al, 2015; Li et al, 2022; Gao et al, 2014). Notably, the brains of the mice have also been reported to exhibit cerebellar hypoplasia (Yamashiro et al, 2020), which may recapitulate the cerebellar hypoplasia seen in patients with AUTS2 syndrome. However, hypoplasia of the cerebral cortex corresponding to microcephaly, seen in many patients, has not been reported in mouse models. In zebrafish, *auts2a* knockdown (KD) by morpholinos resulted in a small brain (Beunders et al, 2013; Oksenberg et al, 2013). Cerebral organoids derived from an AUTS2 syndrome patient exhibited reduced growth (Fair et al, 2023). Although microcephaly may be caused by a decrease in the number of neurons, the mechanism by which microcephaly is caused due to impaired AUTS2 function remains unclear. Therefore, it is

[1]Department of Biochemistry and Cellular Biology, National Institute of Neuroscience, National Center of Neurology and Psychiatry (NCNP), Tokyo 187-8502, Japan. [2]Department of NCNP Brain Physiology and Pathology, Institute of Science Tokyo, Tokyo 113-8510, Japan. [3]Cognitive Genomics Research Group, Exploratory Research Center on Life and Living Systems (ExCELLS), National Institutes of Natural Sciences, Okazaki, Aichi 444-8585, Japan. [4]Department of System Neuroscience, Division of Behavioral Development, National Institute for Physiological Sciences, National Institutes of Natural Sciences, Okazaki, Aichi 444-8585, Japan. [5]Graduate School of Information Science, University of Hyogo, Kobe, Hyogo 650-0047, Japan. [6]Department of Animal Model Development, Brain Research Institute, Niigata University, Niigata 951-8585, Japan. [7]Program of Biomedical Science, Graduate School of Integrated Sciences for Life, Hiroshima University, Hiroshima 739-8526, Japan. [8]Present address: Department of Neurology, University of Texas Southwestern Medical Center, Dallas, TX 75390, USA. ✉E-mail: hoshino@ncnp.go.jp

necessary to recapitulate the phenotype in a mouse model and analyze it to elucidate the molecular mechanism in vivo.

In the mammalian cerebral cortex, neurons are generated directly from progenitor cells called radial glial cells (RGCs) on the ventricular surface during early neurogenesis (direct neurogenesis) (Kriegstein et al, 2006). These neurons differentiate primarily into deep-layer neurons (Molyneaux et al, 2007). As development proceeds, RGCs generate secondary progenitors called IPCs, which divide once to several times in the subventricular zone (SVZ), eventually producing 2–12 neurons (indirect neurogenesis) (Haubensak et al, 2004; Noctor et al, 2004; Miyata et al, 2004; Mihalas and Hevner, 2018). Neurons derived from IPCs differentiate mainly into upper-layer neurons (Vasistha et al, 2015; Mihalas et al, 2016). Mammals have newly acquired IPCs during evolution, which has contributed to the expansion of the mammalian cerebral cortex (Martinez-Cerdeno et al, 2006). While the mechanisms of RGC proliferation and differentiation into neurons have been well investigated (Fernandez et al, 2016; Lui et al, 2011), the regulatory mechanisms of IPC proliferation and differentiation remain largely unexplored.

Previous studies have reported that AUTS2 is localized and functions in cell nuclei and cytoplasm (Gao et al, 2014; Hori et al, 2014). Cytoplasmic AUTS2 regulates the actin cytoskeleton through the RAC1 signaling pathway, which plays a pivotal role in neuronal migration and neurite elongation (Hori et al, 2014). Nuclear AUTS2 limits the number of synapses and regulates the E/I balance in the brain (Hori et al, 2020). It has been reported that nuclear AUTS2 acts as a transcriptional activator or repressor, depending on the cell type (Gao et al, 2014; Liu et al, 2021; Li et al, 2022; Monderer-Rothkoff et al, 2021). Gao et al have shown the transcriptional activation machinery that AUTS2 interacts with non-canonical polycomb repressive complex 1 (PRC1.3/5) and recruits histone acetyltransferase P300 to acetylate at H3K27 (Gao et al, 2014). However, the molecular mechanism of how AUTS2 is involved in transcriptional repression is poorly understood.

In this study, we have successfully generated mouse models of AUTS2 syndrome that recapitulate microcephaly. We have discovered a new mechanism that regulates IPC proliferation and differentiation. We also found that AUTS2 interacts with PRC2 in the nuclei of IPCs, which affects chromatin modifications and represses gene transcription. This study not only leads to a better understanding of the pathogenesis of microcephaly in AUTS2 syndrome and the molecular function of AUTS2, but also sheds light on the evolution of the mammalian brain.

## Results

### Loss of *Auts2* leads to the reduction of upper-layer neurons in the cerebral cortex

Previously, we generated a floxed allele of the *Auts2* gene, which carried two loxP sequences on either side of the exon 8, and named it *Auts2^flox^*. By crossing mice carrying this allele with mice with the *CAG-Cre*-Tg allele that ubiquitously expressed Cre (Sakai and Miyazaki, 1997), we successfully obtained the allele that lacked the exon 8, which we named *Auts2^del8^* (Hori et al, 2014). However, as several research groups have reported different types of floxed alleles for this gene (Gao et al, 2014; Castanza et al, 2021;

Li et al, 2022), we have decided to rename *Auts2^flox^* and *Auts2^del8^* as *Auts2^fl(ex8)^* and *Auts2^del(ex8)^*, respectively (Appendix Fig. S1A). Since homozygotes for *Auts2^del(ex8)^* (*Auts2^del(ex8)/del(ex8)^*) were postnatally lethal, we produced forebrain-specific *Auts2* conditional knockout (*Auts2* cKO) homozygous and heterozygous mice (*Emx1^Cre/+^*; *Auts2^fl(ex8)/fl(ex8)^*, *Emx1^Cre/+^*; *Auts2^fl(ex8)/+^*) by utilizing the *Emx1^Cre^* allele (Hori et al, 2020). We basically use global *Auts2* KO (*Auts2^del(ex8)/del(ex8)^*, *Auts2^del(ex8)/+^*) and forebrain-specific cKO (*Emx1^Cre/+^*; *Auts2^fl(ex8)/fl(ex8)^*, *Emx1^Cre/+^*; *Auts2^fl(ex8)/+^*) for analyses at embryonic and postnatal stages, respectively. We previously confirmed that the full-length AUTS2 (FL-AUTS2) and C-terminal AUTS2 short isoform variant 1 (S-AUTS2-var.1) were successfully eliminated, but C-terminal AUTS2 short isoform variant 2 (S-AUTS2-var.2) expression was increased in the *Auts2* KO homozygous brains (*Auts2^del(ex8)/del(ex8)^* and *Emx1^Cre/+^*; *Auts2^fl(ex8)/fl(ex8)^*, Appendix Fig. S1B) (Hori et al, 2014; Hori et al, 2020).

In the Nissl staining of postnatal day (PD) 30 brain sections, we did not observe apparent morphological differences between the control and *Auts2* cKO homozygous cortices at the macroscopic level (Fig. 1A). However, upon closer observation, the cerebral cortex of cKO homozygotes appeared thinner than that of the controls (Fig. 1B). Next, we performed immunostaining with layer-specific neuronal markers in coronal sections of the PD15 control, heterozygous, and homozygous cortices at three points along the rostrocaudal axis. Notably, the number of upper-layer neurons (CUX1+ cells) was lower in homozygotes and heterozygotes than in controls (Fig. 1C,D). This tendency was more severe in rostral sections than in caudal sections. In contrast, no significant differences were observed in the number of deep-layer neurons (CTIP2+ cells) among genotypes (Fig. 1D). The upper-layer cortex was thinner in heterozygotes and homozygotes than in controls at the rostral point, whereas no significant differences were observed at the central or caudal points (Fig. 1C,E). We also observed that the thickness of the rostral deep-layer cortex was slightly decreased in homozygotes compared with that in controls (Fig. 1E).

We did not find an increased number of cells with cleaved caspase-3, an apoptotic cell death marker, in the cerebral cortex of *Auts2* cKO mice at the embryonic and postnatal stages (Appendix Fig. S1C), suggesting that the reduced size of the cerebral cortex or reduced number of upper-layer neurons in *Auts2* cKO mice was not likely due to cell death. Next, we performed pulse-labeling with 5-ethynyl-2'-deoxyuridine (EdU) on embryos at embryonic days (E) 15.5 and E12.5, when the upper- and deep-layer neurons were preferentially generated, respectively (Molyneaux et al, 2007). At PD10, the number of upper-layer neurons (CUX1+ cells) labeled with EdU at E15.5 was significantly reduced throughout the cortices of *Auts2* cKO homozygotes and heterozygotes compared with controls at PD10 (Fig. 1F). This suggests that upper-layer neurons derived from neural progenitors at E15.5 were reduced at PD10 in *Auts2* cKO homozygotes and heterozygotes. In contrast, the number of CTIP2+ cells labeled with EdU at E12.5 was not significantly different among the genotypes (Fig. 1G). As expected, we found fewer E15.5-labeled CTIP2+ neurons and E12.5-labeled CUX1+ neurons, with no significant differences among the genotypes (Fig. 1F, G). A reduction in E15.5-labeled upper-layer neurons (CUX1+ cells) was observed as early as E18.5 in the global *Auts2* KO (*Auts2^del(ex8)/del(ex8)^* and *Auts2^del(ex8)/+^*) brains (Appendix Fig. S1D). These observations suggest that AUTS2 is required for upper-layer neuron production from neural progenitors.

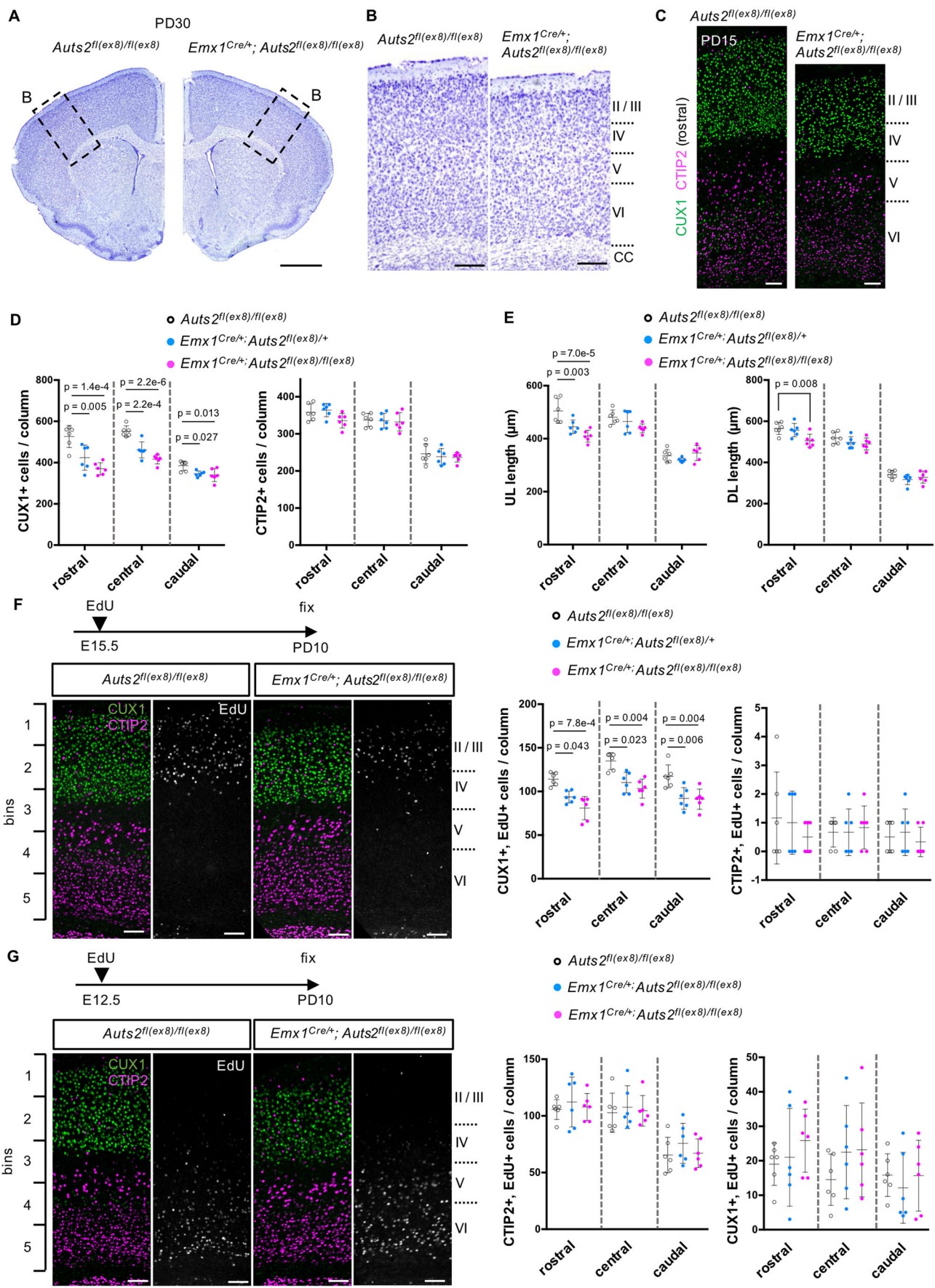

◀

**Figure 1. AUTS2 regulates the production of upper-layer neurons.**

(A) Nissl-stained sections of PD30 brain hemispheres of indicated genotypes. (B) High magnification pictures of rectangle regions in (A), showing the somatosensory cortex. (C) Representative images of immunostaining for CUX1 (green) and CTIP2 (magenta) on the coronal sections at the rostral point of the cerebral cortex in $Auts2^{fl(ex8)/fl(ex8)}$ and $Emx1^{Cre/+}$; $Auts2^{fl(ex8)/fl(ex8)}$ at PD15. (D) Quantification of the number of CUX1$^+$ (left) or CTIP2$^+$ (right) cells in $Auts2^{fl(ex8)/fl(ex8)}$, $Emx1^{Cre/+}$; $Auts2^{fl(ex8)/+}$ and $Emx1^{Cre/+}$; $Auts2^{fl(ex8)/fl(ex8)}$ mice at PD15 within a 300 μm wide field. (E) Quantification of the thickness of upper- or deep-layer in $Auts2^{fl(ex8)/fl(ex8)}$, $Emx1^{Cre/+}$; $Auts2^{fl(ex8)/+}$ and $Emx1^{Cre/+}$; $Auts2^{fl(ex8)/fl(ex8)}$ mice at PD15. (F) EdU was administered intraperitoneally once to E15.5 pregnant mice. Representative images of triple-staining for CUX1 (green), CTIP2 (magenta), and EdU (white) in the cerebral cortices from $Auts2^{fl(ex8)/fl(ex8)}$ and $Emx1^{Cre/+}$; $Auts2^{fl(ex8)/fl(ex8)}$ mice at PD10. The graphs show the number of CUX1$^+$/BrdU$^+$ (left) and CTIP2$^+$/EdU$^+$-double positive cells (right) in $Auts2^{fl(ex8)/fl(ex8)}$, $Emx1^{Cre/+}$; $Auts2^{fl(ex8)/+}$ and $Emx1^{Cre/+}$; $Auts2^{fl(ex8)/fl(ex8)}$ mice within 300 μm wide field, respectively. (G) EdU was administered intraperitoneally once to E12.5 pregnant mice. Representative images of triple-staining for CUX1 (green), CTIP2 (magenta), and EdU (white) in the cerebral cortices from $Auts2^{fl(ex8)/fl(ex8)}$ and $Emx1^{Cre/+}$; $Auts2^{fl(ex8)/fl(ex8)}$ mice at PD10. The graphs show the number of CTIP2$^+$/BrdU$^+$ (left) and CUX1$^+$/EdU$^+$-double positive cells (right) within a 300 μm wide field, respectively. Data were presented as the mean ± SD ($N = 3$ mice, six sections); One-way ANOVA with Dunnett's post hoc test or Kruskal–Wallis test. Scale bars, 1 mm (A), 200 μm (B), and 100 μm (C, F, G). Source data are available online for this figure.

We previously observed that most E14.5-labeled neurons did not reach the pial side by E18.5 in $Auts2$ KO mice ($Auts2^{del(ex8)/del(ex8)}$ and $Auts2^{del(ex8)/+}$), whereas those in wild-type (WT) mice reached this level, suggesting that AUTS2 is involved in neuronal migration (Hori et al, 2014). However, we found that the distribution of E15.5-labeled and E12.5-labeled cells did not differ between controls and homozygotes or heterozygotes at PD10 (Appendix Fig. S1E). This suggests that $Auts2$-deficient neurons migrated slowly but eventually reached their final targets.

Bedogni et al reported that in in situ hybridization experiments, $Auts2$ transcripts were observed in radial glial cells (RGCs) and IPCs (Bedogni et al, 2010). By analyzing previous single-cell RNA-sequencing (scRNA-seq) data from the cerebral cortices at E13.5 and E15.5 (Yuzwa et al, 2017; Data ref: Yuzwa et al, 2017), we found that $Auts2$ transcripts were expressed in postmitotic neurons, RGCs, and IPCs (Appendix Fig. S2A–F). This was also confirmed using another dataset at E12.5 and E15.5 (Appendix Fig. S2G, H) (Di Bella et al, 2021; Data ref: Di Bella et al, 2021). The expression of $Auts2$ was significantly higher in IPCs than in RGCs at E12.5, E13.3 and E15.5 (see the legend for Appendix Fig. S2). These findings suggest that AUTS2 is expressed in neural progenitors (RGCs and IPCs) in the developing cerebral cortex; however, it might be below the detection level in these cells in a previous immunohistochemical experiment (Hori et al, 2014).

## AUTS2 is involved in the proliferation of TBR2$^+$ intermediate progenitor cells

Immunostaining of the E15.5 cerebral cortices with PAX6 antibody, a marker for RGCs, showed that the number of RGCs was not affected in $Auts2^{del(ex8)/+}$ or $Auts2^{del(ex8)/del(ex8)}$ mice (Appendix Fig. S3A). Englund et al previously showed that TBR2 (gene: $Eomes$)-positive cells in the developing cerebral cortex are largely IPCs (Englund et al, 2005); however, they include a small population of immature neurons that have just emerged from IPCs. Immunostaining showed that approximately 85% of TBR2$^+$ cells were KI67$^+$ in the WT cerebral cortex at E15.5 (Appendix Fig. S3B,D), indicating that most TBR2$^+$ cells were IPCs. The number of TBR2$^+$ cells or TBR2/KI67-double positive cells was not affected in $Auts2^{del(ex8)/+}$ or $Auts2^{del(ex8)/del(ex8)}$ mice (Appendix Fig. S3B–D). There was also no significant change in the distribution of TBR2/KI67-double positive cells or putative IPCs in heterozygous or homozygous mutants (Appendix Fig. S3B,E). These findings indicated that the number of neural progenitors (RGCs and IPCs) did not change in the mutants, leading us to investigate the proliferation rate of neural progenitors.

Co-immunostaining for PAX6 and phosphorylated histone H3 (PH3), a mitotic (M)-phase cell marker, to E15.5 cortices showed that the number of dividing RGCs at the ventricular surface did not differ among the genotypes (Fig. 2A). In contrast, the number of mitotic IPCs (TBR2$^+$ and PH3$^+$ cells) was markedly reduced in $Auts2^{del(ex8)/+}$ and $Auts2^{del(ex8)/del(ex8)}$ mice (Fig. 2B). Moreover, short-pulse-labeling with EdU revealed that the proportion of IPCs (TBR2$^+$ and KI67$^+$ cells) in the synthesis phase (S-phase) was significantly decreased in $Auts2^{del(ex8)/+}$ and $Auts2^{del(ex8)/del(ex8)}$ mice at E15.5, whereas that of RGCs (PAX6$^+$ cells) was comparable between the WT and $Auts2$ mutants (Fig. 2C–F). At an earlier embryonic stage (E12.5), we observed no significant differences in the number of M-phase (PH3$^+$) and S-phase (acute EdU$^+$) cells in either the PAX6$^+$ or TBR2$^+$ population between the WT and $Auts2$ mutants (Appendix Fig. S3F–K). Given the observation that the number of IPCs did not differ among genotypes (Appendix Fig. S3B–D) and that IPCs in the M-phase and S-phase were both reduced in $Auts2$ heterozygotes and homozygotes at E15.5, we assumed that the cell cycle length of IPCs might be prolonged in $Auts2$ mutants at this stage.

We then investigated the cell cycle of TBR2$^+$ cells at E15.5, using the dual-labeling method with 5-bromo-2-deoxyuridine (BrdU) and EdU (Fig. 2G) (Watanabe et al, 2015). We found that the length of the overall cell cycle (Tc) of TBR2$^+$ cells was significantly longer in $Auts2^{del(ex8)/+}$ and $Auts2^{del(ex8)/del(ex8)}$ mice than in WT mice, while the S-phase length (Ts) was comparable among the genotypes (Fig. 2G). Most mitotic TBR2$^+$ cells (PH3$^+$, TBR2$^+$) were BrdU$^+$ cells in all genotypes at two hours after BrdU injection, suggesting that G2-length should be within two hours in those cells (Fig. 2H). We also estimated the length of other cell cycle phases (Fig. 2I) (see Methods). G1-lengths for $Auts2^{del(ex8)/+}$ and $Auts2^{del(ex8)/del(ex8)}$ were much longer than that for WT (Fig. 2I). Since a large population of TBR2$^+$ cells was IPCs (Appendix Fig. S3D), and since the number of IPCs (TBR2$^+$ and KI67$^+$ cells) was not affected in $Auts2$ mutants (Appendix Fig. S3C,D), it was suggested that the cell cycle length and G1-length of IPCs were longer in $Auts2$ mutants than those in WT mice. Previously, there have been many reports that G1-length tends to be longer in neurogenic progenitors and shorter in proliferative progenitors (Takahashi et al, 1995; Calegari et al, 2005; Lange et al, 2009; Pilaz et al, 2009). Therefore, the elongated G1-phase in $Auts2$ mutant IPCs may be involved in the reduced proliferation of IPCs and the subsequent production of upper-layer neurons.

## FL-AUTS2 in nuclei is required for neuron production

Next, specific short hairpin RNA (shRNA) for $Auts2$ (Hori et al, 2014) or control scrambled shRNA vector was co-electroporated

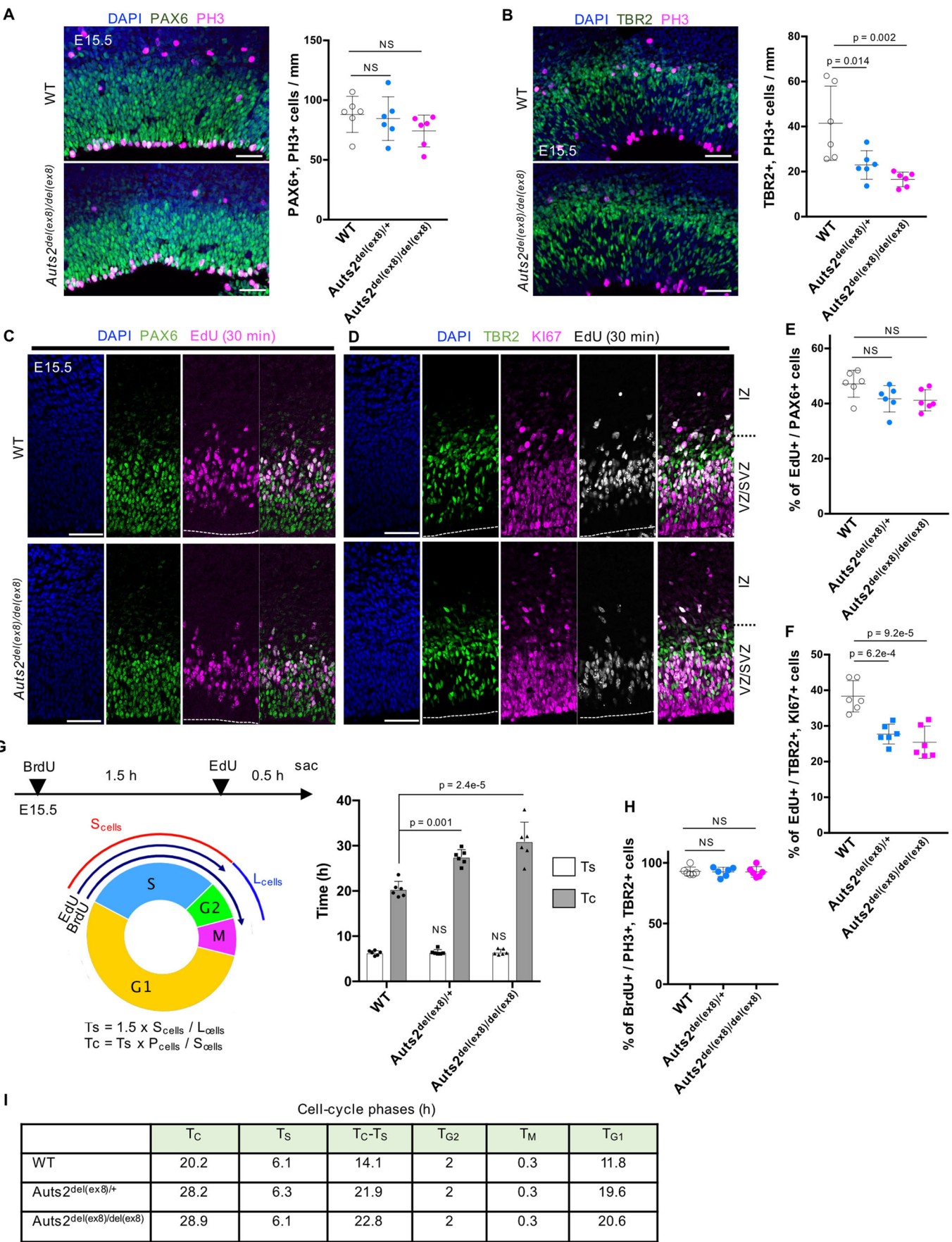

**Figure 2. Loss of *Auts2* leads to the reduction of intermediate progenitor mitoses.**

(A) Representative images of immunostaining for DAPI (blue), PAX6 (green) and PH3 (magenta) in E15.5 WT and *Auts2*$^{del(ex8)/del(ex8)}$ cerebral cortices. The graph shows the number of PAX6$^+$ and PH3$^+$ cells on the ventricular surface in WT, *Auts2*$^{del(ex8)/+}$, and *Auts2*$^{del(ex8)/del(ex8)}$ mice. (B) Representative images of immunostaining for DAPI (blue), TBR2 (green), and PH3 (magenta) in E15.5 WT and *Auts2*$^{del(ex8)/del(ex8)}$ cerebral cortices. The graph shows the number of TBR2$^+$ and PH3$^+$ cells in WT, *Auts2*$^{del(ex8)/+}$, and *Auts2*$^{del(ex8)/del(ex8)}$ mice. (C–F) EdU was administered intraperitoneally to pregnant mice 30 min before sacrifice at E15.5. (C) Representative images of staining for DAPI (blue), PAX6 (green) and EdU (magenta) in WT and *Auts2*$^{del(ex8)/del(ex8)}$ cortical sections. (D) Representative images of staining for DAPI (blue), TBR2 (green), KI67 (magenta), and EdU (white) in WT and *Auts2*$^{del(ex8)/del(ex8)}$ cortical sections. (E, F) The graphs show the percentage of EdU$^+$ cells among PAX6$^+$ cells (E) or TBR2$^+$, KI67$^+$ cells (F) in WT, *Auts2*$^{del(ex8)/+}$ and *Auts2*$^{del(ex8)/del(ex8)}$ mice. VZ ventricular zone, SVZ subventricular zone, IZ intermediate zone. (G, H) The cell cycle length estimate in TBR2$^+$ cells using the BrdU and EdU double-labeling method at E15.5. BrdU and EdU are administered to pregnant mice 2 h and 30 min before sacrifice. Ts, S-phase length; Tc, cell cycle length; L$_{cells}$, BrdU$^+$, and EdU-negative cells; S$_{cells}$, EdU$^+$ cells; P$_{cells}$, TBR2$^+$ cells. (H) The graph shows the percentage of BrdU$^+$ cells among PH3$^+$ and TBR2$^+$ cells at E15.5. (I) Cell-cycle parameters of TBR2$^+$ cells in E15.5 cortex. T$_{G2}$: G2-phase length, T$_M$: M-phase, T$_{G1}$: G1-phase. The number of cells was quantified at the rostral point. Data were presented as the mean ± SD (N = 3 mice, six sections). NS not significant, One-way ANOVA with Dunnett's post hoc test. Scale bars, 100 μm (A, B), 50 μm (C, D). Source data are available online for this figure.

with a histone H3-fused green fluorescent protein (GFP) (H3-GFP) vector into the VZ of WT cortices at E14.5, followed by immunohistochemical analysis with KI67 and HuC/D (a neuronal marker) at E16.5. Under these experimental conditions in control, 36.9 ± 1.1% (mean ± SEM) of the electroporated cells were post-mitotic neurons (KI67-negative and HuC/D$^+$) produced from neural progenitors (Fig. 3A,B). In contrast, introducing *Auts2* shRNA reduced this proportion, consistent with decreased neuron production in *Auts2* cKO cortices (Fig. 1D). However, the reduced proportion of postmitotic neurons in *Auts2*-shRNA introduced cortices was fully rescued by the co-expression of shRNA-resistant full-length AUTS2 (FL-AUTS2$^R$) (Fig. 3B). This rescuing effect was also observed for nuclear localization signal (NLS)-tagged FL-AUTS2$^R$ (NLS-FL-AUTS2$^R$) (Hori et al, 2020) but not for the nuclear export sequence (NES)-tagged FL-AUTS2$^R$ (NES-FL-AUTS2$^R$) (Hori et al, 2014) (Fig. 3B,C). This suggests that nuclear-localizing AUTS2 is required for this developmental event. We also reported that AUTS2 short variants (var.1 and var.2) are expressed in the developing mouse cerebral cortex (Hori et al, 2014). However, the short variants did not rescue the reduction in postmitotic neurons (Fig. 3B,C), indicating that these short variants were not involved in this event. These findings suggest that the nuclear-localizing FL-AUTS2 is vital in neuronal production from progenitors. We also evaluated the rates of PAX6$^+$ cells (RGCs) and TBR2$^+$, KI67$^+$ cells (IPCs) in all the electroporated cells (GFP$^+$ cells) (Fig. 3D,E), but no significant difference was observed.

## Loss of *Auts2* alters the expression of cell differentiation and cell proliferation-related genes in IPCs

To isolate IPCs from the developing cerebral cortex by fluorescence-activated cell sorting (FACS), we newly generated a knock-in (KI) mouse line, in which a destabilized EGFP (d2EGFP) gene with a T2A self-cleaving peptide sequence was inserted into *Eomes* (coding TBR2 protein) genomic locus (Appendix Fig. S4A–C). TBR2 and d2EGFP were bicistronically expressed in this mouse line in the IPCs. Indeed, we confirmed that the TBR2 and d2EGFP proteins were efficiently cleaved in cortical lysates from *Eomes*$^{T2A-d2EGFP}$ KI embryos at E14.5 (Appendix Fig. S4D). It has been reported that *Eomes*-deficient homozygous mice showed embryonic lethality, reduced cortical size, and severe hypoplasia in olfactory bulbs (Sessa et al, 2008). In contrast, *Eomes*$^{T2A-d2EGFP/T2A-d2EGFP}$ homozygotes were born normally and exhibited a normal-sized cerebral cortex and olfactory bulb compared with WT mice (Appendix Fig. S4E). Immunostaining with GFP, TBR2, SOX2, and HuC/D showed that most EGFP signals (93.6 ± 1.2%;

mean ± SEM) were specifically detected in TBR2-immunopositive cells (Appendix Fig. S4F,G), suggesting that most IPCs were specifically labeled with EGFP without interfering with TBR2 protein expression and function in KI mice.

Using this mouse line, we tried to isolate IPCs and RGCs derived from the cerebral cortices of control (*Eomes*$^{T2A-d2EGFP/+}$; *Auts2*$^{fl(ex8)/fl(ex8)}$) or *Auts2* cKO homozygotes (*Eomes*$^{T2A-d2EGFP/+}$; *Emx1*$^{Cre/+}$; *Auts2*$^{fl(ex8)/fl(ex8)}$) at E15.5, by FACS with EGFP-positivity (EGFP$^+$ cells) and high expression of CD133 (PROM1, CD133$^{high}$ cells), respectively (Fig. 4A,B; Appendix Fig. S4H). Next, we performed RNA-sequencing (RNA-seq) on each isolated population. The expression of RGC marker genes, such as *Sox2*, *Nes*, *Pax6*, and *Hes1*, was enriched in CD133$^{high}$ cells (Fig. 4C), which were confirmed by the real-time quantitative PCR (RT-qPCR) analyses (Appendix Fig. S4I). Immunocytochemistry revealed that most CD133$^{high}$ cells expressed SOX2 but not TBR2 (Fig. 4D). These observations indicated that most CD133$^{high}$ cells were RGCs. In contrast, most EGFP$^+$ cells were immunoreactive to TBR2 (Fig. 4D). This suggests that cell sorting in *Eomes*$^{T2A-d2EGFP/+}$ KI mice was efficiently performed. In the RNA-seq and RT-qPCR analyses of EGFP$^+$ cells, *Eomes* expression was concentrated, as expected (Fig. 4C; Appendix Fig. S4I). In the EGFP$^+$ cell population, expression of immature neuronal markers, such as *NeuroD1* and *Tubb3*, was detected compared to CD133$^{high}$ cells (Fig. 4C; Appendix Fig. S4I). This may reflect the small population (15%) of postmitotic cells in TBR2$^+$ cells (Appendix Fig. S3D), consistent with reports that TBR2$^+$ cells contain a small proportion of postmitotic immature neurons (Englund et al, 2005; Arai et al, 2011). Immunocytochemistry with KI67 showed that 98.8 ± 0.3% (mean ± SEM) of CD133$^{high}$ cells and 92.0 ± 1.5% (mean ± SEM) of GFP-positive cells were KI67-positive proliferating cells (Appendix Fig. S4J). These findings suggest that the EGFP$^+$ cell population mainly includes IPCs. *Auts2* expression was more enriched in EGFP$^+$ cells than in CD133$^{high}$ cells (Fig. 4C), suggesting that *Auts2* expression was higher in IPCs than in RGCs. This was also supported by the scRNA-seq data (Appendix Fig. S2).

In RNA-seq analysis, we identified 8619 differentially expressed genes (DEGs) between CD133$^{high}$ cells (mostly RGCs) and EGFP$^+$ cells (mainly IPCs) (Appendix Fig. S5A; Dataset EV1), which may reflect the different cell characters. In EGFP$^+$ cells, we found that the expression of 105 genes was significantly upregulated, and that of 140 genes was downregulated in cKO homozygotes compared with the control (Fig. 4E,F; Dataset EV1). We named these genes "IPC-upregulated-genes" and "IPC-downregulated-genes," respectively.

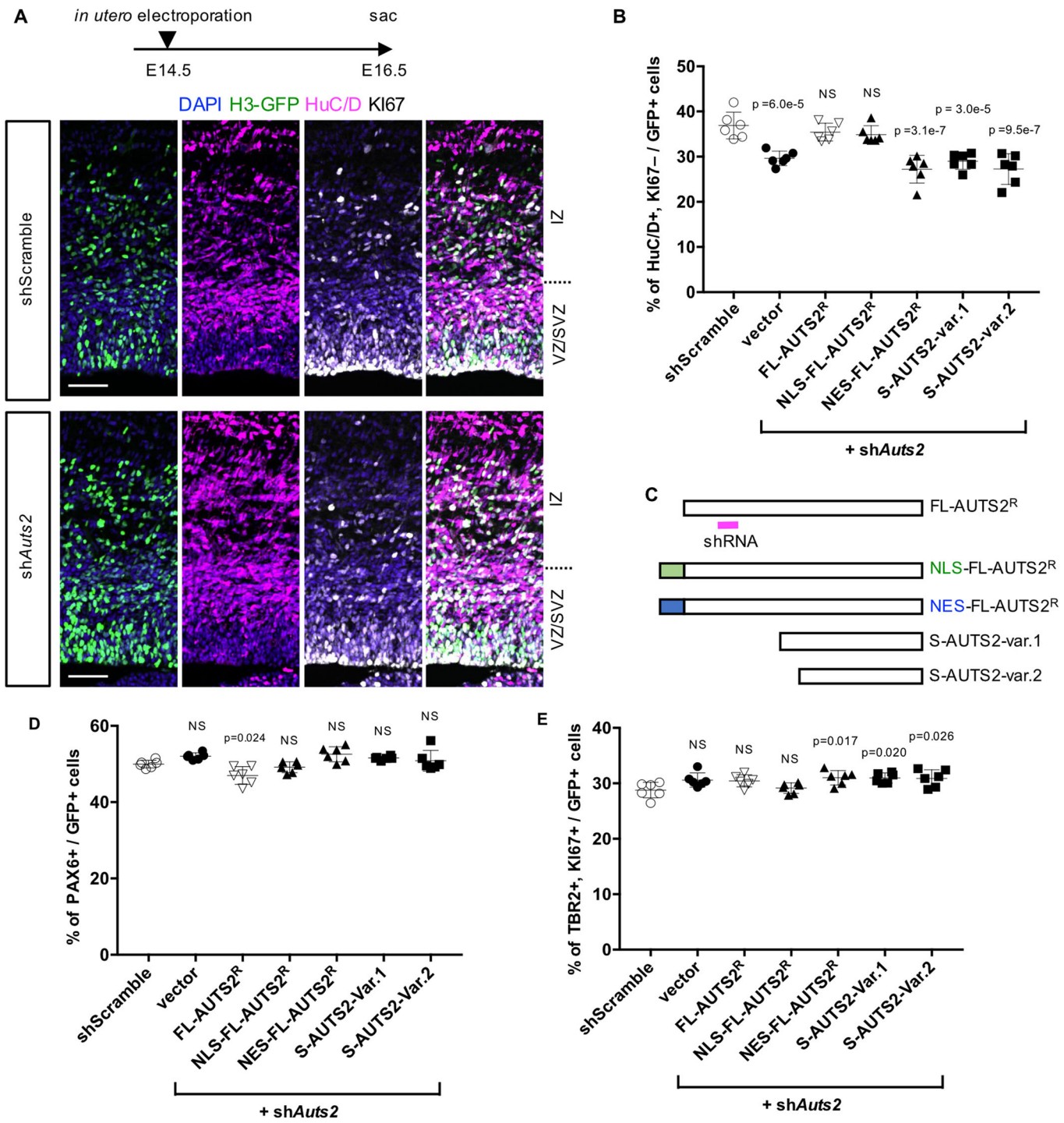

**Figure 3. Nuclear AUTS2 is involved in neuronal production from neural progenitor cells.**

WT mouse embryos at E14.5 were co-electroporated with scrambled shRNA (shScramble) or *Auts2* shRNA (sh*Auts2*) together with empty or AUTS2 expression vectors indicated in (**C**) in utero and analyzed at E16.5. (**A**) Representative images of immunostaining for DAPI (blue), H3-GFP (green), HuC/D (magenta), and KI67 (white) in the dorsolateral cerebral cortices of mice electroporated with shScramble and sh*Auts2* vectors. Scale bars, 50 μm. (**B**) The graph shows the percentage of HuC/D[+] and KI67-negative cells among GFP[+] cells. (**C**) Schematic diagrams of shRNA-resistant full-length-AUTS2 (FL-AUTS2[R]), NLS-FL-AUTS2[R], NES-FL-AUTS2[R], and C-terminal AUTS2 short variants (S-AUTS2-var.1 and var.2). The magenta bar indicates the shRNA binding site. NLS nuclear localization signal, NES nuclear export sequence. (**D**) The graph shows the percentage of PAX6[+] cells in GFP[+] cells. (**E**) The graph shows the percentage of TBR2[+] and KI67[+] cells in GFP[+] cells. Data were presented as the mean ± SD (*N* = 3 mice, six sections); One-way ANOVA with Dunnett's post hoc test. Source data are available online for this figure.

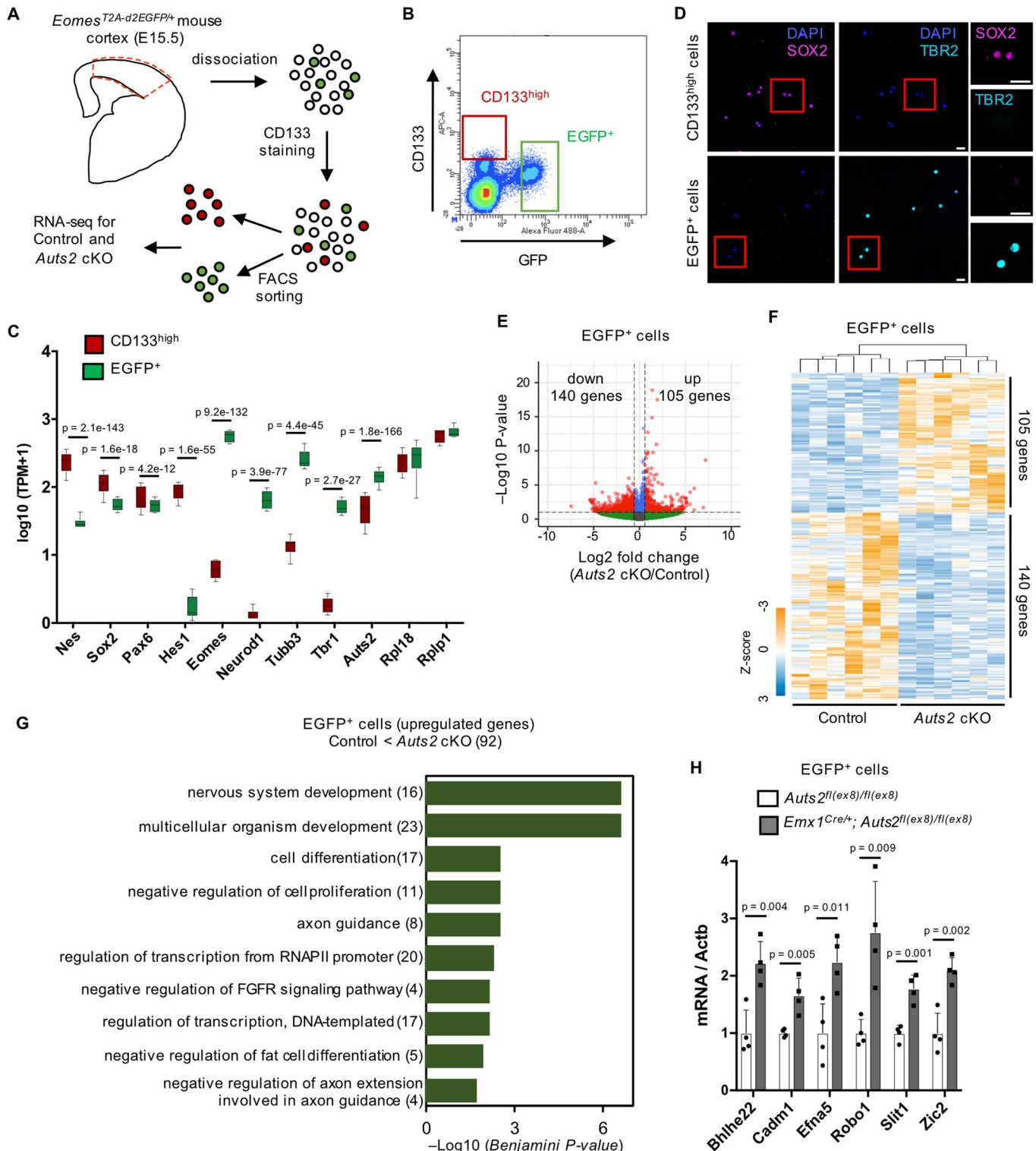

Gene Ontology (GO) term analysis demonstrated that IPC-upregulated-genes were associated with biological processes, including "nervous system development," "cell differentiation," and "negative regulation of cell proliferation" (Fig. 4G; Dataset EV2), implying that AUTS2 functions in IPCs to regulate proliferation and/or differentiation. We performed RT-qPCR using

EGFP+ cells for some of the IPC-upregulated-genes, confirming that their expression was upregulated in the cKO compared with that in control (Fig. 4H). There was no significant GO term for the IPC-downregulated-genes (Dataset EV2). We also identified DEGs in sorted CD133high cells between control and cKO mice. The expression of 192 genes was significantly upregulated, whereas that

**Figure 4.  Transcriptional profiling of IPCs in control and *Auts2* cKO mice at E15.5.**

(A) Experimental design for transcriptional analysis. CD133$^{high}$ and EGFP$^+$ cells were sorted from *Eomes*$^{T2A-d2EGFP/+}$; *Auts2*$^{fl(ex8)/fl(ex8)}$ (Control) or *Eomes*$^{T2A-d2EGFP/+}$; *Emx1*$^{Cre/+}$; *Auts2*$^{fl(ex8)/fl(ex8)}$ (*Auts2* cKO) cortices at E15.5. (B) Representative plot showing sorting gates for CD133$^{high}$ and EGFP$^+$ cells in *Eomes*$^{T2A-d2EGFP/+}$; *Auts2*$^{fl(ex8)/fl(ex8)}$. (C) Expression levels of *Nes, Sox2, Pax6, Hes1, Eomes, Neurod1, Tubb3, Tbr1, Fezf2, Auts2, Rpl18*, and *Rplp1* transcripts in sorted each cell type from E15.5 control mouse cortex. *Rpl18* and *Rplp1* are presented as housekeeping genes. TPM transcripts per million. Adjusted *P* value calculated by DESeq2 was shown. Data were presented as box-and-whisker plots (medians with interquartile range, minimum, and maximum values are represented). (D) Immunostaining of CD133$^{high}$ and EGFP$^+$ cells sorted by FACS from E15.5 *Eomes*$^{T2A-d2EGFP/+}$ cortex with DAPI (blue), anti-SOX2 (magenta) and anti-TBR2 (cyan) antibodies. Scale bars, 20 μm. (E) Volcano plot showing differences in gene expression in EGFP$^+$ cells between control and *Auts2* cKO mice. Red dots indicate upregulated and downregulated genes (*P* value <0.01 and |Log$_2$ fold-change| >0.58). *P* value was calculated by Wald test with DESeq2 default parameter. (F) Heatmap for the RNA levels of differential expressed genes (DEGs) in EGFP$^+$ cells from control and *Auts2* cKO mice. The color scale is shown on the left bottom. (G) DAVID Gene Ontology biological process analysis of upregulated genes in EGFP$^+$ cells. The top ten GO terms with Benjamini–Hochberg adjusted *P* value are shown. Numbers in parentheses indicate the gene count. (H) RT-qPCR analysis for *Bhlhe22, Cadm1, Efna5, Robo1, Slit1*, and *Zic2* in sorted EGFP$^+$ cells from control and *Auts2* cKO mice at E15.5. Data were presented as the mean ± SD (*N* = 4 mice); unpaired Student's *t*-test. All transcriptome analyses used six biological replicates per genotype. Source data are available online for this figure.

of 190 genes was downregulated (Appendix Fig. S5B; Dataset EV1), which gave one GO term (Appendix Fig. S5C; Dataset EV2). Notably, the DEGs for CD133$^{high}$ and EGFP$^+$ cells did not overlap very much (Appendix Fig. S5D), suggesting that AUTS2 might control the expression of different genes in RGCs and IPCs.

## AUTS2 is involved in IPC proliferation by suppressing *Robo1* expression

As we observed a significant phenotype in IPCs, we concentrated on analyzing IPCs in this study. Among IPC-upregulated and IPC-downregulated genes, the *Robo1* gene was investigated. The *Robo1* gene was linked to the GO terms "nervous system development" and "negative regulation of cell proliferation" (Fig. 4G). There have already been reports on the roles of ROBO signaling in RGCs. This signaling promotes direct neurogenesis from RGCs and suppresses indirect neurogenesis, or the production of IPCs (Borrell et al, 2012; Cardenas et al, 2018). However, the role of ROBO signaling in IPCs is still poorly understood. We found that expression of this gene increased approximately 2.5-fold in *Auts2* cKO EGFP$^+$ cells compared with that in control (Fig. 4H). Therefore, we suspected that this overexpressed *Robo1* might affect the division of IPCs and the production of neurons. Furthermore, to evaluate ROBO1 protein levels, we performed immunocytochemistry for ROBO1 to the sorted EGFP$^+$ cells and estimated the intensity of ROBO1. As a result, we found that the immunofluorescent intensities of ROBO1 were significantly stronger in heterozygous and homozygous EGFP$^+$ cells than in control cells (Appendix Fig. S6A), consistent with the result of the RT-qPCR experiment (Fig. 4H).

Next, we electroporated the overexpression (OE) vector for *Robo1* into the VZ of WT cortices at E13.5. We analyzed them at E15.5 when the reduction in IPC division was observed in the *Auts2* mutant (Fig. 2B). We found that *Robo1* OE decreased the ratio of mitotic cells (PH3$^+$) among GFP$^+$, TBR2$^+$ cells compared with the control (Fig. 5A). This effect was not observed in GFP-negative, TBR2$^+$ cells in the brain region electroporated with *Robo1* OE vector (Fig. 5A), suggesting that ROBO1 works in a cell-autonomous manner. In addition, electroporation of *Robo1* OE vector at E14.5 resulted in reduced postmitotic (KI67-negative and GFP$^+$) cells at E16.5 (Fig. 5B), consistent with our observation that electroporation of the *Auts2* KD vector decreased the proportion of neuronal cells at two days after electroporation (Fig. 3A,B). On the other hand, *Robo1* OE did not significantly affect the cell division (PH3-positivity) in presumable RGCs (TBR2-negative cells localized to the VZ) (Appendix Fig. S6B) or the distribution of TBR2$^+$ cells (Appendix Fig. S6C).

In order to overexpress *Robo1* specifically in TBR2$^+$ cells, but not in RGCs, we further generated the *Eomes*$^{T2A-iCre}$ knock-in (KI) allele (Appendix Fig. S6D,E) using the same method as that used to create the *Eomes*$^{T2A-d2EGFP}$ KI allele (Appendix Fig. S4A). In mice carrying this allele, iCre (improved Cre) was expected to be expressed in cells expressing *Eomes* (TBR2). We first electroporated the pCAG-LSL (loxP-Stop-loxP)-H3GFP expression vector plus pCAG-RFP vector into the VZ of *Eomes*$^{T2A-iCre/+}$ mice at E14.5. One day after electroporation, H3-GFP signals were detected in 90 ± 0.8% (mean ± SEM) of the electroporated TBR2$^+$ cells (TBR2$^+$, RFP$^+$) but hardly observed in TBR2-negative cells located in the VZ (Appendix Fig. S6F,G). This suggests that the method using *Eomes*$^{T2A-iCre/+}$ mice and pCAG-LSL vector works well in the overexpression of a certain gene, specifically in IPCs.

Next, we co-electroporated pCAG-LSL-*Robo1* and pCAG-H3GFP vectors into *Eomes*$^{T2A-iCre/+}$ KI embryos at E13.5 and examined the percentage of PH3$^+$ cells in TBR2$^+$, GFP$^+$ cells at two days after electroporation (Appendix Fig. S6H). Compared to the control, we found that the percentage of PH3$^+$ cells decreased in cells that had been introduced with pCAG-LSL-*Robo1*. This suggests that upregulated *Robo1* expression suppresses the proliferation of IPCs. We also observed the upregulated expression of the *Slit1* gene in the sorted EGFP$^+$ cells (Fig. 4H). It is known that SLIT1 can act as a ligand for ROBO1 (Brose et al, 1999; Li et al, 1999). To investigate the role of SLIT1, we electroporated pCAG-*Slit1* with pCAG-H3GFP into WT cortices at E13.5 and analyzed the animals at E15.5 (Appendix Fig. S6I). We found that *Slit1* overexpression also reduced the proportion of PH3$^+$ cells in TBR2$^+$ and GFP$^+$ cells, suggesting that upregulated SLIT1-ROBO1 signaling downregulates IPC division.

We performed a rescue experiment by in utero electroporation of *Robo1* KD vectors into *Auts2* cKO cortices to test whether the reduced IPC proliferation and neuron production in *Auts2* KO mice was caused by increased expression of *Robo1*. We generated two *Robo1*-specific shRNAs that effectively depleted the ROBO1 protein (Appendix Fig. S6J). Each *Robo1* shRNA or scrambled shRNA was electroporated into the VZ of *Auts2* cKO homozygous cortices at E13.5, and immunostaining for TBR2 and PH3 was performed two days after electroporation (E15.5). We found that the ratio of mitotic cells in electroporated TBR2$^+$ cells in *Auts2* cKO cortices was efficiently recovered by electroporation with either *Robo1* shRNA to the same degree as in the scrambled shRNA-transfected control (*Auts2*$^{fl(ex8)/fl(ex8)}$) cortices (Fig. 5C). Next, we performed another rescue experiment in which each *Robo1* shRNA or scrambled shRNA was electroporated into the VZ of *Auts2* cKO

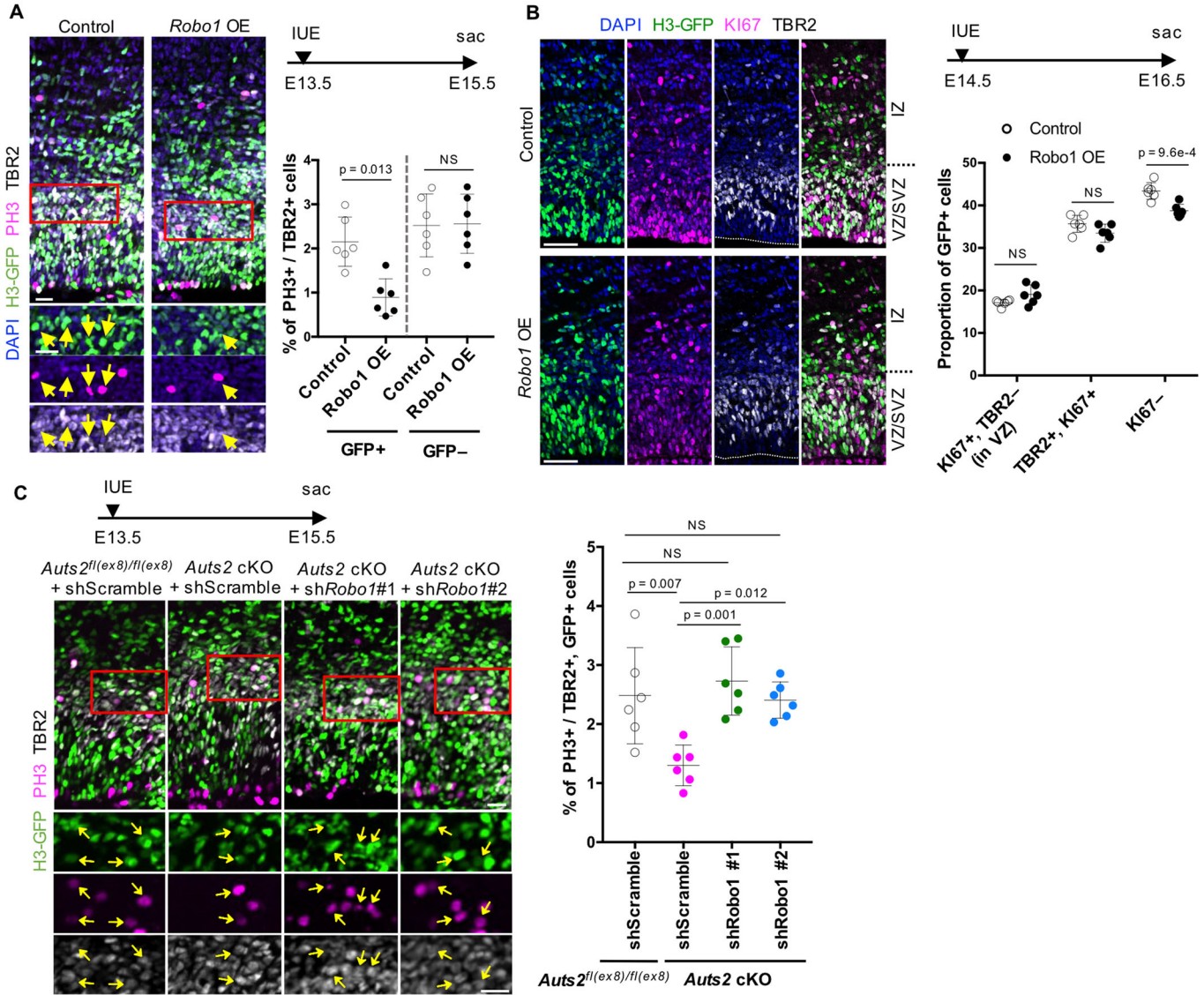

**Figure 5. Elevated expression of *Robo1* leads to a decrease in IPC proliferation.**

(A) In utero electroporation (IUE) of either an empty vector (control) or a *Robo1* expression vector (*Robo1* OE) together with a H3-GFP expression vector was performed into WT cortices at E13.5. The cortical sections are stained with DAPI (blue), anti-GFP (green), anti-PH3 (magenta), and anti-TBR2 (white) antibodies at E15.5. The graph shows the ratio of PH3+ cells in TBR2+ and GFP+ cells (left) or PH3+ cells in TBR2+ and GFP-negative cells (right). Arrows indicate GFP+, PH3+, and TBR2+ cells. (B) IUE of an empty (control) or a *Robo1* expression vector (*Robo1* OE) and a H3-GFP expression vector into WT cortices at E14.5. The cortical sections are immunostained with DAPI (blue), anti-GFP (green), anti-KI67 (magenta), and anti-TBR2 (white) antibodies at E16.5. The graph shows the percentage of KI67+ and TBR2-negative cells located in the VZ (RGCs), TBR2+ and KI67+ cells (IPCs), and KI67-negative differentiated cells (postmitotic neurons) among total GFP+ cells. The percentage of RGCs and IPCs in electroporated cells was not different between *Robo1* OE and control. (C) *Auts2*fl(ex8)/fl(ex8) or *Emx1*Cre/+; *Auts2*fl(ex8)/fl(ex8) (*Auts2* cKO) cortices were electroporated with scrambled or *Robo1* shRNA vector in utero at E13.5 and immunostained with anti-GFP (green), anti-PH3 (magenta) and anti-TBR2 (white) antibodies at E15.5. The graph shows the percentage of PH3+ cells among TBR2+ and GFP+ cells. Arrows indicate GFP+, PH3+, and TBR2+ cells. Data were presented as the mean ± SD (N = 3 mice, six sections). NS, not significant, unpaired Student's t-test (A, B) and one-way ANOVA with Turkey's post hoc test (C). Scale bars, 20 μm (A, C) and 50 μm (B). Source data are available online for this figure.

homozygous cortices at E14.5, and immunostaining for KI67 and TBR2 was performed at E16.5. Among the electroporated cells, the reduction in KI67-negative differentiated cells in *Auts2* cKO cortices was efficiently rescued by introducing either *Robo1* shRNA (Appendix Fig. S6K). These findings suggest that AUTS2 promotes the proliferation of IPCs and the production of an appropriate number of neurons by repressing *Robo1* expression.

## AUTS2 represses the expression of genes related to neuronal differentiation in IPCs by maintaining repressive chromatin status

AUTS2 has been reported to regulate gene expression in the cell nuclei (Gao et al, 2014; Liu et al, 2021; Li et al, 2022; Monderer-Rothkoff et al, 2021; Russo et al, 2018); therefore, we next tried to

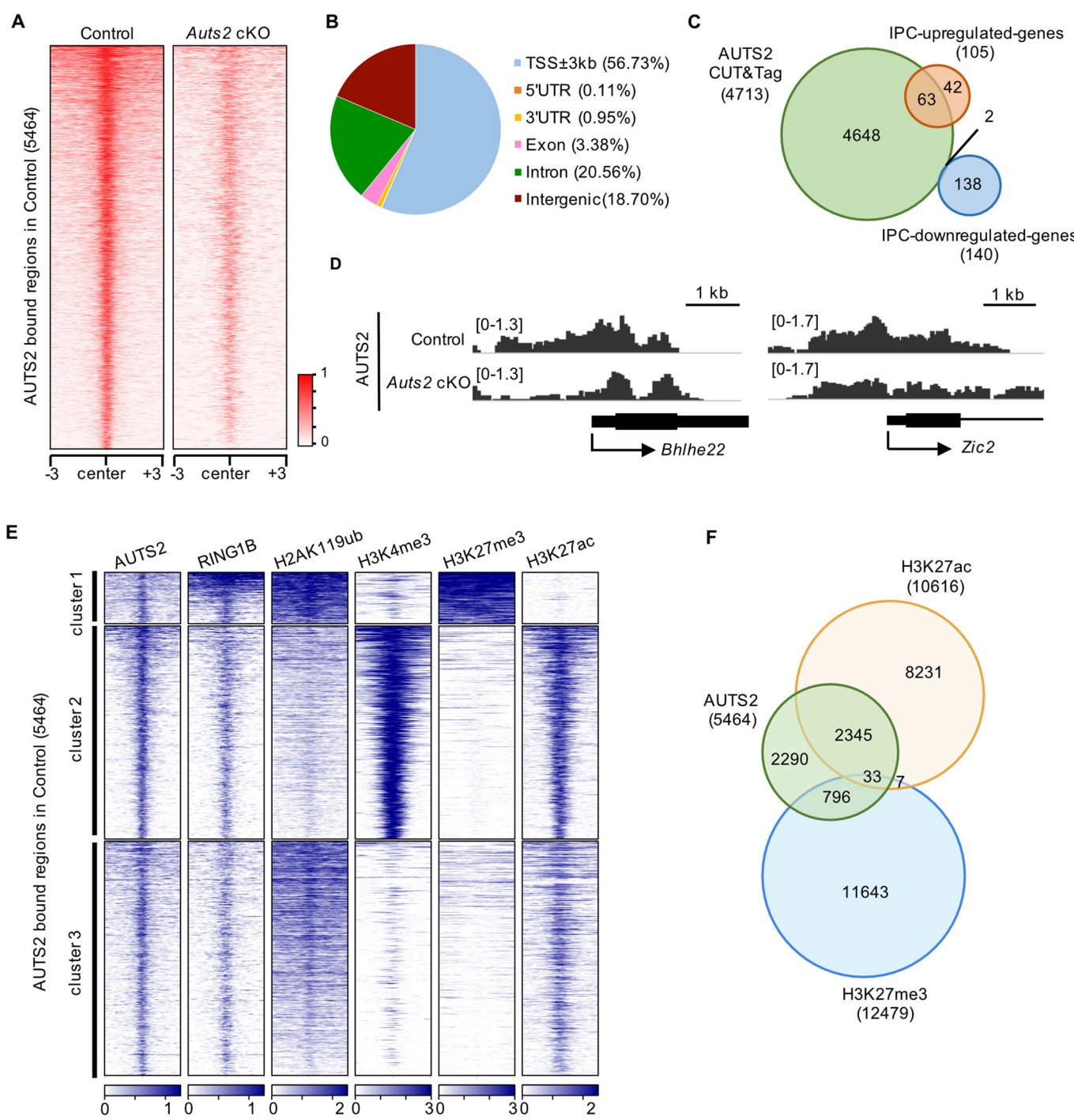

Figure 6. AUTS2 binds to the active chromatin and the repressive chromatin.

(A) Heatmap showing the CUT&Tag signals of AUTS2 in control (left) and *Auts2* cKO homozygous (right) cells centered on AUTS2-binding loci (±3 kb) identified in control at E15.5. (B) Pie chart showing the percentage of AUTS2-binding loci on each genomic region. (C) Genes near the AUTS2-binding loci were identified using the genomic regions enrichment of annotations tool (GREAT). Venn diagrams show the overlap between those genes and IPC-upregulated or IPC-downregulated genes identified by RNA-seq. (D) Interactive genomics viewer (IGV) browser views showing the CUT&Tag signal for AUTS2 in control and *Auts2* cKO cells at the indicated loci. (E) K-means clustering of AUTS2, RING1B, H2AK119ub, H3K4me3, H3K27me3, and H3K27ac CUT&Tag signals in control cells centered on AUTS2 binding peaks (±3 kb) identified in control. Fifteen genes with IPC-upregulated-TSSs (see Fig. 7A) were included in 558 loci of cluster 1. Fifteen and 4 genes with IPC-upregulated-TSSs were in 2340 and 2566 loci of clusters 2 and 3, respectively. (F) Venn diagrams showing the extent of overlap for AUTS2-binding loci, H3K27ac- and H3K27me3-modified loci.

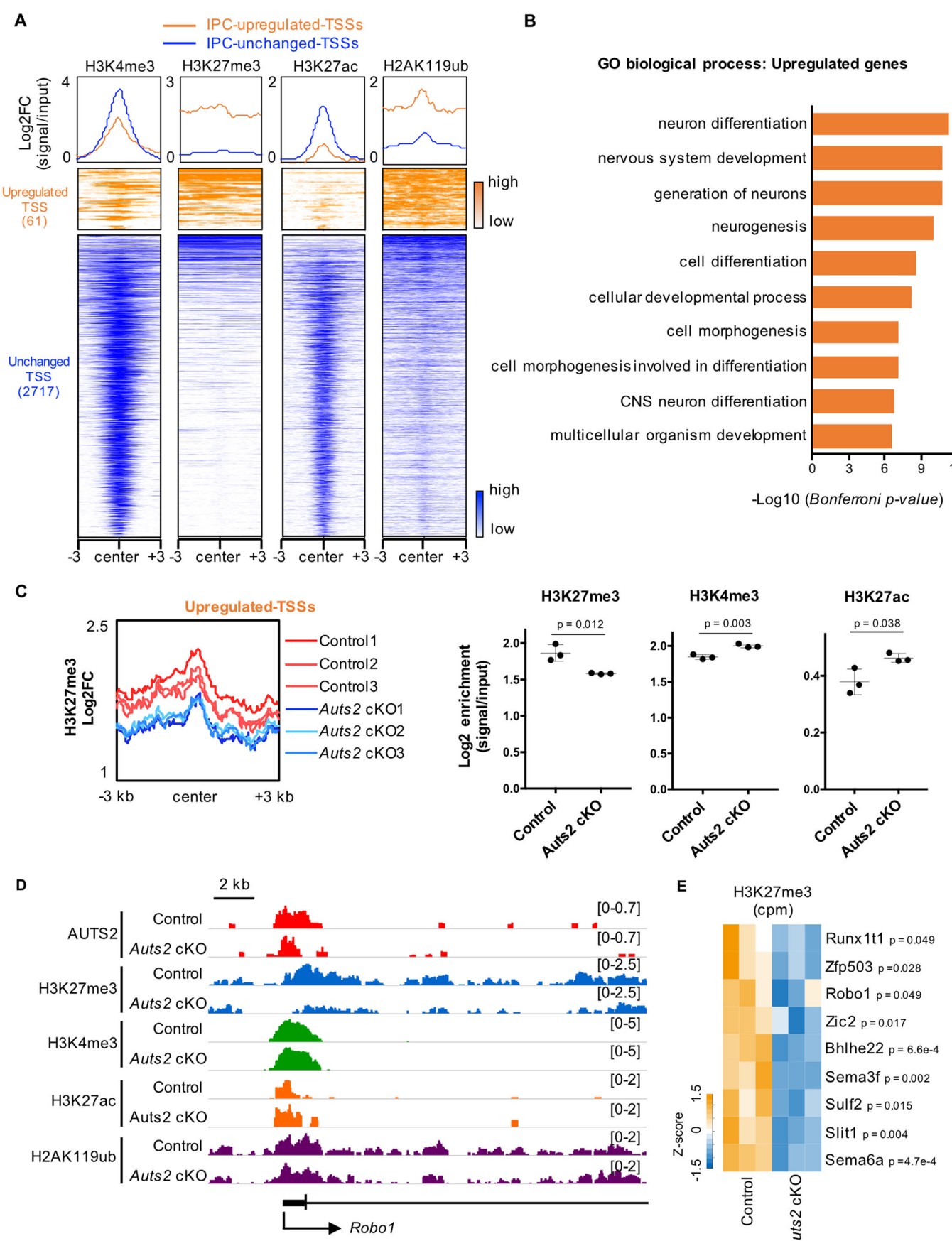

**Figure 7. AUTS2 maintains the repressive chromatin status of genes related to neuronal differentiation.**

(A) Density plots and heatmap of H3K4me3, H3K27me3, H3K27ac, and H2AK119ub CUT&Tag signals centered on AUTS2-binding loci (±3 kb) localized near IPC-upregulated-TSSs (orange) and IPC-unchanged-TSSs (blue). FC fold-change. (B) Graph showing the top 10 GO-term biological processes for genes with IPC-upregulated-TSSs and Bonferroni adjusted $P$ values for each GO-term on GREAT. (C) The density profile shows H3K27me3 occupancy in control and *Auts2* cKO cells centered on IPC-upregulated-TSSs (±3 kb). Graphs showing the enrichment of H3K27me3, H3K4me3, and H3K27ac on IPC-upregulated-TSSs. N = 3 biological replicates. Data were presented as the mean ± SD; unpaired Student's $t$-test. (D) IGV browser views showing the CUT&Tag signals for AUTS2, H3K27me3, H3K4me3, H3K27ac, and H2AK119ub in control and *Auts2* cKO cells at *Robo1* locus. H3K4me3 and H3K27ac levels were significantly increased (H3K4me3, $p = 0.003$; H3K27ac, $p = 0.039$). H2AK119ub level was not significant ($p = 0.182$). N = 3 biological replicates, unpaired Student's $t$-test. (E) Heatmap of count per million (cpm) for H3K27me3 on AUTS2-binding loci located near the representative IPC-upregulated-genes related to "neuron differentiation (GO:0030182)" and/or "cell differentiation (GO:0030154)." Although significant reduction at the *Robo1* locus was found for *Auts2* cKO homozygotes, it was not observed for *Auts2* heterozygotes ($p = 0.384$). N = 3 biological replicates, unpaired Student's $t$-test.

identify AUTS2-interacting genomic regions in IPCs by conducting a cleavage under targets and tagmentation (CUT&Tag) experiment (Kaya-Okur et al, 2019) to the sorted EGFP+ cells from control (*Eomes*$^{T2A-d2EGFP/+}$; *Auts2*$^{fl(ex8)/fl(ex8)}$), cKO homozygous or heterozygous (*Eomes*$^{T2A-d2EGFP/+}$; *Emx1*$^{Cre/+}$; *Auts2*$^{fl(ex8)/fl(ex8)}$ or *Eomes*$^{T2A-d2EGFP/+}$; *Emx1*$^{Cre/+}$; *Auts2*$^{fl(ex8)/+}$) cortices at E15.5 using the anti-AUTS2 antibody, followed by deep-sequencing. In the control cells, we identified multiple genomic loci to which AUTS2 bound significantly and strongly (Fig. 6A). In contrast, global levels of AUTS2 binding were reduced in *Auts2* mutants (Fig. 6A; Appendix Fig. S7A), suggesting that the CUT&Tag experiment worked efficiently. However, the signals in the cKO homozygotes were not completely lost. A truncated isoform (S-AUTS2-var.2) is expressed even in *Auts2* cKO homozygotes (Hori et al, 2014), and the AUTS2 antibody can recognize that isoform; therefore, the signals for cKO homozygotes might correspond to loci bound to S-AUTS2-var.2. In this experiment, we successfully identified 5464 significant AUTS2-binding loci in control cells.

AUTS2-binding loci were predominantly detected near or within genes (81.30%), especially the ±3 kb region around transcriptional start sites (TSS) (56.73%) and occasionally in introns, exons and untranslated regions (UTRs). However, only 18.70% was found in intergenic regions (Fig. 6B). We identified 4713 genes in the vicinity of AUTS2 binding loci, which included 63 of 105 IPC-upregulated-genes (60%) but only 2 of 140 IPC-downregulated genes (1.4%) (Fig. 6C). Previous studies have shown that AUTS2 is involved in transcriptional activation and repression in a cell type-dependent manner (Monderer-Rothkoff et al, 2021). These findings suggest that AUTS2 might be involved in suppressing gene transcription in IPCs and that the 63 genes, which included *Robo1*, might be direct target candidates of AUTS2. Among them, *Bhlhe22* and *Zic2* were representative genes in which AUTS2 binding signals were abundant around the TSS in the control but were reduced in *Auts2* cKO homozygotes (Fig. 6D). A similar tendency was also observed for the binding of AUTS2 to the region around the TSS of the *Robo1* gene (Fig. 7D).

As previous reports have shown that AUTS2 interacts with PRC1.3/5 complexes (Gao et al, 2014; Liu et al, 2021; Monderer-Rothkoff et al, 2021), we performed CUT&Tag experiments on RING1B, a core subunit of PRC1, in sorted EGFP+ cells from control mice (*Eomes*$^{T2A-d2EGFP/+}$; *Auts2*$^{fl(ex8)/fl(ex8)}$) at E15.5. Approximately 60% of AUTS2-binding loci were also RING1B-binding loci (Appendix Fig. S7B). We conducted CUT&Tag experiments for H2AK119ub, H3K4me3, H3K27me3, and H3K27ac in the sorted EGFP+ cells to investigate the histone modifications around the AUTS2-binding loci in IPCs. K-means clustering of the data for

AUTS2, RING1B, H2AK119ub, H3K4me3, H3K27me3, and H3K27ac showed that AUTS2-binding loci could be classified into three clusters (Fig. 6E). Clusters 1, 2, and 3 contained 558, 2340, and 2566 loci, respectively. Cluster 1 loci were suggested to correspond to repressive chromatin because they were enriched with H3K27me3 and H2AK119ub but not with H3K4me3 or H3K27ac. In contrast, cluster 2 loci may correspond to active chromatin because they are enriched with H3K4me3 and H3K27ac but not with H3K27me3 or H2AK119ub. Previously, it was shown that AUTS2-binding loci were predominantly localized in active chromatin in experiments using P1 whole-brain and mouse embryonic stem cell (mESC)-derived motor neurons (Gao et al, 2014; Liu et al, 2021). Notably, our experiments showed that AUTS2-binding loci were present on active chromatin (cluster 2) and repressive chromatin (cluster 1), at least in IPCs (Fig. 6E). In this experiment using control IPCs, we identified 12479 and 10616 significant loci that are histone modified in H3K27me3 and H3K27ac, respectively. There was considerable overlap between both types of loci and the AUTS2-binding loci, suggesting that AUTS2 binds to both active and repressive chromatin (Fig. 6F).

Next, we focused on the AUTS2-binding loci within 3 kb of the TSS of IPC-upregulated genes (61 loci) because more than half of the AUTS2-binding loci were localized within 3 kb of TSSs (Fig. 6B). As a control, we also identified 2717 AUTS2-binding loci within 3 kb of the TSS of genes whose expression did not change in the cKO IPCs ($P$ value >0.1). We named these loci as "IPC-upregulated-TSSs" and "IPC-unchanged-TSSs" (Fig. 7A). We compared histone modifications between IPC-upregulated-TSSs and IPC-unchanged-TSSs in the control genetic background (*Eomes*$^{T2A-d2EGFP/+}$; *Auts2*$^{fl(ex8)/fl(ex8)}$). IPC-upregulated-TSSs were enriched with repressive histone marks (H3K27me3 and H2AK119ub), whereas they harbored poorly active histone marks (H3K4me3 and H3K27ac) compared with IPC-unchanged-TSSs (Fig. 7A; Appendix Fig. S8A). This suggests that the chromatin of IPC-upregulated-TSSs might be relatively in the repressive condition. In addition, GO analysis in biological processes showed that the genes with IPC-upregulated-TSSs were associated with the GO terms, including "neuron differentiation," "generation of neurons," "neurogenesis," and "central nervous system (CNS) neuron differentiation" in contrast to GO terms for the genes with IPC-unchanged-TSSs (Fig. 7B; Appendix Fig. S8B). These findings suggest that AUTS2 suppresses the expression of genes associated with neuron production in IPCs.

We further conducted the CUT&Tag experiments on sorted EGFP+ cells from *Auts2* cKO mice (*Eomes*$^{T2A-d2EGFP/+}$; *Emx1*$^{Cre/+}$; *Auts2*$^{fl(ex8)/fl(ex8)}$) at E15.5, using antibodies for histone modifications and RING1B to test whether histone modifications and RING1B-binding were affected

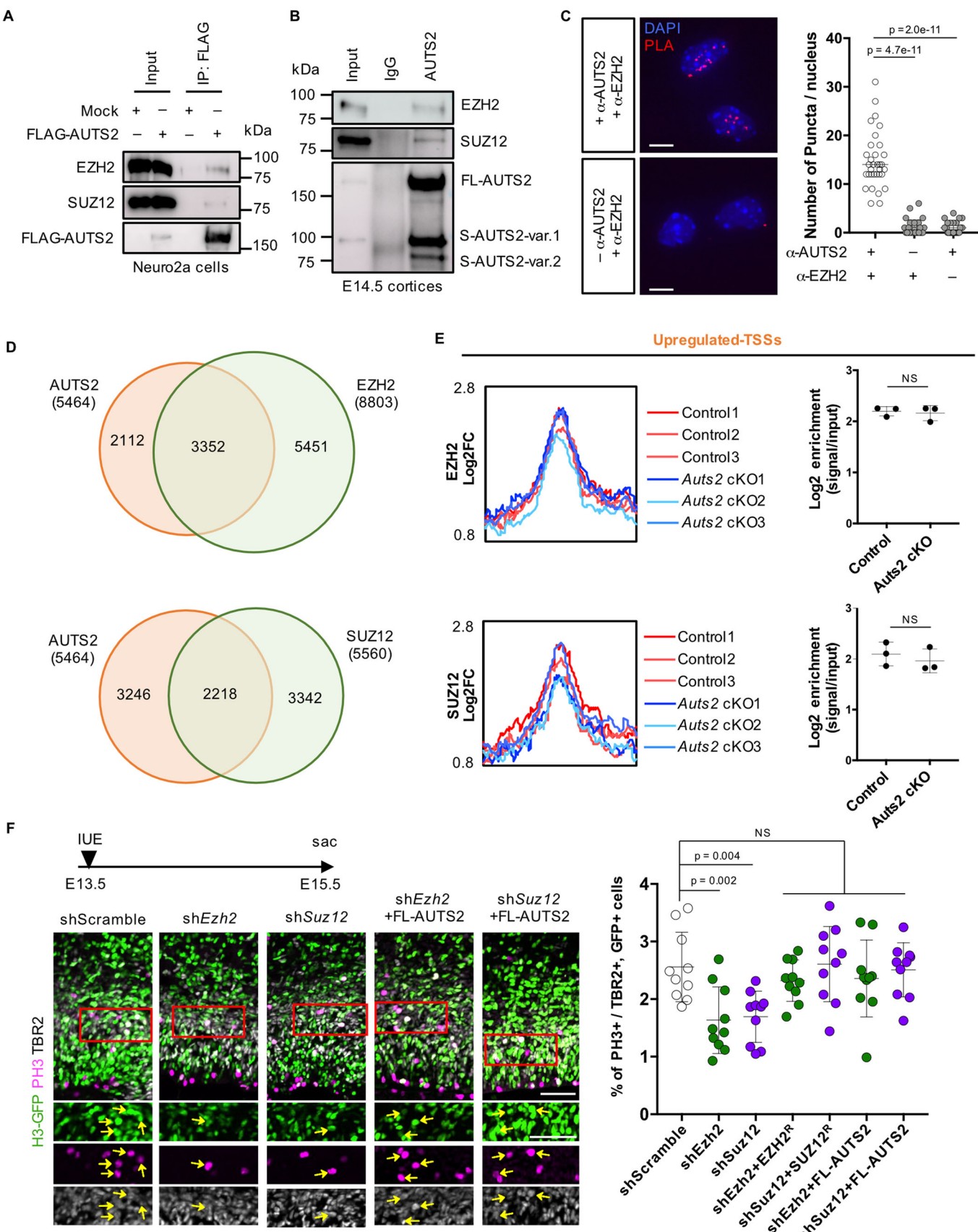

**Figure 8.  AUTS2 interacts with PRC2 in IPCs.**

(A) Co-immunoprecipitation assay using Neuro2a cells. 3xFLAG-tagged FL-AUTS2 (FLAG-AUTS2) expression vector or 3xFlag-tagged empty vector (Mock) was transfected into Neuro2a cells. The nuclear lysates were incubated with an anti-FLAG antibody. Western blotting was performed with anti-EZH, anti-SUZ12, and anti-FLAG antibodies. (B) Co-immunoprecipitation assay using E14.5 cerebral cortices. Rabbit anti-AUTS2 and rabbit control IgG antibodies were used for immunoprecipitation. The precipitates were immunoblotted with anti-EZH2, anti-SUZ12, and anti-AUTS2 antibodies. (C) Proximity ligation assay (PLA) with rabbit anti-AUTS2 and mouse anti-EZH2 antibodies in sorted EGFP$^+$ cells from E15.5 *Eomes*$^{T2A-d2EGFP/+}$ cortices. Representative images show the PLA signals (red) and DAPI (blue) in cells incubated with both antibodies (top) or a single antibody (only anti-EZH2, bottom). The graph shows the number of puncta in the nucleus of each sample. $N = 3$ biological replicates, 28–35 cells. (D) Venn diagrams show the extent of overlap between AUTS2 peaks and EZH2 (top) or SUZ12 (bottom) peaks. (E) Density profiles show the occupancy of EZH2 (top) and SUZ12 (bottom) in control and *Auts2* cKO cells centered on IPC-upregulated-TSSs (±3 kb). Graphs show the enrichment of EZH2 and SUZ12 on IPC-upregulated-TSSs. $N = 3$ biological replicates. (F) IUE of indicated vectors and an H3-GFP expression vector were performed into WT cortices at E13.5. The cortical sections are stained for GFP (green), PH3 (magenta), and anti-TBR2 (white) antibodies at E15.5. The graph shows the ratio of PH3$^+$ cells in TBR2$^+$ and GFP$^+$ cells. Red rectangles indicate the location of the images below. Arrows indicate GFP$^+$, PH3$^+$, and TBR2$^+$ cells. $N = 5$ mice, 10 sections. Data were presented as the median (C) and the mean ± SD (E, F). NS not significant, Kruskal–Wallis test (C), unpaired Student's *t*-test (E), and one-way ANOVA with Dunnett's post hoc test (F). Scale bars, 5 µm (C), 50 µm (F). Source data are available online for this figure.

without the AUTS2 protein. The levels of RING1B binding to IPC-upregulated-TSSs and IPC-unchanged-TSSs did not significantly change in cKO cells compared with control cells (Appendix Fig. S8C), suggesting that AUTS2 is not involved in the localization of the PRC1 complex on the chromatin in IPCs. H2AK119ub modification was not affected in either IPC-upregulated- or unchanged-TSSs (Appendix Fig. S8D), indicating that AUTS2 may not play a major role in chromatin ubiquitination, at least in IPCs. Regarding IPC-upregulated-TSSs, the levels of the repressive modification (H3K27me3) were reduced in cKO cells, whereas those of the active modifications (H3K4me3 and H3K27ac) were elevated (Fig. 7C; Appendix Fig. S8E). However, these tendencies were not observed for IPC-unchanged-TSSs (Appendix Fig. S8F). At the TSS of the *Robo1* gene, a representative IPC-upregulated-gene, the repressive mark (H3K27me3) was reduced, and active marks were slightly but significantly increased in the mutant cells compared with the control (Fig. 7D,E). Similarly, a significant reduction of the H3K27me3 modification was observed in other representative genes with IPC-upregulated-TSSs, associated with the GO terms "neuron differentiation," "generation of neuron," and/or "cell differentiation" (Fig. 7E). These findings suggest that AUTS2 maintains the repressive states of genes associated with neuron production in IPCs.

## AUTS2 interacts with PRC2 to regulate IPC division

H3K27me3 modification is catalyzed by PRC2 (Cao et al, 2002; Muller et al, 2002; Kuzmichev et al, 2002). We performed co-immunoprecipitation experiments to investigate whether AUTS2 interacts with PRC2. First, to efficiently immunoprecipitate sufficient amounts of exogenous AUTS2 protein, 3xFlag-tagged FL-AUTS2 (FLAG-AUTS2) was transfected into Neuro2a cells and immunoprecipitated using an anti-FLAG antibody two days after transfection. Endogenous EZH2 and SUZ12, the core proteins of PRC2, were found to be co-immunoprecipitated only in FLAG-AUTS2 expressing cells (Fig. 8A). Next, we immunoprecipitated endogenous AUTS2 proteins with an anti-AUTS2 antibody from E14.5 cerebral cortices. Western blotting showed that EZH2 and SUZ12 were co-immunoprecipitated with AUTS2 (Fig. 8B), suggesting that AUTS2 interacts with PRC2 in the developing cerebral cortex. Furthermore, we performed a proximity ligation assay (PLA) using anti-AUTS2 and anti-EZH2 antibodies in the sorted EGFP$^+$ cells from the *Eomes*$^{T2A-d2EGFP/+}$ cerebral cortices at E15.5. Notably, many puncta were observed in the cell nuclei with

both antibodies compared to those with a single antibody (Fig. 8C), suggesting that AUTS2 interacts with PRC2 in the IPC nuclei.

We performed CUT&Tag experiments for EZH2 and SUZ12 on sorted EGFP$^+$ cells from control cortices at E15.5 to examine the correlation of PRC2-binding loci with AUTS2 ones. Global distributions of EZH2 and SUZ12 were observed at AUTS2-binding loci (Appendix Fig. S9A), and 3352 and 2218 AUTS2 peaks overlapped with EZH2 and SUZ12 peaks, respectively (Fig. 8D). These findings suggest that AUTS2 co-localizes with PRC2 in the genome of IPCs. We further conducted CUT&Tag experiments in the sorted EGFP$^+$ cells from *Auts2* cKO (*Eomes*$^{T2A-d2EGFP/+}$; *Emx1*$^{Cre/+}$; *Auts2*$^{fl(ex8)/fl(ex8)}$) cortices at E15.5. Then, we compared the levels of EZH2- and SUZ12-binding to IPC-upregulated-TSSs between control and *Auts2* cKO cells to investigate whether PRC2 localization was dependent on AUTS2. We did not detect differences in the binding levels to IPC-upregulated-TSSs or IPC-unchanged-TSSs (Fig. 8E; Appendix Fig. S9B). Consistently, EZH2- and SUZ12-binding levels were not affected around the TSS of *Robo1* (Appendix Fig. S9C). These findings suggest that AUTS2 is not involved in the localization of PRC2 on chromatins.

We performed KD experiments using shRNAs for *Ezh2* and *Suz12* to test the involvement of PRC2 in IPC division in vivo. First, we confirmed that the shRNA vectors sufficiently reduced the exogenous expression of EZH2 and SUZ12 in HEK293T cells (Appendix Fig. S9D). These shRNA vectors plus the H3-GFP vector were co-electroporated into the VZ of the WT cortices at E13.5, and the sections were immunostained with PH3, TBR2, and GFP at E15.5. The ratio of PH3$^+$ cells in TBR2$^+$, GFP$^+$ cells was significantly reduced by sh*Ezh2* or sh*Suz12*, which was rescued by the co-introduction of shRNA-resistant EZH2 or SUZ12 (EZH2$^R$ and SUZ12$^R$) (Fig. 8F). This suggests that PRC2 is required for the division of IPCs in the embryonic cortices. Interestingly, the co-introduction of FL-AUTS2 with *Ezh2* or *Suz12* shRNA vector restored the ratio of PH3$^+$ cells (Fig. 8F), implying that AUTS2 and PRC2 may work together in the same pathway for IPC division.

To further investigate the role of PRC2 specifically in IPCs, we tried to overexpress the dominant negative (DN) mutant of EZH2 (EZH2$^{DN}$, Y728* nonsense mutation described in Deevy et al) in TBR2$^+$ cells (preprint: Deevy et al, 2024). We co-electroporated pCAG-LSL-*Ezh2*$^{DN}$ and pCAG-H3GFP vectors into *Eomes*$^{T2A-iCre/+}$ KI embryos at E13.5 and examined the percentage of PH3$^+$ cells in TBR2$^+$, GFP$^+$ cells at two days after electroporation (Appendix Fig. S9E). Compared to the control, we found that the ratio of PH3$^+$

cells decreased in cells that had been introduced with pCAG-LSL-Ezh2$^{DN}$. However, such effects were restored by co-electroporation of wild-type Ezh2 (pCAG-LSL-Ezh2$^{WT}$) (Appendix Fig. S9E). This suggests that EZH2, or PRC2, functions in IPCs and is involved in IPC division.

## Discussion

Previously, AUTS2 syndrome has been reported to present with a high frequency of microcephaly in addition to psychiatric disorders such as ID and ASD (Sanchez-Jimeno et al, 2021; Beunders et al, 2016; Beunders et al, 2013). Since microcephaly-like phenotypes have not been reproduced in in vivo models of animals with neocortex, the mechanism by which microcephaly is caused by loss of AUTS2 remains unclear. In this study, we demonstrated that Auts2 cKO mice exhibit a reduced number of cortical upper-layer neurons and reduced cortical thickness (Fig. 1C–E). Notably, this phenotype was observed in Auts2 heterozygous cKO mice, which may mimic the pathology of human AUTS2 syndrome patients. In AUTS2 syndrome, different genomic mutations lead to various clinical features and symptoms, but not all symptoms are present in individual patients (Oksenberg and Ahituv, 2013; Sanchez-Jimeno et al, 2021). Approximately 65% of patients with AUTS2 syndrome exhibit microcephaly (Sanchez-Jimeno et al, 2021). Symptomatic diversity is believed to be caused by different mutations at various sites in the very long AUTS2 gene. Correspondingly, a recent study reported that Auts2 KO mice with an allele that differs from Auts2$^{del(ex6)}$ did not show any change in the number of neurons in the cortex (Li et al, 2022). So far, several mouse lines with different mutations in the AUTS2 gene have been reported (Gao et al, 2014; Hori et al, 2014; Castanza et al, 2021; Li et al, 2022). These mouse models may serve as models for AUTS2 syndrome with various genomic variants.

In addition to investigating disease animal models, organoids made from human induced pluripotent stem cells (iPSCs) are also powerful tools for understanding the pathology of diseases. A previous study demonstrated that a patient with the AUTS2$^{T534P}$ missense mutation displayed reduced cortical area and ventriculomegaly on MRI. In the organoids generated from the patient (AUTS2$^{T534P}$)-derived iPSCs, the division of neural progenitor cells, presumably apical RGCs, was reduced compared to controls during the early neurogenic phase, resulting in impaired organoid growth (Fair et al, 2023). This suggests that abnormalities in RGCs may also be involved in microcephaly in AUTS2 syndrome. Although we could not detect abnormalities in RGC division in our mouse model, a decrease in the thickness of the deep layer was detected in the rostral cortex of those mice (Fig. 1E), implying some functional abnormality in the RGCs. This might be supported by the result that hundreds of DEGs in RGCs were identified between control and the Auts2 cKO (Emx1$^{Cre/+}$; Auts2$^{fl(ex8)/fl(ex8)}$) mice (Appendix Fig. S5B). As mentioned above, there are various mutation types in AUTS2 syndrome, and each mutation type may cause different molecular functional abnormalities and phenotypes. The AUTS2$^{T534P}$ used in the organoid research was a missense mutation in the HX repeat motif (Histone acetyltransferase P300-binding site) of the AUTS2 protein. This AUTS2 mutant protein cannot bind to P300, which may alter the transcription of genes. Since functional changes in the AUTS2$^{T534P}$ protein differ from those in

our mouse model (Emx1$^{Cre/+}$; Auts2$^{fl(ex8)/fl(ex8)}$), in which expression of FL-AUTS2 and S-AUTS2-var.1 is lost and that of S-AUTS2-var.2 is increased (Appendix Fig. S1B), the microcephaly phenotype may be expressed through slightly different mechanisms in each pathological model.

Microcephaly is often caused by a reduction in the number of cortical neurons. During evolution, mammals are thought to have acquired IPCs in addition to RGCs, thereby leading to an increase in the number of neurons and overall size of the cerebral cortex (Martinez-Cerdeno et al, 2006). Moreover, during the evolution of primates, including humans, the acquisition of outer radial glial cells (oRGCs) has led to a further increase in the number of neurons and expansion of the cerebral cortex (Hansen et al, 2010; Fietz et al, 2010). Unfortunately, because oRGCs exist in very limited numbers in mice (Lui et al, 2011), analyses using mouse models allow us to study the effects of the Auts2 mutations on RGCs and IPCs, but not to observe the effects on oRGCs. Nevertheless, our Auts2 cKO mice (Emx1$^{Cre/+}$; Auts2$^{fl(ex8)/fl(ex8)}$ or Auts2$^{fl(ex8)/+}$) may recapitulate at least some of the pathophysiology of microcephaly in human AUTS2 syndrome, since we observed a decrease in dividing IPCs and a decrease in the number of upper-layer neurons and thickness of the cerebral cortex.

Previous clonal analyses have shown that mouse IPCs divide once to multiple times, generating approximately 2–12 neurons (Mihalas and Hevner, 2018; Noctor et al, 2004). However, the regulatory mechanisms of their proliferation and differentiation are still largely unexplored compared to RGCs. We found that the protein ROBO1 suppresses IPC division (Fig. 5A; Appendix Fig. S6H). AUTS2, in turn, suppresses Robo1 expression in IPCs, leading to an increase in IPC division and the production of upper-layer neurons. Notably, the division of TBR2$^+$ cells is absent or rare in reptiles or birds but substantially present in mammals, with increased IPC division in primates (Nomura et al, 2013; Nomura et al, 2016; Betizeau et al, 2013; Lewitus et al, 2014; Stepien et al, 2021). Therefore, this study provides insights not only into the regulatory mechanisms of IPC division and neuronal differentiation but also into the expansion of the mammalian brain during evolution. From the pathological point of view, we show that loss of Auts2 expression increases Robo1 expression in IPCs, elongates the cell cycle and G1-phase of IPCs, suppresses IPC division, and decreases the number of upper-layer neurons. We further showed the involvement of the SLIT1-ROBO1 pathway in this process. This may explain, at least in part, the pathomechanisms of microcephaly of AUTS2 syndrome. The contribution of oRGCs to microcephaly in AUTS2 syndrome may have to wait for future studies, which may include long-term organoid culturing using primate cells or the establishment of primate animal models.

Previous studies have revealed that nuclear AUTS2 can function as a transcriptional activator or repressor, depending on the type of cell (Monderer-Rothkoff et al, 2021; Gao et al, 2014; Liu et al, 2021; Li et al, 2022). Using a luciferase reporter with a UAS sequence, Monderer-Rothkoff et al showed that GAL4-AUTS2 expression increased luciferase activity in HEK293 cells but decreased in Neuro2a cells (Monderer-Rothkoff et al, 2021). This indicates that AUTS2 might have a role in either activating or repressing transcription, depending on the specific cell type. Gao et al showed that, in HEK293 cells, AUTS2 interacts with non-canonical PRC1 and contributes to transcriptional activation by recruiting P300 and engaging in acetylation of H3K27 (Gao et al, 2014). In this study,

we show that the AUTS2 protein is primarily involved in transcriptional repression in IPCs. In the *Auts2* cKO IPCs, we found many upregulated and downregulated genes, but among them, the AUTS2 protein binds mainly to the genomic regions around the TSSs of the upregulated genes (Fig. 6B,C). Around the TSSs of those genes, histone modification of H3K27me3 was significantly reduced in the *Auts2* cKO IPCs (Fig. 7C). We further found that AUTS2 interacts with EZH2 and SUZ12, the core components of PRC2 that catalyze the trimethylation of H3K27 (Fig. 8A–C) (Cao et al, 2002; Muller et al, 2002; Kuzmichev et al, 2002). In both CUT&Tag experiment and PLA, we showed that AUTS2 and PRC2 components (EZH2 and SUZ12) bind to common genomic regions in IPCs (Fig. 8C,D; Appendix Fig. S9A). Electroporation of a shRNA vector for *Ezh2* or *Suz12* into the embryonic VZ decreased dividing IPCs, which was rescued by overexpression of AUTS2 (Fig. 8F). These findings suggest that PRC2 and AUTS2 are cooperatively involved in the regulation of IPC division by trimethylating H3K27 and repressing gene expression. Regarding the *Robo1* gene, we found that PRC2 binds to the region around the TSS in the control IPCs (Appendix Fig. S9C). However, that region showed reduced levels of H3K27me3 in the cKO condition (Fig. 7D). Consistently, previous studies have reported that *Robo1* expression is elevated in the cerebral cortex of *Ezh2* KO embryos (Pereira et al, 2010). Since the binding sites of PRC2 components to the genome in IPCs were not altered in *Auts2* cKO (Fig. 8E), it seems unlikely that AUTS2 is involved in the genomic positioning of PRC2. AUTS2 might be involved in the H3K27 trimethylation activity of PRC2, but unfortunately, we have no direct evidence. We need to conduct additional analysis to clarify this issue.

There have been reports on the role of PRC2 in cortical development. In the KO mice for essential components of the PRC2 complex, *Ezh2* and *Eed*, the balance between self-renewal and differentiation in NPCs shifted towards differentiation, resulting in the reduced number of neuron production and microcephaly-like phenotype (Pereira et al, 2010; Telley et al, 2019). Many scientists believe these phenotypes were caused by the loss of function of PRC2 in RGCs, albeit lacking direct evidence. In this study, we specifically disrupted the function of PRC2 in IPCs by specific induction of EZH2 dominant negative form (Appendix Fig. S9E), suggesting that PRC2 function in IPCs is involved in IPC division and appropriate production of upper-layer neurons. As PRC2 complexes are expressed in both RGCs and IPCs, PRC2 may play important roles in each population during corticogenesis. Previously, a comprehensive genomic analysis was conducted to investigate the H3K27me3 modification status in neuroepithelial cells at E9.5 and apical RGCs, oRGCs, and neurons at E14.5 (Albert et al, 2017). The researchers reported that the genes modified by H3K27me3 changed dynamically in each cell type, suggesting that the target genes of PRC2 change depending on the cell type and developmental stage.

In this study, we generated a mouse model for AUTS2 syndrome that exhibits a microcephaly-like phenotype. We analyze model mice and present pathomechanisms for microcephaly in AUTS2 syndrome and the molecular machinery to regulate IPC division and differentiation. We further suggest the new function of AUTS2 to suppress gene expression via H3K27 trimethylation. This study leads to a better understanding of the pathogenesis of AUTS2 syndrome and the regulatory mechanisms of gene

expression and also provides insight into the evolution of the mammalian brain.

# Methods

### Reagents and tools table

| Reagent/resource | Reference or source | Identifier or catalog number |
|---|---|---|
| **Experimental models** | | |
| *Auts2*<sup>del(ex8)</sup> mice | Hori et al, 2014 | N/A |
| *Auts2*<sup>fl(ex8)</sup> mice | Hori et al, 2014 | N/A |
| Slc:ICR mice | SLC, Shizuoka, Japan | N/A |
| *Eomes*<sup>T2A-d2EGFP</sup> mice | This study | N/A |
| *Eomes*<sup>T2A-iCre</sup> mice | This study | N/A |
| **Recombinant DNA** | | |
| pCAG-Myc-FL-Auts2 | Hori et al, 2014 | N/A |
| pCAG-Myc-FL-Auts2<sup>R</sup> | Hori et al, 2014 | N/A |
| pCAG-Myc-NLS-FL-Auts2<sup>R</sup> | Hori et al, 2020 | N/A |
| pCAG-Myc-NES-FL-Auts2<sup>R</sup> | Hori et al, 2014 | N/A |
| pCAG-Myc-S-Auts2-var.1 | Hori et al, 2014 | N/A |
| pCAG-Myc-S-Auts2-var.2 | Hori et al, 2014 | N/A |
| pmU6-shScramble | Hori et al, 2014 | N/A |
| pmU6-shAuts2 | Hori et al, 2014 | N/A |
| pCAG-Robo1 | This study | N/A |
| pCAG-Slit1 | This study | N/A |
| pCAG-Ezh2<sup>R</sup> | This study | N/A |
| pCAG-Suz12<sup>R</sup> | This study | N/A |
| pmU6-Robo1#1 | This study | N/A |
| pmU6-Robo1#2 | This study | N/A |
| pmU6-Ezh2 | This study | N/A |
| pmU6-Suz12 | This study | N/A |
| pCAG-PA-Ezh2 | This study | N/A |
| pCAG-PA-Suz12 | This study | N/A |
| pCAG-LSL-Robo1 | This study | N/A |
| pCAG-LSL-Ezh2<sup>DN</sup> | This study | N/A |
| pCAG-LSL-Ezh2<sup>WT</sup> | This study | N/A |
| **Antibodies** | | |
| Rabbit anti-CUX1 1:1000 for immunofluorescence | Santa Cruz Biotechnology | sc-130424 |
| Rat anti-CTIP2 1:3000 for immunofluorescence | Abcam | ab18465 |
| Rabbit anti-PAX6 1:500 for immunofluorescence | BioLegend | 901301 |
| Rabbit anti-TBR2 1:1000 for immunofluorescence 1:1000 for immunoblotting | Abcam | ab183991 |

| Reagent/resource | Reference or source | Identifier or catalog number |
|---|---|---|
| Rat anti-TBR2 1:500 for immunofluorescence | Invitrogen | 14-4875-82 |
| Rabbit anti-cleaved caspase-3 1:400 for immunofluorescence | Cell Signaling Technology | 9661S |
| Rat anti-PH3 (pSer28) 1:400 for immunofluorescence | Sigma-Aldrich | H9908 |
| Rat anti-KI67 1:500 for immunofluorescence | Invitrogen | 14-5698-82 |
| Sheep anti-BrdU 1:200 for immunofluorescence | Abcam | ab1893 |
| Chicken anti-GFP 1:1000 for immunofluorescence | Abcam | ab13970 |
| Mouse anti-HuC/HuD 1:1000 for immunofluorescence | Invitrogen | A-21271 |
| Goat anti-SOX2 1:500 for immunofluorescence | R&D | AF2018 |
| Goat anti-ROBO1 1:200 for immunofluorescence | R&D | AF1749 |
| Rat anti-RFP 1:1000 for immunofluorescence | Proteintech | 5F8 |
| Rabbit anti-ROBO1 1:2000 for immunoblotting | Proteintech | 20219-1-AP |
| Mouse anti-β-actin 1:500 for immunoblotting | MBL | M177-3 |
| Rat anti-PA 1:1000 for immunoblotting | Fuji Film | 016-25861 |
| Mouse anti-FLAG | Sigma-Aldrich | F1804 |
| Rabbit anti-FLAG 1:500 for immunoblotting | Sigma-Aldrich | F7425 |
| Rabbit anti-EZH2 1:1000 for immunoblotting | Cell Signaling Technology | 5246S |
| Rabbit anti-SUZ12 1:1000 for immunoblotting | Cell Signaling Technology | 3737S |
| Rabbit anti-AUTS2 1:500 for immunoblotting 1:400 for PLA | Sigma-Aldrich | HPA000390 |
| Mouse anti-EZH2 1:100 for PLA | Cell Signaling Technology | 3147S |
| Rabbit anti-RING1B | Cell Signaling Technology | 5694S |
| Rabbit anti-H3K27me3 | Cell Signaling Technology | 9733S |
| Rabbit anti-H3K4me3 | Cell Signaling Technology | 9751S |
| Rabbit anti-H3K27ac | Cell Signaling Technology | 8173S |
| Rabbit anti-H2AK119ub | Cell Signaling Technology | 8240S |
| Rat anti-CD133 APC 1:400 for cell sorting | BioLegend | 141207 |

| Reagent/resource | Reference or source | Identifier or catalog number |
|---|---|---|
| Donkey anti-rabbit alexa488 1:1000 for immunofluorescence | Abcam | ab150065 |
| Donkey anti-rabbit alexa568 1:1000 for immunofluorescence | Abcam | ab175692 |
| Donkey anti-rabbit alexa647 1:1000 for immunofluorescence | Abcam | ab150067 |
| Donkey anti-rat alexa568 1:1000 for immunofluorescence | Abcam | ab175475 |
| Donkey anti-rat alexa647 1:1000 for immunofluorescence | Abcam | ab150155 |
| Donkey anti-goat alexa647 1:1000 for immunofluorescence | Abcam | ab150135 |
| Donkey anti-Sheep alexa594 1:1000 for immunofluorescence | Thermo Fisher Scientific | A11016 |
| Donkey anti-chicken alexa488 1:1000 for immunofluorescence | Jackson ImmunoResearch | 703-545-155 |
| goat anti-rabbit HRP 1:5000 for immunoblotting | Jackson ImmunoResearch | 111-035-144 |
| goat anti-mouse HRP 1:5000 for immunoblotting | Jackson ImmunoResearch | 115-035-003 |
| Goat anti-rat HRP 1:5000 for immunoblotting | Jackson ImmunoResearch | 112-035-143 |
| Rabbit IgG | Jackson ImmunoResearch | 011-000-003 |
| Goat anti-Rabbit IgG | EpiCypher | 13-0047 |
| **Oligonucleotides and other sequence-based reagents** | | |
| qPCR primers | This study | Table EV1 |
| **Chemicals, Enzymes and other reagents** | | |
| Click-iT™ EdU Cell Proliferation Kit for Imaging | Invitrogen | C10340 |
| Neuron Dissociation Solutions | Fujifilm Wako | 291-78001 |
| ECL Prime Western Blotting Detection Reagent | GE Healthcare | RPN2236 |
| RNeasy Micro Kit | QIAGEN | 74004 |
| ReverTra Ace qPCR RT kit | Toyobo | FSQ-101 |
| PowerUp SYBR Green Master Mix | Thermo Fisher Scientific | A25742 |
| Dynabeads™ protein G | Invitrogen | 10004D |
| Poly-D-lysine | Sigma-Aldrich | P6407 |
| Duolink in situ detection reagent Red | Sigma-Aldrich | DUO92008 |
| Duolink In Situ PLA Probe anti-Rabbit PLUS | Sigma-Aldrich | DUO92002 |

| Reagent/resource | Reference or source | Identifier or catalog number |
|---|---|---|
| Duolink In Situ PLA Probe anti-Mouse MINUS | Sigma-Aldrich | DUO92004 |
| NEBNext ultra II directional RNA library prep kit | NEB | E7760 |
| pAG-Tn5 | EpiCypher | 15-1117 |
| PrimeSTAR MAX | Takara | R045A |
| NEBNext HiFi 2x Master Mix | NEB | M0541 |
| DNA cleaner & concentrator-5 | ZYMO Research | D4014 |
| **Software** | | |
| Fastp | Chen et al, 2018 | |
| HISAT2 | Kim et al, 2015 | |
| StringTie | Pertea et al, 2016 | |
| DESeq2 | Love et al, 2014 | |
| DAVID | https://david.ncifcrf.gov/summary.jsp | |
| Bowtie2 | Langmead and Salzberg, 2012 | |
| SAMtools | Li et al, 2009 | |
| MACS2 | Zhang et al, 2008 | |
| deepTools | Ramirez et al, 2016 | |
| ChIPseeker | Yu et al, 2015 | |
| GREAT | http://great.stanford.edu/public/html/index.php | |
| IGV | Robinson et al, 2011 | |
| Prism 7 | GraphPad | |
| ImageJ | NIH | |
| **Other** | | |
| scRNA-seq | Yuzwa et al, 2017 | GSE107122 |
| scRNA-seq | Di Bella et al, 2021 | GSE153164 |
| RNA-seq | This study | PRJDB17034 |
| CUT&Tag | This study | PRJDB17038 |

## Methods and protocols

### Animals

Generation of $Auts2^{del(ex8)}$ and $Emx1^{Cre/+}$; $Auts2^{fl(ex8)}$ alleles have been described previously (Hori et al, 2014; Hori et al, 2020). $Auts2^{del(ex8)/del(ex8)}$ homozygotes are lethal at birth; therefore, the $Auts2^{del(ex8)}$ allele was maintained in heterozygosity in the C57BL6/N background. $Auts2^{fl(ex8)/fl(ex8)}$ and $Emx1^{Cre/+}$; $Auts2^{fl(ex8)/fl(ex8)}$ mice were maintained on a mixed Balb/c and C57BL/6 N background. Littermates of experimental mutants were used as controls. Pregnant Institute of Cancer Research (ICR) mice were purchased from Japan SLC Inc. (Shizuoka, Japan) and used for in utero electroporation. The day of the vaginal plug was considered embryonic day (E) 0.5. The mice were reared in a temperature-controlled, pathogen-free facility under a 12-h light/dark cycle with free access to food and water in ventilated racks. The Animal Care

and Use Committee of the National Center for Neurology and Psychiatry (2022006R2) approved all the animal experiments.

## Plasmids construction

The construction of pCAG-Myc-$Auts2$-full length (FL-$Auts2$), FL-$Auts2^R$ (shRNA-resistant FL-$Auts2$), NLS-FL-$Auts2^R$, NES-FL-$Auts2^R$, S-$Auts2$-var.1, S-$Auts2$-var.2 expression plasmids, pmU6-$Auts2$, and scrambled shRNA vectors have been described previously (Hori et al, 2014; Hori et al, 2020). A cDNA fragment encoding full-length $Robo1$, $Ezh2$, and $Suz12$ were amplified using a cDNA library from the E14.5 mouse brain. The fragment was inserted into a pCAGGS vector (GE Healthcare, Chicago, IL). shRNA vectors were generated by inserting double-stranded oligonucleotides into the mU6 pro vector. Target sequences are as follows; sh$Robo1$ #1 (5′-GGATGATAAAGAT-GAAAGAAT-3′), sh$Robo1$ #2 (5′-GGAAGTTACTGATGT-GATTGC-3′), sh$Ezh2$ (5′-GAAGTAAAGACTATGTTTAGT-3′), sh$Suz12$ (5′-GAATTTAATGGAATGATTAAT-3′). Targeting sequences were designed using siDirect 2.0 (Naito et al, 2009). The shRNA-resistant $Ezh2$ and $Suz12$ expression vectors were constructed by introducing six silent mutations into cDNA sequence by PCR-mediated site-directed mutagenesis using the following primer set: sh$Ezh2$-resistant forward (5′-GGTTAAAACAATGTTGAGCTC-CAATCGTCAGAAAATT-3′), sh$Ezh2$-resistant reverse (5′-GCTCAACATTGTTTTAACCTCATCAGCTCTTCTGAAC-3′), sh$Suz12$-resistant forward (5′-GTTCAACGGTATGATAAACGGA-GAAACCAATGAAAAT-3′), sh$Suz12$-resistant reverse (5′-GTTTATCATACCGTTGAACTCTCTTCTTCCTGGACGA-3′) (Underlines indicate the mismatched nucleotide with shRNA). Dominant negative of $Ezh2$ (Y728*; $Ezh2^{DN}$) expression vector was constructed by PCR-mediated mutagenesis using the following primer set: forward (5′-CAGATAAAGCCAGGCTGATGCCCTG-3′), reverse (5′-CTGGCTTTATCTGTAATCAAAAAAC -3′). pCAG-$loxP$-polyA-$loxP$ (LSL)-$Robo1$, $Ezh2^{DN}$ or $Ezh2^{WT}$ vector was constructed by inserting $Robo1$, $Ezh2^{DN}$ or $Ezh2^{WT}$ into pCAG-$loxP$-polyA-$loxP$-mKO2-F (provided by A. Shitamukai from RIKEN CDB) using EcoRI/NotI (for $Robo1$) or BamHI/EcoRV (for $Ezh2^{DN}$, $Ezh2^{WT}$). pCAG-$Slit1$ expression vector was designed and generated using services offered by VectorBuilder (CA, USA). The pCAG-H3EGFP vector was a gift from N. Masuyama. Plasmids were purified using an Endo-Free Plasmid Purification Kit (QIAGEN, Hilden, Germany).

## Immunofluorescence

Postnatal brains were dissected after perfusion fixation with 4% paraformaldehyde (PFA) in 0.1 M sodium phosphate buffer (pH 7.2) under deep isoflurane anesthesia. The brains were further fixed in 4% PFA for 2 h at room temperature (RT; 25 ± 2 °C). Embryonic brains were fixed in 4% PFA for 3–4 h at 4 °C. Fixed brains were cryoprotected by immersion in 30% sucrose in PBS, embedded in optimum cutting temperature (Tissue-Tek OCT compound, Sakura Fine-Tek, Tokyo, Japan) and cryosectioned at 14–16 μm (CM3050 S; Leica, Wetzlar, Germany). The points of the sections observed were set as follows: "rostral" is the anterior side where the corpus callosum is located, "central" is the anterior side where the hippocampus is located, and "caudal" is the posterior side where the hippocampus is located. Cells were fixed with 4% PFA for 20 min for immunocy-tochemistry at RT. Sections or cells were pre-incubated in blocking buffer containing 2–10% normal donkey serum (Merck Millipore,

Burlington, MA) and 0.1% Triton X-100 at RT for 1 h and incubated overnight at 4 °C with primary antibodies and subsequently incubated with secondary antibodies and DAPI (5 µg/µl; Invitrogen, Carlsbad, CA) in blocking buffer for 1–2 h at RT. For BrdU staining, before incubation with the blocking solution, the sections were incubated with 2 N HCl at 37 °C for 30 min. Fluorescent images were acquired using a laser scanning confocal microscope FV1000 (Olympus, Tokyo, Japan), SpinSR10 (Olympus), and a Zeiss LSM 780 confocal microscope system (Carl Zeiss, Oberkochen, Germany). The number of cells and intensity were measured using "Cell Counter" and "Measure" in ImageJ software, respectively.

## Nissl staining

Sections were stained with 0.1% cresyl violet in 1% acetic acid for 5 min, dehydrated using ethanol series and xylene, and mounted in Entellan. Stained sections were observed under a Keyence All-in-One microscope (BZ-X700, Osaka, Japan).

## EdU staining

Pregnant mice were intraperitoneally injected with 5-ethynyl-2′-deoxyuridine (EdU) (30 mg/kg). Sections were stained using the Click-iT EdU imaging kit according to the manufacturer's instructions (Thermo Fisher Scientific, Waltham, MA, USA).

## In utero electroporation

In utero electroporation experiments were performed as described previously with some modifications (Kawauchi et al, 2003). WT ICR, *Auts2*$^{fl(ex8)/fl(ex8)}$, homozygous *Auts2* cKO, or *Eomes*$^{T2A-iCre/+}$ mice were anesthetized with isoflurane. pmU6-*Auts2*, *Robo1*, *Ezh2*, *Suz12*, or scramble shRNA vector, pCAG-*Robo1* or *Slit1* expression vector, and pCAG-H3GFP vector were diluted to 1.5, 2, and 0.5 µg/µl, respectively. For rescue experiments in Fig. 3, indicated expression vectors (0.5 µg/µl) were co-electroporated with pmU6-*Auts2* shRNA vector and pCAG-H3GFP vector. About 1.5 µg/µl pCAG-*Ezh2*$^R$ or *Suz12*$^R$ vector was co-electroporated with pmU6-*Ezh2* or *Suz12* shRNA vector and pCAG-H3GFP vector. pCAG-LSL expression vectors were diluted to 2 µg/µl. The plasmid solution with Fast Green (Sigma-Aldrich) was injected into the lateral ventricle of the mouse embryos and electroporated (37–40 V, 50 ms, 450 ms intervals, five pulses) using a NEPA Gene Electroporator (NEPA21, Nepa Gene, Chiba, Japan).

## Calculation of cell cycle length

Based on previous methods (Watanabe et al, 2015; Martynoga et al, 2005), the cell cycle length was calculated using BrdU and EdU. BrdU (50 mg/kg) and EdU (30 mg/kg) were administered intraperitoneally to pregnant mice 2 h and 30 min before sacrifice, respectively. The lengths of the S-phase ($T_S$) and total cell cycle ($T_C$) were calculated as follows: $T_S = 1.5 \times S_{cells}/L_{cells}$ and $T_C = T_S \times P_{cells}/S_{cells}$ [$L_{cells}$ = cells leaving S-phase (identified as TBR2$^+$, BrdU$^+$, EdU$^-$ cells); $S_{cells}$ = cells in S-phase (TBR2$^+$, EdU$^+$ cells); $P_{cells}$ = total proliferating cells (TBR2$^+$ cells)]. The length of M-phase ($T_M$) was calculated as follows: $T_M = T_C \times M_{cells}/P_{cells}$ [$M_{cells}$ = Average value of PH3/TBR2-double positive cells (Fig. 2B); $P_{cells}$ = Average value of TBR2$^+$ cells (Appendix Fig. S3C)].

## Cell culture and transfection

HEK293T (gifted from N. Masuyama) or Neuro2a (CCL-131; ATCC, VG, USA) cells were maintained in Dulbecco's modified Eagle's medium (DMEM, Sigma-Aldrich) supplemented with 10% fetal bovine serum (FBS) in a humidified atmosphere containing 5% $CO_2$ at 37 °C. Plasmids were transfected into cells using Lipofectamine LTX Reagent (Invitrogen) following the manufacturer's instructions. Transfected cells were cultured for 36–48 h. Mycoplasma contamination was not tested.

## Western blotting analysis

Lysates of HEK293T cells and cerebral cortices from the mouse brain at E14.5 were solubilized in sodium dodecyl sulfate (SDS) sample buffer and separated by SDS-polyacrylamide gel electrophoresis. Proteins transferred onto a nitrocellulose membrane were immunoblotted and visualized using HRP-conjugated secondary antibodies (Jackson ImmunoResearch) followed by ECL Prime (GE Healthcare, Chicago, IL). Signals were detected using FUSION SOLO S (Vilber, Paris, France).

## Generation of *Eomes*$^{T2A-d2EGFP}$ and *Eomes*$^{T2A-iCre}$ mice by CRISPR/Cas9

The method of KI mouse generation has been described previously (Inoue et al, 2021). The coding sequence of T2A-d2EGFP or T2A-iCre is inserted before the stop codon of the *Eomes* locus. The left (5′- GGAAACTCGCCCCCCATAAAGTGTGAGGACATTAA-CACTGAAGAGTACAGTAAAGACACCTCCAAAGGCATGGGG GCTTATTATGCTTTTTACACAAGTCCC-3′) and right (5′-GGA-TACATCAAAGGTGGAAGGCAAAAGTCTTTTTGGTAACC-TAGGCAAAGAACACAACAAAACACCACCAGGTCCATCTG-GAAAGGTTAAAGGTTAAAATAATGCT-3′) homology arms were connected to the T2A-d2EGFP sequence (Section S1). The T2A-iCre-BamHI sequence (Section S2) was connected to the left (297 base) and the right (313 base) homology arms. Cas9 protein, crRNA/tracrRNA, and donor ssDNA were microinjected into B6C3F1 mouse zygotes following standard protocols. The following three primers were used for genotyping the progenies using GoTaq DNA Polymerase (Promega, Madison, WI): for *Eomes*$^{T2A-d2EGFP}$, F (5′-GGTGTACAACAGCGCTTGCA-3′), R1 (5′-GACGTCACCGC ATGTTAGCA-3′, R2 (5′-TTTGGCGCCTTCTCTCAGAG-3′); for *Eomes*$^{T2A-iCre}$, F1 (5′-TAACGGTGAGAGAACCGTGC-3′), F2 (5′-TGTCCATCCCTGAAATCATGCAGG-3′), R (5′-GGTCCAAC CCTTCTCTTCCC-3′). The founders were backcrossed with C57BL/6N mice.

## In vitro digestion assay

The genomic region (813 bp) containing the *Eomes*-crRNA target sequences was PCR amplified using PrimeSTAR Max (Takara Bio, Shiga, Japan), and a pair of primers, F (5′-TAACGGTGAGA-GAACCGTGC-3′) and R (5′-AAAACACTCCTGCGTCCTCC-3′). Cas9 protein (50 ng/µL), *Eomes*-crRNA (15.8 ng/µL; 5′AAAG-GUUAAAAUAAUGCUCUAGGGUUUUAGAGGUAUGCUCUU UUG-3′), and tracrRNA (25.8 ng/µL; 5′-AAACAGCAUAGCAA GUUAAAAUAAGGCUAGUCCGUUAUCAACUUGAAAAAGU GGCACCGAGUCGGUGCU-3′) were incubated with the *Eomes*

target PCR products in Cas9 Nuclease Reaction Buffer for 60 min at 37 °C. Reactions were stopped using 10X DNA loading buffer containing 40% glycerol, 2% SDS, and 180 mM EDTA. The samples were analyzed by electrophoresis using 1.5% agarose gels.

## Cell sorting

The lateral parts of the cerebral cortices of the embryos were dissected and dissociated using a neuron dissociation solution (Fujifilm Wako Pure Chemical Corporation, Osaka, Japan). Dissociated cells were stained with Allophycocyanin (APC)-conjugated antibodies against CD133 (1:400; 141207; BioLegend). Cells were sorted on a FACS Aria fusion cell sorter (Becton Dickinson, Tokyo, Japan) with a 100-μm nozzle. The sorting gate for CD133 was set based on a previous study (Morimoto-Suzki et al, 2014).

## RT-qPCR

Total RNA was extracted using an RNeasy Micro Kit (QIAGEN). Purified total RNA was reverse-transcribed into cDNA using the ReverTra Ace qPCR RT kit (Toyobo, Osaka, Japan) according to the manufacturer's instructions. Real-time qPCR was performed with PowerUp SYBR Green Master Mix (Thermo Fisher Scientific) in a Light Cycler 96 (Roche Diagnostics, Basel, Switzerland), and relative expression levels were calculated using the 2Δ method. The quantified amount of target mRNA was normalized to that of internal control *Actb* mRNA. Primers used are listed in Table EV1.

## Co-immunoprecipitation

Nuclei were extracted with Nuclear Extraction Buffer (20 mM HEPES-KOH pH 7.9, 10 mM KCl, 0.1% Triton X-100, 20% Glycerol, 0.5 mM Spermidine, 1x Protease Inhibitor Cocktail (Roche)) from transfected Neuro2a cells or E14.5 cerebral cortices. The nuclei were dissolved with lysis buffer (50 mM Tris-HCl, pH 7.5, 125 mM NaCl, 1 mM EDTA, 5% glycerol, 0.5% NP-40, 1 mM PMSF, 1x Protease Inhibitor Cocktail (Roche), and 100 μg/μl RNaseA). The lysate was sonicated using a Bioruptor (Five cycles of 30 s ON and 30 s OFF) (SonicBio, Kanagawa, Japan). After centrifugation, the supernatant was incubated with mouse anti-FLAG (F1804; Sigma-Aldrich), rabbit anti-AUTS2 (HPA00390; Sigma-Aldrich) or rabbit IgG (011-000-003; Jackson ImmumoResearch)-conjugated dynabeads protein G (Thermo Fisher Scientific) for 4 h at 4 °C. After washing with lysis buffer, all precipitates were solubilized in SDS sample buffer and detected by western blotting with rabbit anti-FLAG (1:500; F7425; Sigma-Aldrich), rabbit anti-EZH2 (1:1000; 5246S; Cell Signaling Technology), rabbit anti-SUZ12 (1:1000; 3737S; Cell Signaling Technology) and rabbit anti-AUTS2 (1:500; HPA00390; Sigma-Aldrich) antibodies.

## Proximity ligation assay

EGFP+ cells were plated on cover grasses coated with 1 mg/ml poly-D-lysine (Sigma-Aldrich) for 1 h in a humidified atmosphere containing 5% $CO_2$ at 37 °C. Cells were fixed with 4% PFA for 20 min and permeabilized with 0.2% Triton X-100 for 15 min at RT. The Duolink Proximity Ligation Assay (Sigma-Aldrich) was performed following the manufacturer's instructions. Cells were incubated overnight at 4 °C with primary antibodies as follows; rabbit anti-AUTS2 antibody (1:400; HPA00390; Sigma-Aldrich)

and mouse anti-EZH2 antibody (1:100; 3147S; Cell Signaling Technology). Signals were acquired using a super-resolution confocal microscope SpinSR10 (Olympus).

## RNA-seq

Total RNA was extracted from sorted cells from six control (*Auts2*^fl(ex8)/fl(ex8)) and six *Auts2* cKO (*Emx1*^Cre/+; *Auts2*^fl(ex8)/fl(ex8)) mice at E15.5, using an RNeasy Micro Kit (QIAGEN). Quality analyses and quantification of the extracted RNA were performed using a NanoDrop and Qubit Fluorometer (Thermo Fisher Scientific). Sequencing libraries were prepared using the NEBNext Ultra II Directional RNA Library Prep Kit (New England BioLabs, Ipswich, MA) according to the manufacturer's instructions. RNA-seq libraries were sequenced using the Illumina HiSeq 4000 platform.

The obtained reads were processed using fastp to trim adapter sequences and remove low-quality nucleotides at the read ends (Chen et al, 2018). The processed reads were aligned to the reference mouse genome (GRCm38/mm10) using HISAT2 (Kim et al, 2015). Genome-wide expression levels were measured as transcripts per kilobase million (TPM) using StringTie (Pertea et al, 2016). The number of reads per gene was counted for each sample using StringTie (Pertea et al, 2016). Differentially expressed genes (DEGs) were identified using DESeq2 (Love et al, 2014). Gene Ontology biological process analysis was performed using the DAVID bioinformatics Resources (National Institute of Allergy and Infectious Diseases, National Institute of Health; https://david.ncifcrf.gov).

## CUT&Tag

CUT&Tag was performed as previously described with some modifications (Kaya-Okur et al, 2019). Briefly, 100,000 EGFP+ cells sorted by FACS were used for each experimental replicate. Cell nuclei were extracted with Nuclear Extraction Buffer (20 mM HEPES-KOH pH 7.9, 10 mM KCl, 0.1% Triton X-100, 20% Glycerol, 0.5 mM Spermidine, 1x Protease Inhibitor Cocktail (Roche)), bound to wheat germ agglutinin (WGA) coated magnetic beads (Bangs Laboratories, Fishers, IN), and incubated for overnight at 4 °C with a primary antibody in Antibody Binding Buffer (20 mM HEPES pH 7.5, 150 mM NaCl, 0.5 mM Spermidine, 2 mM EDTA, 0.01% Digitonin, 1x Protease Inhibitor Cocktail (Roche)). The following primary antibodies were used, rabbit anti-AUTS2 (HPA000390; Sigma-Aldrich), rabbit anti-RING1B (5694S; Cell Signaling Technology), rabbit anti-H2AK119ub (8240S; Cell Signaling Technology), rabbit anti-H3K27me3 (9733S; Cell Signaling Technology), rabbit anti-H3K4me3 (9751S; Cell Signaling Technology), rabbit anti-H3K27ac (8173S; Cell Signaling Technology), rabbit anti-EZH2 (5246S; Cell Signaling Technology), and rabbit anti-SUZ12 (3737S; Cell Signaling Technology). The beads were incubated with a secondary antibody (EpiCypher, Durham, NC) in a Dig-wash Buffer (20 mM HEPES pH 7.5, 150 mM NaCl, 0.5 mM Spermidine, 0.01% Digitonin, 1x Protease Inhibitor Cocktail (Roche)) for 30 min at RT. The beads were washed twice in Dig-wash Buffer and incubated with protein A/G-Tn5 conjugates (EpiCypher) in Dig-wash 300 Buffer (20 mM HEPES pH 7.5, 300 mM NaCl, 0.5 mM Spermidine, 0.01% Digitonin, 1x Protease Inhibitor Cocktail) for 1 h at RT. After washing twice with Dig-wash 300 Buffer, the beads were resuspended with Tagmentation

Buffer (20 mM HEPES pH 7.5, 300 mM NaCl, 0.5 mM Spermidine, 0.01% Digitonin, 10 mM MgCl$_2$, 1x Protease Inhibitor Cocktail (Roche)) and incubated for 1 h at 37 °C. Furthermore, 0.5 M EDTA (final 17 mM), 10% SDS (final 0.1%), and 20 mg/ml Proteinase K (final 0.17 mg/ml) were added to 100 μl of the sample and incubated at 55 °C for 1 h to stop tagmentation. Sequencing libraries were amplified using NEBNext HiFi 2x Master Mix (NEB) as follows: 72 °C for 5 min and 98 °C for 30 s followed by 15 cycles (for histone modifications), 18 cycles (for AUTS2 and RING1B) or 20 cycles (for EZH2 and SUZ12) of 10 s at 98 °C and 10 s at 63 °C. The DNA fragments and libraries were purified using a DNA cleaner and concentrator (ZYMO Research, Irvine, CA). For input, EGFP$^+$ cells were sonicated with a Bioruptor (25 cycles of 30 s ON and 30 s OFF) (SonicBio), and the DNA fragment was purified. Input libraries were prepared using ThruPLEX (Takara Bio) following the manufacturer's instructions. The size distribution and concentration of libraries were determined using Agilent 4150 TapeStation analysis (Agilent Technologies, Santa Clara, CA). Deep sequencing of CUT&Tag was performed using 150-bp paired-end sequencing on the Illumina HiSeq X or NovaSeq 6000 platform.

### CUT&Tag data analysis

Paired-end reads were processed with fastp to trim adapters and remove short- and low-quality reads (Chen et al, 2018), followed by alignment to the reference mouse genome mm10 using Bowtie2 (Langmead and Salzberg, 2012). SAM files were converted to the BAM format using SAMtools (Li et al, 2009). PCR duplicates were removed using Picard Mark Duplicates (https://github.com/broadinstitute/picard). Peaks were identified using an IDR pipeline (https://hbctraining.github.io/Intro-to-ChIPseq/lessons/07_handling-replicates-idr.html) and MACS2 (Zhang et al, 2008) (parameters: for AUTS2 and SUZ12; -q 0.05, for the others; -q 0.01) from pseudo-replicates with the input library as a control. Genomic peak annotation was performed using the R package ChIPseeker (Yu et al, 2015). Genomic annotation was performed using GREAT software (McLean et al, 2010). Reads were normalized to one million paired-end reads for histone modifications or 100 thousand paired-end reads for AUTS2, RING1B, EZH2, and SUZ12 to generate BigWig files using bamCompare from deepTools (Ramirez et al, 2016). Heat maps and density plots were generated using DeepTools. Log$_2$ enrichment of CUT&Tag signals was performed using multiBigwigSummary in BED file mode from deepTools. Bed files of the TSS of upregulated or unchanged genes were generated using UCSC (https://genome.ucsc.edu/). BigWig files were visualized using the Integrative Genomics Viewer (IGV) (Robinson et al, 2011). We merged three independent samples into one sample with the cat command, followed by the above analysis using three merged samples to supplement the read coverage of the EZH2 and SUZ12 libraries.

### Processing of scRNA-seq data

Raw count data of only cortical cells at E13.5 and E15.5 was obtained from GSE107122 and processed with the Seurat R package as described (Hao et al, 2021). For clustering, the parameter "resolution = 0.5" was used. Annotation of each cluster was performed using representative marker genes such as *Sox2*, *Eomes*, *Neurod1*, *Tbr1*, *Bcl11b*, *Satb2*, and *Pou3f2*. Raw count data of the mouse cerebral cortex at E12.5 and E15.5 were obtained from the Single Cell Portal under accession number SCP1290 (Di Bella et al,

2021). The raw count data were normalized using the NormalizeData function in the Seurat package, and the classification of cell types was based on the original annotations provided in the metadata obtained from the Single Cell Portal (SCP1290).

### Statistical analysis

Statistical analyses, with the exception of RNA-seq, were performed using GraphPad Prism 7 software (GraphPad Software, La Jolla, CA, USA). No statistical methods were used to pre-determine the sample size; however, the sample size was determined based on previous studies (Hori et al, 2014; Watanabe et al, 2015; Sessa et al, 2008; Cardenas et al, 2018; Kostic et al, 2019; Eto et al, 2020). The sample size was indicated in the figure legends. The normal distribution of data was confirmed using the Shapiro-Wilks test. If significant, the data were compared by a nonparametric Mann–Whitney *U*-test. The data with normal distribution were compared by a two-tailed unpaired *t*-test. For comparison of more than two groups, the normally distributed data was compared by a one-way analysis of variance (ANOVA) followed by Dunnett's or Turkey's multiple comparison test, and the non-normally distributed data was compared by a Kruskal–Wallis test. No randomization of samples was performed. Data analyses were not performed blinded to the genotype. Statistical significance was set at a *P* value < 0.05.

## Data availability

RNA-seq data and CUT&Tag data have been deposited into the DNA Data Bank of Japan (DDBJ) database with the accession number PRJDB17034 (https://ddbj.nig.ac.jp/search/entry/bioproject/PRJDB17034) and PRJDB17038 (https://ddbj.nig.ac.jp/search/entry/bioproject/PRJDB17038), respectively. Source data are provided with this paper. KI mouse line generated in this study and used research materials are available on request to MH with a completed Material Transfer Agreement. Every code used in this study was executed with default settings and the parameters indicated in the Methods sections. Detailed codes used in this study are available upon request to MH.

The source data of this paper are collected in the following database record: biostudies:S-SCDT-10_1038-S44318-024-00343-7.

## Peer review information

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

## Acknowledgements

We thank Yusuke Kishi (The University of Tokyo) for the guidance on the use of FACS, and Atsushi Iwama (The University of Tokyo) for the discussions on the CUT&Tag data. This work was supported by JSPS KAKENHI (Grant Numbers JP16H06531 to YG, JP22K15134 to KS, and JP22H02730 to MH); AMED (Grant Numbers 24wm0425005h0004 and 24ek0109764h0001 to MH, 24wm0625508h0001 to SM), an Intramural Research Grant of NCNP (4-5 to MH, 4-6, 6-9 to MH and SM); Japan Health Research Promotion Bureau (JH) under Research Fund (2020-B-07 and 2024-D-01 to MH); Multilayered Stress Diseases (JPMXP1323015483 to MH), and Tokumori Yasumoto Memorial Trust (MH). The cartoon for the synopsis was created with Biorender.com.

## Author contributions

**Kazumi Shimaoka**: Resources; Data curation; Formal analysis; Funding acquisition; Investigation; Visualization; Methodology; Writing—original draft; Project administration; Writing—review and editing. **Kei Hori**: Conceptualization; Resources; Data curation; Methodology; Writing—original draft; Project administration. **Satoshi Miyashita**: Formal analysis; Funding acquisition; Methodology; Writing—review and editing. **Yukiko U Inoue**: Resources; Investigation; Methodology; Writing—review and editing. **Nao KN Tabe**: Investigation. **Asami Sakamoto**: Investigation. **Ikuko Hasegawa**: Investigation. **Kayo Nishitani**: Investigation. **Kunihiko Yamashiro**: Investigation. **Saki F Egusa**: Investigation. **Shoji Tatsumoto**: Formal analysis; Methodology. **Yasuhiro Go**: Resources; Data curation; Formal analysis; Funding acquisition; Methodology; Writing—review and editing. **Manabu Abe**: Resources. **Kenji Sakimura**: Resources; Writing—review and editing. **Takayoshi Inoue**: Resources; Methodology; Writing—review and editing. **Takuya Imamura**: Methodology; Writing—review and editing. **Mikio Hoshino**: Conceptualization; Supervision; Funding acquisition; Writing—original draft; Project administration; Writing—review and editing.

Source data underlying figure panels in this paper may have individual authorship assigned. Where available, figure panel/source data authorship is listed in the following database record: biostudies:S-SCDT-10_1038-S44318-024-00343-7.

## Disclosure and competing interests statement

The authors declare no competing interests.

