## [Peer Review File · The EMBO Journal]

The microcephaly-associated transcriptional regulator AUTS2 cooperates with Polycomb complex PRC2 to produce upper-layer neurons in mice

Mikio Hoshino, Kazumi Shimaoka, Kei Hori, Satoshi Miyashita, Yukiko Inoue, Nao Tabe, Asami Sakamoto, Ikuko Hasegawa, Kayo Nishitani, Kunihiko Yamashiro, Saki Egusa, Shoji Tatsumoto, Yasuhiro Go, Manabu Abe, Kenji Sakimura, Takayoshi Inoue, and Takuya Imamura

Corresponding author(s): Mikio Hoshino (hoshino@ncnp.go.jp)

Review Timeline:

Submission Date:	4th Apr 24
Editorial Decision:	31st May 24
Revision Received:	30th Oct 24
Editorial Decision:	14th Nov 24
Revision Received:	20th Nov 24
Editorial Decision:	21st Nov 24
Revision Received:	22nd Nov 24
Accepted:	28th Nov 24

Editor: Ioannis Papaioannou

Transaction Report:

Dear Prof. Hoshino,

Thank you again for submitting your manuscript EMBOJ-2024-117495-T for consideration by The EMBO Journal. It has been seen by two experts in the field, and we have received their comments, which I have already shared with you (included again below). I would also like to thank you for your detailed responses to the referee comments and your provisional revision plan, which were very helpful for us to reach a fair and balanced decision on your manuscript.

Given the referees' encouraging comments and recommendations, as well as your willingness to address their concerns in an appropriately revised version of your manuscript, I would like to invite you to submit the revised manuscript along with a detailed point-by-point response addressing all referees' comments. I should add that it is EMBO Journal policy to allow only a single round of major revision, and acceptance of your manuscript will therefore depend on the completeness of your responses in this revised version. Please let me know if you have any further questions or comments that you would like to discuss with me.

We generally allow three months as standard revision time (August 30, 2024). As a matter of policy, competing manuscripts published during this period will not negatively impact our assessment of the conceptual advance presented by your study. However, we request that you contact us as soon as possible upon publication of any related work, to discuss how to proceed. Should you foresee a problem in meeting this three-month deadline, please let us know in advance and we may be able to grant an extension.

Thank you for the opportunity to consider your work for publication in The EMBO Journal. I look forward to your revision.

Best regards,

Ioannis

Instructions for preparing your revised manuscript

1. When you are ready to submit the revision, please upload:

- A Word file of the manuscript text (including legends of main Figures, EV Figures and Tables). Please make sure that changes are highlighted (or "tracked") to be clearly visible.

- Individual production-quality figure files (one file per figure). When assembling your figures, please refer to our figure preparation guidelines in order to ensure proper formatting and readability in print as well as on screen:

If the data shown in a figure are obtained from n {less than or equal to} 2, please use scatter plots showing the individual data points.

- i. the name of the statistical test used to generate error bars and P values
- ii. the number (n) of independent experiments (please specify technical or biological replicates) underlying each data point (discussion of statistical methodology can be reported in the Materials and Methods section, but figure legends should contain a basic description of n , P , and the test applied)
- iii. the nature of the bars and error bars (s.d., s.e.m.).

- A point-by-point response to the referees' comments, with a detailed description of the changes made (as a word file). All referees' concerns must be fully addressed and their suggestions taken on board. When preparing your letter of response to the referees' comments, please bear in mind that this will form part of the Review Process File and will therefore be available online to the community. Please note that you have the possibility to opt out of the transparent process at any stage prior to publication by letting the editorial office know (contact@embojournal.org); if you do opt out, the Review Process File link will point to the following statement: "No Review Process File is available with this article, as the authors have chosen not to make the review process public in this case.". For more details on our Transparent Editorial Process, please visit our website:

<https://www.embopress.org/page/journal/14602075/authorguide#transparentprocess>

- Expanded View (EV) files (replacing Supplementary Information) that are collapsible/expandable online. A maximum of 5 EV Figures can be typeset. EV Figures should be cited as "Figure EV1, Figure EV2" etc. in the text, and their respective legends should be included in the manuscript file after the legends of regular figures. See detailed instructions regarding Expanded View files here:

- For the figures that you do NOT wish to display as Expanded View figures, they should be bundled together with their legends in a single PDF file called "Appendix", which should start with a short Table of Contents (including page numbers). Appendix figures should be referred to in the main text as: "Appendix Figure S1, Appendix Figure S2" etc. Please see detailed instructions here: <https://www.embopress.org/page/journal/14602075/authorguide#expandedview>

- A complete author checklist, which you can download from our author guidelines (<https://www.embopress.org/page/journal/14602075/authorguide>). Please note that the checklist will also be part of the Review Process File.

2. Please note that no statistics should be calculated and shown in Figures if $n=2$. Please also note that each p value should be reported as an exact value.

3. Before submitting your revision, primary datasets (and computer code, where appropriate) produced in this study need to be deposited in appropriate public databases (see <https://www.embopress.org/page/journal/14602075/authorguide#dataavailability>).

In particular, you are kindly requested to make sure that all RNA-seq and CUT&Tag data produced in your study have been deposited in appropriate databases and are accessible to the referees. The accession numbers and databases should be listed in a formal "Data availability" section (placed after Materials and Methods) that follows the model below (see also <https://www.embopress.org/page/journal/14602075/authorguide#dataavailability>):

Data availability

- RNA-seq data: Gene Expression Omnibus GSE46843 (<https://www.ncbi.nlm.nih.gov/geo/query/acc.cgi?acc=GSE46843>)
- [data type]: [name of the resource] [accession number/identifier/doi] ([URL or identifiers.org/DATABASE:ACCESSION])

*** All links should resolve to a page where the data can be accessed. ***

*** Please remember to provide in the Data availability section of your revised manuscript reviewer passwords if the datasets are not yet public. ***

*** The Data Availability Section is restricted to new primary data that are part of this study. In case you have no data that require deposition in a public database, please state so instead of referring to the database: "Our study includes no data deposited in public repositories." under the heading "Data availability". ***

4. Please check that the title and the abstract of the manuscript are brief, yet explicit, even to non-specialists. The length of the title should not exceed 100 characters, and the abstract should be a single paragraph not exceeding 175 words.

5. Please also note our reference format: <https://www.embopress.org/page/journal/14602075/authorguide#referencesformat>.

7. Please remember: digital image enhancement is acceptable practice, as long as it accurately represents the original data and conforms to community standards. If a figure has been subjected to significant electronic manipulation, this must be noted in the figure legend or in the "Materials and Methods" section. The editors reserve the right to request original versions of figures and the original images that were used to assemble the figure.

8. Our journal encourages inclusion of data citations in the reference list to directly cite datasets that were obtained from public databases. Data citations in the article text are distinct from normal bibliographical citations and should directly link to the database records from which the data can be accessed. In the main text, data citations are formatted as follows: "Data ref: Smith et al, 2001" or "Data ref: NCBI Sequence Read Archive PRJNA342805, 2017". In the Reference list, data citations must be labeled with "[DATASET]". A data reference must provide the database name, accession number/identifiers, and a resolvable link to the landing page from which the data can be accessed at the end of the reference. Further instructions are available at:

<https://www.embopress.org/page/journal/14602075/authorguide#referencesformat>.

9. We request authors to consider both actual and perceived competing interests. Please review our policy (<https://www.embopress.org/page/journal/14602075/authorguide#conflictsofinterest>) and update your competing interests statement if necessary. Please name this section 'Disclosure and competing interests statement' and place it after the Acknowledgements section.

10. Please note that all corresponding authors are required to provide an ORCID ID upon submission of a revised manuscript (<https://orcid.org/>). Please find instructions on how to link your ORCID ID to your account in our manuscript tracking system in our Author guidelines (<https://www.embopress.org/page/journal/14602075/authorguide#authorshipguidelines>).

11. We use CRediT to specify the contributions of each author in the journal submission system. CRediT replaces the author contribution section, which should be removed from the manuscript. Please use the free text box to provide more detailed descriptions. See also guide to authors: <https://www.embopress.org/page/journal/14602075/authorguide#authorshipguidelines>.

13. We would also welcome the submission of cover suggestions or motifs to be used by our Graphics Illustrator in designing a cover.

14. Please use the link below to submit your revision:
<https://emboj.msubmit.net/cgi-bin/main.plex>

Referee #1:

In the present manuscript, Shimaoka et al investigate the role of *Auts2* in the development of the cortex using conditional mouse models. Mutations of *AUTS2* in humans are associated with autism spectrum disorders, intellectual disability and microcephaly. The authors show that mutant mice display a reduction in upper-layer cortical neurons, which the authors propose to be caused by a reduction in intermediated progenitor cell (IPC) division. By performing knockdown and rescue experiments, the authors show that full length nuclear *Auts2* is required for the regulation of proliferation of neural progenitor cells. Since *Auts2* has previously been shown to function in the nucleus and to be associated with the Polycomb Repressive Complex 1 (PRC1), the authors continue to investigate the role of *Auts2* in gene regulation. To specifically isolate IPCs, they generate a novel transgenic reporter line which expresses GFP from the *Eomes* locus. Transcriptome analysis identifies roughly 100 up- and down-regulated genes, respectively, including the *Robo1* gene which encodes a receptor that is activated by SLIT-family proteins. The authors show that *Robo1* overexpression can rescue the IPC phenotype in *Auts2* mutant mice. Finally, CUT&Tag analysis reveals that *Auts2* binds predominantly to TSS and that H3K27me3 is reduced in *Auts2* mutant mice, which might result from a direct interaction of *Auts2* and PRC2 in IPCs.

Overall, the study is highly interesting as it dissects the role of *Auts2*, a gene implicated in a human developmental disorder, in the development of the cerebral cortex. In particular, the authors propose a cellular and molecular mechanism that may contribute to the development of the microcephaly phenotype described in patients. The manuscript is well written and the data that is presented is of high quality and appropriately quantified.

Before publication, a number of points should be addressed to further clarify some issues and to strengthen the conclusions:

1) Was the analysis performed in two different mouse models (*Emx1Cre/+; Auts2^{flox/flox}* versus *Auts2^{del8/del8}* mice, respectively)? The two models should be described in detail and a justification should be provided.

2) In line 142, it says that "A few CUX1+ neurons were consistently observed in the deep layers (V and VI) of *Auts2* cKO homozygotes at P15 (Fig. 1c)." The corresponding images suggest that the same is true for the control.

3) The heading "FL-AUTS2 in nuclei is required for neuron production" is misleading as no neuronal marker was analysed.

4) A DNA staining should be shown in addition to immunofluorescence to allow assessment of brain tissue architecture and orientation within the tissue. The boundaries of the zones should be indicated by a line. How were the boundaries determined?

- 5) The authors have generated a new reporter line and state that "EGFP was exclusively expressed in TBR2-immunopositive cells" (Fig 4f). This is not supported by any quantification. The GFP-positive cells should be characterized with respect to Sox2/Pax6, Tbr2 and neuron marker expression.
- 6) For the isolation of RGCs, CD133 expression was used. However, the population that was gated only included the most highly CD133 expressing cells, which represents a tiny fraction and does not match the expected numbers of RGC at E15.5 in the mouse. Even though the RGC fraction expresses canonical RGC marker genes, it is not clear what this sub-population represents and therefore the RGC transcriptome data should be treated with caution.
- 7) The authors focus on the Robo1 gene as a candidate gene among IPC-upregulated genes. The extensive work by Victor Borrell's group on the role of Robo-signaling in cortical neurogenesis and IPC regulation should be mentioned when introducing the gene (Borrell et al, Neuron 2012; Cardenas et al, Cell 2018).
- 8) The authors conclude that "These findings suggest that AUTS2 promotes the proliferation of IPCs and the production of an appropriate number of neurons by repressing Robo1 expression." Can that be concluded? In Figures 2, 3 and 5, different analyses were performed. Cell cycle length is increased. How does this fit in with the regulation of proliferation? And with previous data on different cell cycle length in proliferative vs neurogenic NPCs?
- 9) Related to Figure 6e, which cluster contains the differentially expressed genes?
- 10) Do Ring1b, H2AK119ub and H3K27me3 levels change when all AutS2-bound genes are interrogated (rather than only differentially expressed genes)?
- 11) How does the data related to PRC2 integrate with what is already known on the role of PRC2 and H3K27me3 in cortical development (Pereira, PNAS 2010; Albert, EMBO J 2017; Telley, Science 2019)? These publications should be discussed.
- 12) The genomics data was not accessible to the reviewer.

Referee #2:

The study investigates the mechanisms underlying changes in cortical thickness in mouse models of the AUTS2 syndrome. In humans the syndrome is associated with mutations of the AUTS2 gene causing a wide range of phenotypes including intellectual disability, autism epilepsy and depression. The analysis of the gene function is complicated by the size of the gene and the presence of different length are expressed at different time and in different regions during development. More than 60% of the patients also show microcephaly and frequent alteration include loss of volume in the cortex and hypoplasia in the cerebellum and corpus callosum. Hypoplasia has been also observed in mice models in the cerebellum and in the dentate gyrus. A reduced progenitor proliferation has also been observed in brain organoids. However, loss of cortical volume and change in proliferation has not been reproduced in a mouse model. The main novelty of the paper resides in the reproduction of the microcephalic phenotype in mutant mice. The authors conclude that this is associated to a change in the proliferation of IPCs and not RGCs. Mechanistically, the authors propose that mutation of the gene leads to overexpression of Robo1, which in turns affects Cell proliferation. Whereas this is a novel finding, it is in apparent contradiction with a recent study showing that patient derived organoids display altered proliferation of RGCs and not IPCs. The possible reasons underlying this discrepancy should be at least discussed. Moreover, lack of direct manipulation of gene expression in IPCs, of investigation of residual AUTS2 expression and analysis of Robo1 changes in heterozygous mice impact the strength of the conclusions.

- 1) The study is very confusing due to the fact that different experiments are done on heterozygous or homozygous KO, which show microcephaly, or animals lacking only the terminal part of the protein, which have similar brain size. The reason for using this latter model in some experiments, see also below, is not clear. Differences in the pattern of AUTS2 expression in the various models are never investigated in RGC and IPCs only reference is made to previous studies but due to the complexity of the system and lack of analysis in heterozygous mice I feel this issue should be addressed.
- 2) A main finding in the paper is the cortical volume reduction which in mice seems to be associated to changes in cell cycle progression of IPCs but not RGCs. A recent study investigating the effect of AUTS2 in organoids reached recently opposite conclusions indicating that the gene behaves differently in humans and mice. Is the deletion of the protein similarly efficient in the two populations of cells? What about expression? The authors provide evidence that AUTS 2 is similarly expressed at E13.5 in the two populations. However, most of the experiments to monitor apical progenitor populations are done at E12.5
- 3) Figure1 the authors conclude that early generated neurons at E12.5 are not affected by AutS2 ablation and interpret this as evidence that AutS2 affects primarily IPC behavior. However, would be the lack of an effect at E12.5 predictable due to the fact that at this early age AutS2 protein is hardly expressed in the cortex? Moreover, in figure1 C the thickness of the deep layer appears affected at E15, also shown in the quantification, but not in postnatal animals (Fig. 1f). Does this suggest a change in differentiation or migration of the early born neurons?

- 4) Figure 2 and extended figure 3 it is unclear why these analyses are performed in a mouse model that does not show microcephaly. How to reconcile the apparent change in proliferation of IPCs cells with the lack of a microcephalic phenotype and the overall conclusion that a change in IPC proliferation leads to microcephaly? Indeed, these results suggest that when the change in proliferation is limited to TRB2+ cells there is no change in cortical volume.
- 5) More on figure 2 and other cellular quantification. Were the number of cells counted? In many of the figures, including figure 2, although the number of the cells appears not affected the distribution of the cells along the apical/basal axis, i.d. the distance of the cell from the lateral ventricle, appears to be altered. This is particularly visible in Trb2+/Ki67+ cells. A count of the cells on bins parallel to the lateral ventricle should be performed
- 6) In figure 3 are the changes in Ki67+ cells obtained after the various manipulations happening only in RGCs, also in IPCs or in both cell groups?
- 7) In figure 4 gate settings leads to sorting of only a small proportion of Cd133+/GFP- cells and to include a consistent proportion of CD133+ cells in the GFP+ populations, which according to immunostaining shown in extended data Figure4F may represent GFP+ cells in the VZ not expressing TRB2 protein. Consistent with this the difference in Prom1 expression between the two sorted populations is minimal. Is there a reason for this particular choice? How were the sorting gates set? Please add images of sorting plots from mutant mice. Could this sorting setting explain the apparent contradiction that in these experiments Auts2 gene is more expressed in GFP+ cells than in CD133 cells whereas in extended data figure 3 Auts2 transcripts appear to be similarly expressed in the two populations? The sorting of a specific subpopulation of CD133+ cells may also explain the fact that gene ontology analysis of the differentially regulated genes in this population, albeit involving more genes than in the putative IPCs, highlights only the angiogenesis process.
- 8) IPCs are highly proliferating cells. However, Ki67 expression is higher in the FACS isolated RGCs than in GFP+ putative IPCs. What is the percentage of isolated cells that is Ki67 positive?
- 9) Figure 5 does OE of Robo1 affects number of apically dividing TRb2- progenitors? Does the manipulation alter the distribution of TRB2 expression or of the electroporated cells? From the figure shown there appears more TRB2+ cells close to the apical side of the VZ and that less green cells are present in the VZ.
- 10) It has been shown that Robo1/ slit signaling is required to maintain ventricular mitosis between E12.5 and E15.5 therefore manipulation of Robo1 expression should be performed specifically in IPCs to monitor the effect in this population. Moreover, it has been shown that the Robo1/ slit signaling interacts with notch to affect proliferation Therefore, cell non-autonomous mechanisms cannot be excluded. Does the manipulation change the number of proliferating cells in the non-transfected population?
- 11) Are levels of Robo protein also changed in Auts2 KO homo and heterozygous? Why to focus on Robo1 and not also on its ligand Slit1 whose expression, both in RGC and IPCs, is even more affected by the mutation than Robo1?
- 12) What is the profile of Auts2 binding in Cut and Tag analysis of IPCs isolated from heterozygous mutant mice? Are repressive marks altered at the Robo1 locus also in this case?
- 13) The interpretation of the effect on IPCs proliferation illustrated in Figure 8f is again complicated by the fact that the manipulation does not target specifically IPC cells and therefore the effect on the number of mitotic IPCs could be a secondary one.

Responses to Referees.**Referee #1**

In the present manuscript, Shimaoka et al investigate the role of *Auts2* in the development of the cortex using conditional mouse models. Mutations of *AUTS2* in humans are associated with autism spectrum disorders, intellectual disability and microcephaly. The authors show that mutant mice display a reduction in upper-layer cortical neurons, which the authors propose to be caused by a reduction in intermediated progenitor cell (IPC) division. By performing knockdown and rescue experiments, the authors show that full length nuclear *Auts2* is required for the regulation of proliferation of neural progenitor cells. Since *Auts2* has previously been shown to function in the nucleus and to be associated with the Polycomb Repressive Complex 1 (PRC1), the authors continue to investigate the role of *Auts2* in gene regulation. To specifically isolate IPCs, they generate a novel transgenic reporter line which expresses GFP from the *Eomes* locus. Transcriptome analysis identifies roughly 100 up- and down-regulated genes, respectively, including the *Robo1* gene which encodes a receptor that is activated by SLIT-family proteins. The authors show that *Robo1* overexpression can rescue the IPC phenotype in *Auts2* mutant mice. Finally, CUT&Tag analysis reveals that *Auts2* binds predominantly to TSS and that H3K27me3 is reduced in *Auts2* mutant mice, which might result from a direct interaction of *Auts2* and PRC2 in IPCs.

Overall, the study is highly interesting as it dissects the role of *Auts2*, a gene implicated in a human developmental disorder, in the development of the cerebral cortex. In particular, the authors propose a cellular and molecular mechanism that may contribute to the development of the microcephaly phenotype described in patients. The manuscript is well written and the data that is presented is of high quality and appropriately quantified.

Before publication, a number of points should be addressed to further clarify some issues and to strengthen the conclusions:

1) Was the analysis performed in two different mouse models (*Emx1Cre/+; Auts2^{flox/flox}* versus *Auts2^{del8/del8}* mice, respectively)? The two models should be described in detail and a justification should be provided.

I apologize for the confusing and misleading explanation of these different mouse models in the original manuscript. The genomic changes in the knocked-out cells are identical in the two mouse models. The only difference is whether it is a systemic conventional KO or a conditional KO. Therefore, since the genomic changes are the same in the knocked-

out cells, the changes in protein expression are also expected to be the same.

In the original manuscript, we used two different names for the alleles of the *Auts2* gene (*Auts2^{fllox}* and *Auts2^{del8}*). However, several different types of *Auts2^{fllox}* alleles have been reported from other groups. The notation for alleles has been changed in the revised manuscript to avoid confusion.

- (i) *Auts2^{fllox}* in the previous manuscript is denoted as *Auts2^{fl(ex8)}* in the revised manuscript
- (ii) *Auts2^{del8}* in the previous manuscript is denoted as *Auts2^{del(ex8)}* in the revised manuscript.

In the *Auts2^{fl(ex8)}* allele, two loxP sequences are inserted on either side of the exon8 of the *Auts2* gene. Therefore, in *Emx1^{Cre/+}; Auts2^{fl(ex8)/fl(ex8)}* individuals, exon8 of the *Auts2* gene is removed forebrain-specifically (Hori et al, 2014, *Cell Reports*, 9, 2166-2179).

Regarding *Auts2^{del(ex8)}*:

When *CAG-Cre* transgenic mice, which systemically express Cre recombinase, were crossed with *Auts2^{fl(ex8)/+}*, exon8 was removed in the germ-line (sperm) of the individuals born from the cross (*Auts2^{fl(ex8)/+}; CAG-Cre*). Then, in further descendants of that individual, heterozygotes for the allele (*Auts2^{del(ex8)/+}*) were born with a systemic loss of exon8. The *Auts2^{del(ex8)}* allele was created this way (Hori et al., 2014, *Cell Reports*, 9, 2166-2179). Unfortunately, *Auts2^{del(ex8)}* homozygotes (*Auts2^{del(ex8)/del(ex8)}*) die just after birth. Therefore, in this study, we basically use the *Auts2^{del(ex8)}* allele, a conventional KO, for analysis at embryonic stages, and the *Auts2^{fl(ex8)}; Emx1^{Cre}*, a conditional KO, for postnatal analysis.

In both types of KO mice, the changes in protein expression in the knocked-out cells are thought to be identical because the knocked-out cells have the same genomic mutation, that is, the loss of the exon8. Eventually, FL (full-length)-AUTS2 and S-AUTS2-var.1 are entirely lost, while S-AUTS2-var.2 expression is increased, as was shown in the previous paper (Hori et al., 2014, *Cell Reports*, 9, 2166-2179).

To assess this, we have added the following sentences in Results.

“Previously, we generated a floxed allele of the *Auts2* gene, which carried two loxP sequences on either side of the exon 8, and named it *Auts2^{fllox}*. By crossing mice carrying this allele with mice with the *CAG-Cre-Tg* allele that ubiquitously expressed Cre (Sakai & Miyazaki, 1997), we successfully obtained the allele that lacked the exon 8, which we named *Auts2^{del8}* (Hori et al, 2014). However, as several research groups have reported different types of floxed alleles for this gene (Gao et al, 2014; Castanza et al, 2021; Li et al, 2022), we have decided to rename *Auts2^{fllox}* and *Auts2^{del8}* as *Auts2^{fl(ex8)}* and

Auts2^{del(ex8)}, respectively (Appendix Fig. S1A). Since homozygotes for *Auts2*^{del(ex8)} (*Auts2*^{del(ex8)/del(ex8)}) were postnatally lethal, we produced forebrain-specific *Auts2* conditional knockout (*Auts2* cKO) homozygous and heterozygous mice (*Emx1*^{Cre/+}; *Auts2*^{fl(ex8)/fl(ex8)}, *Emx1*^{Cre/+}; *Auts2*^{fl(ex8)/+}) by utilizing the *Emx1*^{Cre} allele (Hori *et al*, 2020). We basically use global *Auts2* KO (*Auts2*^{del(ex8)/del(ex8)}, *Auts2*^{del(ex8)/+}) and forebrain specific cKO (*Emx1*^{Cre/+}; *Auts2*^{fl(ex8)/fl(ex8)}, *Emx1*^{Cre/+}; *Auts2*^{fl(ex8)/+}) for analyses at embryonic and postnatal stages, respectively.”

We have added **Appendix Fig. S1A** to explain the genomic structure of each allele. The AUTS2 protein produced from each allele is shown in **Appendix Fig. S1B**.

The above will help readers understand the two alleles and eliminate confusing and misleading elements.

2) In line 142, it says that "A few CUX1+ neurons were consistently observed in the deep layers (V and VI) of *Auts2* cKO homozygotes at P15 (Fig. 1c)." The corresponding images suggest that the same is true for the control.

We were sorry, but the description in the original manuscript was inappropriate. As the reviewer pointed out, a few CUX+ neurons could be observed both in control and mutants. In the original manuscript, we tended to describe that there was a difference in the distribution of CUX1+ neurons at embryonic stages, but that difference disappeared by P10. Therefore, we have deleted the last sentence (with a strike-through) from the following description in Results, as below.

“We previously observed that most E14.5-labeled neurons did not reach the pial side by E18.5 in *Auts2* KO mice (*Auts2*^{del(ex8)/del(ex8)} and *Auts2*^{del(ex8)/+}), whereas those in wild-type (WT) mice reached this level, suggesting that AUTS2 is involved in neuronal migration (Hori *et al*, 2014). However, we found that the distribution of E15.5-labeled and E12.5-labeled cells did not differ between controls and homozygotes or heterozygotes at PD10 (Appendix Fig. S1E). This suggests that *Auts2*-deficient neurons migrated slowly but eventually reached their final targets. ~~A few CUX1+ neurons were consistently observed in the deep layers (V and VI) of *Auts2* cKO homozygotes at P15 (Fig. 1c).”~~

3) The heading "FL-AUTS2 in nuclei is required for neuron production" is misleading as no neuronal marker was analysed.

We appreciate the reviewer's valuable comment.

In Fig. 3 of the original manuscript, the experiment aimed to evaluate the number of neurons produced two days after the electroporation of control or shRNA for *Auts2*.

However, as the reviewer mentioned, we did not perform immunostaining with a neuronal marker. We have to admit that the heading was misleading.

In the revised manuscript, we have improved the experimental design using a neuronal marker (HuC/D). We directly counted postmitotic neurons (HuC/D-positive, Ki67-negative cells) in the electroporated (GFP+) cells (**Fig. 3A, B**), obtaining identical results as in the previous experiment. Now, we believe that the heading is accurate.

4) A DNA staining should be shown in addition to immunofluorescence to allow assessment of brain tissue architecture and orientation within the tissue. The boundaries of the zones should be indicated by a line. How were the boundaries determined?

According to the reviewer's suggestion, we have added DNA staining images to **Figs. 2A–D, 3A, 5A, 5B, Appendix Figs. S3F, S3G, S3M, S6K**.

It is difficult to identify the boundary between the SVZ and VZ in DAPI-stained images, so we have not distinguished between them and written them as VZ/SVZ in the images. The boundaries between VZ/SVZ and IZ are shown on the left of the images.

5) The authors have generated a new reporter line and state that "EGFP was exclusively expressed in TBR2-immunopositive cells" (Fig 4f). This is not supported by any quantification. The GFP-positive cells should be characterized with respect to Sox2/Pax6, Tbr2 and neuron marker expression.

In the revised manuscript, we have quantitatively evaluated EGFP-expressing cells by performing multiple staining of those markers (TBR2, SOX2, and HuC/D) with GFP in *Eomes*^{T2A-d2EGFP/+} mice at E14.5 (**Appendix Fig. S4F, G**). Since 94% of EGFP+ cells were TBR2-positive, we have changed the description in Results as below.

“Immunostaining with GFP, TBR2, SOX2, and HuC/D showed that most EGFP signals ($93.6 \pm 1.2\%$; mean \pm SEM) were specifically detected in TBR2-immunopositive cells (Appendix Fig. S4F, G), suggesting that most IPCs were specifically labeled with EGFP without interfering with TBR2 protein expression and function in KI mice.”

6) For the isolation of RGCs, CD133 expression was used. However, the population that was gated only included the most highly CD133 expressing cells, which represents a tiny fraction and does not match the expected numbers of RGC at E15.5 in the mouse. Even though the RGC fraction expresses canonical RGC marker genes, it is not clear what this sub-population represents and therefore the RGC transcriptome data should be treated with caution.

The gate setting for CD133 in the original manuscript (CD133-high, red rectangle in Figure A left panel), based on the paper by Morimoto-Suzuki et al., *Development* 141, 4343-4353, 2014. This paper states that the cells in the high CD133 intensity area are a more proliferative NPC population and that the mid area contains many differentiating NPCs. As a trial, we also tried widening the gate a little (CD133-wide, red rectangle in Figure A right panel).

We collected three batches of CD133-high, CD133-wide, and EGFP+ cells and performed RNAseq on each. Fig. B shows a graph of each batch's principal component analysis (PCA) of the RNAseq data.

As expected, all three batches of EGFP+ cells were plotted far apart from the other CD133-high or CD133-low batches. This was thought to reflect the large differences in the nature of RGCs and IPCs. In contrast, the three batches of CD133-high cells and the

three batches of CD133-wide cells did not separate but were mixed (Figure B). This suggested that there was not a large difference in the RNA expression trends between CD133-high cells and CD133-wide cells. Therefore, we have adopted the original gate condition in the revised manuscript.

We are very grateful for the reviewer's comment, which helped us gain a deeper understanding of the FACS-sorted cells.

7) The authors focus on the *Robo1* gene as a candidate gene among IPC-upregulated genes. The extensive work by Victor Borrell's group on the role of Robo-signaling in cortical neurogenesis and IPC regulation should be mentioned when introducing the gene (Borrell et al., Neuron 2012; Cardenas, Cell 2018).

We have changed the description in Results below, assessing the previous works regarding *Robo1*.

“As we observed a significant phenotype in IPCs, we concentrated on analyzing IPCs in this study. Among IPC-upregulated and IPC-downregulated genes, the *Robo1* gene was investigated. The *Robo1* gene was linked to the GO terms “nervous system development” and “negative regulation of cell proliferation” (Fig. 4G). There have already been reports on the roles of ROBO signaling in RGCs. This signaling promotes direct neurogenesis from RGCs and suppresses indirect neurogenesis, or the production of IPCs (Borrell *et al*, 2012; Cardenas *et al*, 2018). However, the role of ROBO signaling in IPCs is still poorly understood. We found that expression of this gene increased approximately 2.5-fold in *Auts2* cKO EGFP⁺ cells compared with that in control (Fig. 4H). Therefore, we suspected that this overexpressed *Robo1* might affect the division of IPCs and the production of neurons.”

8) The authors conclude that "These findings suggest that AUTS2 promotes the proliferation of IPCs and the production of an appropriate number of neurons by repressing Robo1 expression." Can that be concluded? In Figures 2, 3 and 5, different analyses were performed. Cell cycle length is increased. How does this fit in with the regulation of proliferation? And with previous data on different cell cycle length in proliferative vs neurogenic NPCs?

As the reviewer mentioned, we have to admit that we shouldn't have related cell cycle length to the proliferative vs. neurogenic states of IPCs in the previous manuscript. Instead, we should have related the G1 length to these states.

It was previously reported that the cell cycle length was shorter in neurogenic NPCs compared to proliferative NPCs in certain experimental conditions (Arai et al., *Nat.*

Commun. 2, 154, 2011). However, even in their experimental conditions, G1-length was longer in neurogenic NPCs compared to proliferative NPCs, while S-phase was shorter in neurogenic NPCs. Furthermore, many papers have reported that G1-length tends to be longer in neurogenic progenitors and shorter in proliferative progenitors (Takahashi et al., *J. Neurosci.* 15, 6046-6057, 1995; Calegari et al., *J. Neurosci.* 25, 6533-6538, 2005; Lange et al., *Cell stem cell* 5, 320-331, 2009; Pilaz et al, *PNAS* 106, 21924-21929, 2009). This was assessed in the Results and Discussion of the revised manuscript.

In the revised manuscript, the G1, G2, and M-phase lengths of TBR+ cells were estimated and shown in **Fig. 2H, I**. The length of the G1 phase was found to be significantly longer in heterozygous and homozygous cells than in WT cells. Therefore, we thought that in KO mice, the increase in neurogenic IPCs and the decrease in the division of total IPCs were somewhat related to the prolongation of the G1 phase.

In addition, the data below (1-4) suggest that “AUTS2 regulates the proliferation of IPCs to produce the appropriate number of upper-layer neurons by suppressing *Robo1* expression” even in the absence of cell cycle results.

- 1 *Robo1* overexpression by electroporation reduced the PH3+ IPCs in embryos (Fig. 5A).
- 2 In *Auts2* KO mice, *Robo1* expression was increased in IPCs (Fig. 4H). PH3+ IPCs were decreased (Fig. 2B).
- 3 The introduction of the *Robo1* knockdown vector restored the reduction of PH3+ IPCs in *Auts2* KO mice (Fig. 5C).
- 4 The number of upper-layer neurons was reduced in *Auts2* KO mice (Fig. 1D).

We have assessed the cell cycle analyses as below in the revised manuscript.

In Results,

“We also estimated the length of other cell-cycle phases (Fig. 2I) (see Methods). G1-lengths for *Auts2*^{del(ex8)/+} and *Auts2*^{del(ex8)/del(ex8)} were much longer than that for WT (Fig. 2I). Since a large population of TBR2⁺ cells was IPCs (Appendix Fig. S3D), and since the number of IPCs (TBR2⁺ and KI67⁺ cells) was not affected in *Auts2* mutants (Appendix Fig. S3C, D), it was suggested that the cell cycle length and G1-length of IPCs were longer in *Auts2* mutants than those in WT mice. Previously, there have been many reports that G1-length tends to be longer in neurogenic progenitors and shorter in proliferative progenitors (Takahashi *et al*, 1995; Calegari *et al*, 2005; Lange *et al*, 2009; Pilaz *et al*, 2009). Therefore, the elongated G1-phase in *Auts2* mutant IPCs may be involved in the reduced proliferation of IPCs and the subsequent production of upper-layer neurons.”

In legend for Fig. 2G–I

“(G, H) The cell-cycle length estimate in TBR2⁺ cells using the BrdU and EdU double-labeling method at E15.5. BrdU and EdU are administered to pregnant mice 2 h and 30 min before sacrifice. T_S, S-phase length; T_C, cell cycle length; L_{cells}, BrdU⁺, and EdU-negative cells; S_{cells}, EdU⁺ cells; P_{cells}, TBR2⁺ cells. (H) The graph shows the percentage of BrdU⁺ cells among PH3⁺ and TBR2⁺ cells at E15.5.

(I) Cell-cycle parameters of TBR2⁺ cells in E15.5 cortex. T_{G2}: G2-phase length, T_M: M-phase, T_{G1}: G1-phase.”

In Methods for Calculation of cell cycle length

“Based on previous methods (Watanabe *et al*, 2015; Martynoga *et al*, 2005), the cell cycle length was calculated using BrdU and EdU. BrdU (50 mg/kg) and EdU (30 mg/kg) were administered intraperitoneally to pregnant mice 2 h and 30 min before sacrifice, respectively. The lengths of the S-phase (T_S) and total cell cycle (T_C) were calculated as follows: $T_S = 1.5 \times S_{\text{cells}} / L_{\text{cells}}$ and $T_C = T_S \times P_{\text{cells}} / S_{\text{cells}}$ [L_{cells}=cells leaving S-phase (identified as TBR2⁺, BrdU⁺, EdU⁻ cells); S_{cells}=cells in S-phase (TBR2⁺, EdU⁺ cells); P_{cells}= total proliferating cells (TBR2⁺ cells)]. The length of M-phase (T_M) was calculated as follows: $T_M = T_C \times M_{\text{cells}} / P_{\text{cells}}$ [M_{cells}=Average value of PH3/TBR2-double positive cells (Fig. 2B); P_{cells}=Average value of TBR2⁺ cells (Appendix Fig. S3C)].”

9) Related to Figure 6e, which cluster contains the differentially expressed genes?

Given the reviewer’s suggestion, we counted the genes with “IPC-upregulated TSSs” in each cluster and added the description below in the legend of Fig. 6E.

“15 genes with IPC-upregulated-TSSs (see Fig. 7A) were included in 558 loci of cluster 1. 15 and 4 genes with IPC-upregulated-TSSs were in 2340 and 2566 loci of cluster 2 and 3, respectively.”

As expected, of the genes in each cluster, the probability of gene expression being affected in *Auts2* cKO was highest in cluster 1 (15/558=2.69%). In contrast, only 0.64% (15/2340=0.64%) of the genes in cluster 2 had altered gene expression in *Auts2* cKO.

10) Do Ring1b, H2AK119ub and H3K27me3 levels change when all *Auts2*-bound genes are interrogated (rather than only differentially expressed genes)?

In the original manuscript, we also described the levels of Ring1b, H2AK119ub, and H3K27me3 for genes with no change in expression (Extended Fig. 8 c, d, f – Unchanged-TSSs) in addition to data with upregulated DEGs (Fig. 7c, Extended Fig. 8c–e – Upregulated TSS). Neither Ring1b (Fig. S8C), H2AK119ub (Fig. S8D), nor H3K27me3 (Fig. S8F) changed their levels on the genes with no change in expression (Unchanged-TSSs). We believe we are meeting the reviewer's request because we have provided data for both types of TSSs of the genes (Upregulated and Unchanged). However, if the reviewer requests the merged results, we will include them.

11) How does the data related to PRC2 integrate with what is already known on the role of PRC2 and H3K27me3 in cortical development (Pereira, PNAS 2010; Albert, EMBO J 2017; Telley, Science 2019)? These publications should be discussed.

Given the reviewer's valuable suggestion, we have added the following sentences in Discussion.

“There have been reports on the role of PRC2 in cortical development. In the KO mice for essential components of the PRC2 complex, *Ezh2* and *Eed*, the balance between self-renewal and differentiation in NPCs shifted towards differentiation, resulting in the reduced number of neuron production and microcephaly-like phenotype (Pereira *et al*, 2010; Telley *et al*, 2019). Many scientists believe these phenotypes were caused by the loss of function of PRC2 in RGCs, albeit lacking direct evidence. In this study, we specifically disrupted the function of PRC2 in IPCs by specific induction of EZH2 dominant negative form (Appendix Fig. S9E), suggesting that PRC2 function in IPCs is involved in IPC division and appropriate production of upper-layer neurons. As PRC2 complexes are expressed in both RGCs and IPCs, PRC2 may play important roles in each population during corticogenesis. Previously, a comprehensive genomic analysis was conducted to investigate the H3K27me3 modification status in neuroepithelial cells at E9.5 and apical RGCs, oRGCs, and neurons at E14.5 (Albert *et al*, 2017). The researchers reported that the genes modified by H3K27me3 changed dynamically in each cell type, suggesting that the target genes of PRC2 change depending on the cell type and developmental stage.”

12) The genomics data was not accessible to the reviewer.

We apologize for the inconvenience.

The data uploaded to the DDBJ (PRJDB17034 and PRJDB17038) has been released.

Referee #2

The study investigates the mechanisms underlying changes in cortical thickness in mouse models of the AUTS2 syndrome. In humans the syndrome is associated with mutations of the AUTS2 gene causing a wide range of phenotypes including intellectual disability, autism epilepsy and depression. The analysis of the gene function is complicated by the size of the gene and the presence of different length are expressed at different time and in different regions during development. More than 60% of the patients also show microcephaly and frequent alteration include loss of volume in the cortex and hypoplasia in the cerebellum and corpus callosum. Hypoplasia has been also observed in mice models in the cerebellum and in the dentate gyrus. A reduced progenitor proliferation has also been observed in brain organoids. However, loss of cortical volume and change in proliferation has not been reproduced in a mouse model. The main novelty of the paper resides in the reproduction of the microcephalic phenotype in mutant mice. The authors conclude that this is associated to a change in the proliferation of IPCs and not RGCs. Mechanistically, the authors propose that mutation of the gene leads to overexpression of Robo1, which in turns affects Cell proliferation. Whereas this is a novel finding, it is in apparent contradiction with a recent study showing that patient derived organoids display altered proliferation of RGCs and not IPCs.

The possible reasons underlying this discrepancy should be at least discussed.

We thank the reviewer's valuable suggestion. We agree that it is essential. The revised manuscript discusses the discrepancy in depth and detail, and ample space is devoted to it.

Firstly, we consider the human organoid and mouse models to be excellent and valuable disease model systems. Both systems have advantages and disadvantages, as noted below. Therefore, to understand the pathogenesis of the disease or the physiological function of AUTS2, we believe it is better to integrate and discuss the analysis results of the two systems.

The beauty of the mouse model is that it can be analyzed at the animal level. Conversely, in the case of cultured cells or organoids, the results are inevitably affected by changes in culture and experimental conditions. Therefore, organoids do not necessarily reproduce the exact same conditions in the brain as in vivo.

A drawback of the mouse model is that while brain development and function are similar between mice and humans, they are not always identical. For example, mouse brains

contain only a limited number of outer radial glial cells (oRGCs) that are abundant in humans. If the pathology is caused by abnormalities in oRGCs, analysis of the mouse model may not provide good insight into the nature of the pathology. However, organoids from human cells allow us to observe brain development that is unique to humans.

Thus, since both the human organoid model and the mouse model have pros and cons, we think both studies are valuable. By considering the results from both, we can get to the essence of the pathology.

In addition, in our original manuscript, we focused on AUTS2's functions in IPCs. However, it does not mean that AUTS2 does not play an important role in RGCs. Furthermore, dysfunction of AUTS2 in RGCs may also be involved in the pathogenesis of the AUTS2 syndrome. In fact, in *Auts2* cKO mice, we observed a thinning of the deep layer at the rostral cerebral cortex (Fig. 1E in the revised and original manuscript). This implies that AUTS2 plays some role in the function of RGCs.

However, we must admit that the original manuscript could have been better written and was misleading. It gave the impression that AUTS2 only works in IPCs and not in RGCs. In the revised manuscript, we have assessed the possibility that AUTS2 may also work in RGCs, which may be related to microcephaly.

Considering these, we have added description below in Discussion.

“In addition to investigating disease animal models, organoids made from human induced pluripotent stem cells (iPSCs) are also powerful tools for understanding the pathology of diseases. A previous study demonstrated that a patient with the *AUTS2*^{T534P} missense mutation displayed reduced cortical area and ventriculomegaly on MRI. In the organoids generated from the patient (*AUTS2*^{T534P})-derived iPSCs, the division of neural progenitor cells, presumably apical RGCs, was reduced compared to controls during the early neurogenic phase, resulting in impaired organoid growth (Fair *et al*, 2023). This suggests that abnormalities in RGCs may also be involved in microcephaly in AUTS2 syndrome. Although we could not detect abnormalities in RGC division in our mouse model, a decrease in the thickness of the deep layer was detected in the rostral cortex of those mice (Fig. 1E), implying some functional abnormality in the RGCs. This might be supported by the result that hundreds of DEGs in RGCs were identified between control and the *Auts2* cKO (*Emx1*^{Cre/+}; *Auts2*^{fl(ex8)/fl(ex8)}) mice (Appendix Fig. S5B). As mentioned above, there are various mutation types in AUTS2 syndrome, and each mutation type may cause different molecular functional abnormalities and phenotypes. The *AUTS2*^{T534P} used in the organoid research was a missense mutation in the HX repeat motif (Histone

acetyltransferase P300-binding site) of the AUTS2 protein. This AUTS2 mutant protein cannot bind to P300, which may alter the transcription of genes. Since the functional changes in the *AUTS2*^{T534P} protein differ from those in our mouse model (*Emx1*^{Cre/+}; *Auts2*^{fl(ex8)/fl(ex8)}), in which expression of FL-AUTS2 and S-AUTS2-var.1 is lost and that of S-AUTS2-var.2 is increased (Appendix Fig. S1B), the microcephaly phenotype may be expressed through slightly different mechanisms in each pathological model.”

Moreover, lack of direct manipulation of gene expression in IPCs, of investigation of residual AUTS2 expression and analysis of Robo1 changes in heterozygous mice impact the strength of the conclusions.

1) The study is very confusing due to the fact that different experiments are done on heterozygous or homozygous KO, which show microcephaly, or animals lacking only the terminal part of the protein, which have similar brain size.

The reason for using this latter model in some experiments, see also below, is not clear. Differences in the pattern of AUTS2 expression in the various models are never investigated in RGC and IPCs only reference is made to previous studies but due to the complexity of the system and lack of analysis in heterozygous mice I feel this issue should be addressed.

I apologize for the confusing and misleading explanation of these different mouse models in the original manuscript. The genomic changes in the knocked-out cells are identical in the two mouse models. The only difference is whether it is a systemic conventional KO or a conditional KO. Therefore, since the genomic changes are the same in the knocked-out cells, the changes in protein expression are also expected to be the same.

In the original manuscript, we used two different names for the alleles of the *Auts2* gene (*Auts2*^{fllox} and *Auts2*^{del8}). However, several different types of *Auts2*^{fllox} alleles have been reported from other groups. The notation for alleles has been changed in the revised manuscript to avoid confusion.

(i) *Auts2*^{fllox} in the previous manuscript is denoted as *Auts2*^{fl(ex8)} in the revised manuscript
(ii) *Auts2*^{del8} in the previous manuscript is denoted as *Auts2*^{del(ex8)} in the revised manuscript.

In the *Auts2*^{fl(ex8)} allele, two loxP sequences are inserted on either side of the exon8 of the *Auts2* gene. Therefore, in *Emx1*^{Cre/+}; *Auts2*^{fl(ex8)/fl(ex8)} individuals, exon8 of the *Auts2* gene is removed forebrain-specifically (Hori et al., 2014, *Cell Reports*, 9, 2166-2179).

Regarding *Auts2*^{del(ex8)}:

When *CAG-Cre* transgenic mice, which systemically express Cre recombinase, were

crossed with $Auts2^{fl(ex8)/+}$, exon8 was removed in the germ-line (sperm) of the individuals born from the cross ($Auts2^{fl(ex8)/+}; CAG-Cre$). Then, in further descendants of that individual, heterozygotes for the allele ($Auts2^{del(ex8)/+}$) were born with a systemic loss of exon8. The $Auts2^{del(ex8)}$ allele was created this way (Hori et al., 2014, *Cell Reports*, 9, 2166-2179). Unfortunately, $Auts2^{del(ex8)}$ homozygotes ($Auts2^{del(ex8)/del(ex8)}$) die just after birth. Therefore, in this study, we basically use the $Auts2^{del(ex8)}$ allele, a conventional KO, for analysis at embryonic stages, and the $Auts2^{fl(ex8)}; Emx1^{Cre}$, a conditional KO, for postnatal analysis.

In both types of KO mice, the changes in protein expression in the knocked-out cells are thought to be identical because the knocked-out cells have the same genomic mutation, that is, the loss of the exon8. Eventually, FL (full-length)-AUTS2 and S-AUTS2-var.1 are entirely lost, while S-AUTS2-var.2 expression is increased, as was shown in the previous paper (Hori et al, 2014, *Cell Reports*, 9, 2166-2179).

To assess this, we have added the following sentences in Results.

“Previously, we generated a floxed allele of the *Auts2* gene, which carried two loxP sequences on either side of the exon 8, and named it $Auts2^{lox}$. By crossing mice carrying this allele with mice with the *CAG-Cre-Tg* allele that ubiquitously expressed Cre (Sakai & Miyazaki, 1997), we successfully obtained the allele that lacked the exon 8, which we named $Auts2^{del8}$ (Hori et al, 2014). However, as several research groups have reported different types of floxed alleles for this gene (Gao et al, 2014; Castanza et al, 2021; Li et al, 2022), we have decided to rename $Auts2^{lox}$ and $Auts2^{del8}$ as $Auts2^{fl(ex8)}$ and $Auts2^{del(ex8)}$, respectively (Appendix Fig. S1A). Since homozygotes for $Auts2^{del(ex8)}$ ($Auts2^{del(ex8)/del(ex8)}$) were postnatally lethal, we produced forebrain-specific *Auts2* conditional knockout (*Auts2* cKO) homozygous and heterozygous mice ($Emx1^{Cre/+}; Auts2^{fl(ex8)/fl(ex8)}$, $Emx1^{Cre/+}; Auts2^{fl(ex8)/+}$) by utilizing the $Emx1^{Cre}$ allele (Hori et al, 2020). We basically use global *Auts2* KO ($Auts2^{del(ex8)/del(ex8)}$, $Auts2^{del(ex8)/+}$) and forebrain specific cKO ($Emx1^{Cre/+}; Auts2^{fl(ex8)/fl(ex8)}$, $Emx1^{Cre/+}; Auts2^{fl(ex8)/+}$) for analyses at embryonic and postnatal stages, respectively.”

We have added **Appendix Fig. S1A** to explain the genomic structure of each allele. The AUTS2 protein produced from each allele is shown in **Appendix Fig. S1B**.

The above will help readers understand the two alleles and eliminate confusing and misleading elements.

2) A main finding in the paper is the cortical volume reduction which in mice seems to be associated to changes in cell cycle progression of IPCs but not RGCs. A recent study investigating the effect of AUTS2 in organoids reached recently opposite

conclusions indicating that the gene behaves differently in humans and mice.

Is the deletion of the protein similarly efficient in the two populations of cells?

We previously performed RNA sequencing on RGCs and IPCs collected by FACS from control and *Auts2* mutants (*Auts2*^{fl(ex8)/fl(ex8)}; *Emx1*^{Cre/+}; *Eomes*^{T2A-d2EGFP/+}). As a result, we found that exon8 was excluded in both cell types. If the reviewer requests it, the data will be included in the manuscript.

As described above, we have discussed the discrepancy between the results in the previous organoid study and our mouse model study.

The present study shows that, at least in mice, AUTS2 is involved in the proliferation of IPCs and the development of upper-layer neurons, which we believe is also a novel finding. In addition, we have added the following sentences that state the possibility that dysfunction of RGCs is also involved in the pathology of microcephaly, as was described above.

“A previous study demonstrated that a patient with the *AUTS2*^{T534P} missense mutation displayed reduced cortical area and ventriculomegaly on MRI. In the organoids generated from the patient (*AUTS2*^{T534P})-derived iPSCs, the division of neural progenitor cells, presumably apical RGCs, was reduced compared to controls during the early neurogenic phase, resulting in impaired organoid growth (Fair *et al*, 2023). This suggests that abnormalities in RGCs may also be involved in microcephaly in AUTS2 syndrome. Although we could not detect abnormalities in RGC division in our mouse model, a decrease in the thickness of the deep layer was detected in the rostral cortex of those mice (Fig. 1E), implying some functional abnormality in the RGCs. This might be supported by the result that hundreds of DEGs in RGCs were identified between control and the *Auts2* cKO (*Emx1*^{Cre/+}; *Auts2*^{fl(ex8)/fl(ex8)}) mice (Appendix Fig. S5B).”

What about expression? The authors provide evidence that AUTS2 is similarly expressed at E13.5 in the two populations. However, most of the experiments to monitor apical progenitor populations are done at E12.5.

Given the reviewer’s question, we have analyzed the expression of the *Auts2* gene at E12.5 and E15.5 using another dataset of scRNA-seq of the developing cerebral cortex (Di Bella *et al.*, *Nature* 595, 554-559, 2021). We have added it to the revised manuscript (**Appendix Fig. S2G, H**). We found *Auts2* was expressed in RGCs and IPCs to a significant extent at E12.5, just as it was at E13.5.

3) Figure1 the authors conclude that early generated neurons at E12.5 are not

affected by *Auts2* ablation and interpret this as evidence that *Auts2* affects primarily IPC behavior. However, would be the lack of an effect at E12.5 predictable due to the fact that at this early age *Auts2* protein is hardly expressed in the cortex?

As mentioned above and shown in newly added **Appendix Fig. S2G**, *Auts2* is expressed in RGCs and IPCs even at E12.5. In addition, FL-AUTS2 protein was detected in the brain lysate at E12.5 brain by the Western blotting in our previous report (Hori et al., *Cell Reports* 9, 2166-79, 2014).

Moreover, in figure 1 C the thickness of the deep layer appears affected at E15, also shown in the quantification, but not in postnatal animals (Fig. 1f). Does this suggest a change in differentiation or migration of the early born neurons?

Figure 1C in the original manuscript showed P (postnatal day) 15, but not E15. We are sorry, but this notation “P15” was likely to cause misunderstanding among readers. Therefore, we have changed the notation from P (e.g., P15) to PD (e.g., PD15) in the revised manuscript.

4) Figure 2 and extended figure 3 it is unclear why these analyses are performed in a mouse model that does not show microcephaly. How to reconcile the apparent change in proliferation of IPCs cells with the lack of a microcephalic phenotype and the overall conclusion that a change in IPC proliferation leads to microcephaly? Indeed, these results suggest that when the change in proliferation is limited to TRB2+ cells there is no change in cortical volume.

In Figure 2 and Figure S3, we are using *Auts2*^{del(ex8)/del(ex8)} mutants. As described above, use these (conventional KO) mutants to analyze at embryonic stages. As *Auts2*^{del(ex8)/del(ex8)} die just after birth, we do not know whether they would show microcephaly if they survived after birth. However, because *Auts2*^{del(ex8)/del(ex8)} have the same genomic mutation as *Auts2*^{fl(ex8)/fl(ex8)}; *Emx1*^{Cre} in the telencephalon, we presume that they would present microcephaly if they survived until adults.

However, we agree that genotype notation could have been more appropriate and could easily lead to misunderstanding in the previous manuscript. Therefore, we have changed the notation as described above.

5) More on figure 2 and other cellular quantification. Were the number of cells counted? In many of the figures, including figure 2, although the number of the cells appears not affected the distribution of the cells along the apical/basal axis, i.d. the distance of the cell from the lateral ventricle, appears to be altered. This is particularly visible in Trb2+/Ki67+ cells. A count of the cells on bins parallel to the

lateral ventricle should be performed

Yes, the number of cells was counted in the original manuscript.

According to the reviewer's suggestion, we have estimated the distribution of the TBR2+/KI67+ cells along the apical/basal axis on bins parallel to the lateral ventricle (**Appendix Fig. S3E**) in the revised manuscript. No significant differences were observed between WT, heterozygotes, and homozygotes.

6) In figure 3 are the changes in Ki67+ cells obtained after the various manipulations happening only in RGCs, also in IPCs or in both cell groups?

Thank you for your valuable comment.

In Fig. 3 of the previous manuscript, the experiment aimed to evaluate how many neurons were produced two days after the EP of control or shRNA for *Auts2*. However, we did not perform immunostaining with a neuron marker in the previous experiment. Instead, we counted Ki67-negative cells, deducing that they would represent neuronal cells.

In the revised manuscript, we have improved the experimental design using a neuronal marker (HuC/D). We directly counted postmitotic neurons (HuC/D-positive, Ki67-negative cells) in the electroporated (GFP+) cells (Fig. 3A, B), obtaining identical results as in the previous experiment.

According to the reviewer's suggestion, we also evaluated PAX6+ cells (RGCs) and TBR2+, KI67+ cells (IPCs) in all the electroporated cells (GFP+ cells) (**Fig. 3D, E**). The results were not significantly affected by any of the shRNA vectors electroporated.

7) In figure 4 gate settings leads to sorting of only a small proportion of Cd133+/GFP- cells and to include a consistent proportion of CD133+ cells in the GFP+ populations, which according to immunostaining shown in extended data Figure4F may represent GFP+ cells in the VZ not expressing TRB2 protein. Consistent with this the difference in Prom1 expression between the two sorted populations is minimal. Is there a reason for this particular choice? How were the sorting gates set?

The gate setting for CD133 was based on the paper by Morimoto-Suzki *et al.*, *Development* 141, 4343-4353, 2014. This paper (Morimoto-Suzki *et al.*) states that the cells in the high CD133 intensity area are a more proliferative NPC population and that the mid area contains many differentiating NPCs. As a trial, we also performed widening the gate a little, but similar cell groups were gathered. So, we have adopted the original gate condition.

Please add images of sorting plots from mutant mice.

We have added an image of the sorting plot of the mutant mice in the revised manuscript (**Appendix Fig. S4H**).

Could this sorting setting explain the apparent contradiction that in these experiments *Auts2* gene is more expressed in GFP+ cells than in CD133 cells whereas in extended data figure 3 *Auts2* transcripts appear to be similarly expressed in the two populations?

Regarding the scRNAseq data shown in Fig. S2, the original manuscript did not mention the difference in the expression levels of *Auts2* in RGCs and IPCs. Then, we re-analyzed the data and found that the expression of *Auts2* was significantly higher in IPCs than in RGCs at E15.5 ($P_{adj}=1.99E-05$). In addition, analysis using the newly added E15.5 scRNA-seq data (**Appendix Fig. S2H**) also showed that *Auts2* expression was significantly higher in IPCs than in RGCs ($P_{adj}=2.04E-84$).

We have assessed this issue in the revised manuscript as below.

In Results.

“The expression of *Auts2* was significantly higher in IPCs than in RGCs at E12.5, E13.5 and E15.5 (see the legend for Appendix Fig. S2).”

In the legend for Fig. S2,

“The expression of *Auts2* was significantly higher in IPCs than in RGCs at E12.5 ($P_{adj}=2.59E-63$ (G)), E13.5 ($P_{adj}=5.50E-07$ (C)), and E15.5 ($P_{adj}=1.99E-05$ (F), $P_{adj}=2.04E-84$ (H))”

The sorting of a specific subpopulation of CD133+ cells may also explain the fact that gene ontology analysis of the differentially regulated genes in this population, albeit involving more genes than in the putative IPCs, highlights only the angiogenesis process.

The above results were obtained using cells sorted according to the conditions of past literature (Morimoto-Suzuki *et al.*, *Development* 141, 4343-4353, 2014). We are not attempting to say in this manuscript that *Auts2* is not functioning in RGCs, but that it is probably functioning in some way in IPCs. Concerning microcephaly, we do not deny the involvement of the loss of function of *Auts2* in RGCs. On the contrary, in the brains of *Auts2* cKO (*Emx1*^{Cre/+}; *Auts2*^{fl(ex8)/fl(ex8)}) mice, the deep layer was observed to be slightly thinner on the rostral side (Fig. 1E). Therefore, we think that the loss of function of *Auts2* in RGCs may also contribute to microcephaly. Again, this was assessed in Discussion of the revised manuscript as follows.

“In addition to investigating disease animal models, organoids made from human induced pluripotent stem cells (iPSCs) are also powerful tools for understanding the pathology of diseases. A previous study demonstrated that a patient with the *AUTS2*^{T534P} missense mutation displayed reduced cortical area and ventriculomegaly on MRI. In the organoids generated from the patient (*AUTS2*^{T534P})-derived iPSCs, the division of neural progenitor cells, presumably apical RGCs, was reduced compared to controls during the early neurogenic phase, resulting in impaired organoid growth (Fair *et al*, 2023). This suggests that abnormalities in RGCs may also be involved in microcephaly in *AUTS2* syndrome. Although we could not detect abnormalities in RGC division in our mouse model, a decrease in the thickness of the deep layer was detected in the rostral cortex of those mice (Fig. 1E), implying some functional abnormality in the RGCs. This might be supported by the result that hundreds of DEGs in RGCs were identified between control and the *Auts2* cKO (*Emx1*^{Cre/+}; *Auts2*^{fl(ex8)/fl(ex8)}) mice (Appendix Fig. S5B). As mentioned above, there are various mutation types in *AUTS2* syndrome, and each mutation type may cause different molecular functional abnormalities and phenotypes. The *AUTS2*^{T534P} used in the organoid research was a missense mutation in the HX repeat motif (Histone acetyltransferase P300-binding site) of the *AUTS2* protein. This *AUTS2* mutant protein cannot bind to P300, which may alter the transcription of genes. Since the functional changes in the *AUTS2*^{T534P} protein differ from those in our mouse model (*Emx1*^{Cre/+}; *Auts2*^{fl(ex8)/fl(ex8)}), in which expression of FL-*AUTS2* and S-*AUTS2*-var.1 is lost and that of S-*AUTS2*-var.2 is increased (Appendix Fig. S1B), the microcephaly phenotype may be expressed through slightly different mechanisms in each pathological model.”

8) IPCs are highly proliferating cells. However, Ki67 expression is higher in the FACS isolated RGCs than in GFP+ putative IPCs. What is the percentage of isolated cells that is KI67 positive?

Given the reviewer’s question, we have immunostained EGFP+ cells or CD133^{high} cells using the KI67 antibody and examined the percentage of KI67-positive cells in each cell group (Appendix Fig. S4J). We found that 99% of CD133^{high} cells and 92% of EGFP+ cells were KI67-positive proliferating cells (graph in Appendix Fig. S4J).

9) Figure 5

Does OE of Robo1 affects number of apically dividing TBR2- progenitors? Does the manipulation alter the distribution of TRB2 expression or of the electroporated cells? From the figure shown there appears more TRB2+ cells close to the apical side of the VZ and that less green cells are present in the VZ.

To answer the reviewer's question, "Does OE of Robo1 affect the number of apically dividing TBR2- progenitors?" we examined the proportion of PH3-positive cells in GFP-positive/TBR2-negative cells localized to the VZ (**Appendix Fig. S6B**). As a result, there was no significant difference in the proportion of apical dividing cells between control and *Robo1* overexpression.

To answer the reviewer's question, "Does the manipulation alter the distribution of TBR2 expression or of the electroporated cells?" we have evaluated the distribution of TBR2-positive cells in bins parallel to the ventricular surface (**Appendix Fig. S6C**). As a result, there was no significant difference in the distribution between control and *Robo1* overexpression.

We apologize that the images in the original manuscript were inappropriate and misleading. They were not necessarily representative images. Therefore, we have replaced the images with appropriate and representative ones (Fig 5A).

10) It has been shown that Robo1/ slit signaling is required to maintain ventricular mitosis between E12.5 and E15.5 therefore manipulation of Robo1 expression should be performed specifically in IPCs to monitor the effect in this population.

As suggested by the reviewer, we first electroporated a vector designed to express *Robo1* under the control of the *Tbr2*-promoter. However, since *Robo1* expression was also observed in RGCs, we abandoned this method.

Next, we generated the *Eomes*^{T2A-iCre} knock-in (KI) allele (**Appendix Fig. S6D, E**) using the same method as that used to create the *Eomes*^{T2A-d2EGFP} knock-in (KI) allele (Appendix Fig. S4A). In mice carrying this allele, iCre (improved Cre) was expected to be expressed in cells expressing *Eomes* (*TBR2*). We first electroporated the pCAG-LSL(loxP-Stop-loxP)-H3GFP expression vector plus pCAG-RFP vector into the VZ of *Eomes*^{T2A-iCre/+} mice at E14.5. At one day after electroporation, H3GFP signals were detected in 90% of the electroporated TBR2+ cells (TBR2+, RFP+) but hardly observed in TBR2-negative cells located to the VZ (**Appendix Fig. S6F, G**). This suggests that the method using *Eomes*^{T2A-iCre/+} and pCAG-LSL vector works well in overexpression of a certain gene specifically in IPCs.

Next, we co-electroporated pCAG-LSL-*Robo1* and pCAG-H3GFP vectors into *Eomes*^{T2A-iCre/+} KI embryos at E13.5 and examined the percentage of PH3-positive cells in TBR2+/GFP+ cells at two days after electroporation (**Appendix Fig. S6H**). Compared to the control, we found that the percentage of PH3-positive cells decreased in cells that had been introduced with pCAG-LSL-*Robo1*, suggesting that specific overexpression of

Robo1 in TBR2-positive cells suppressed divisions of the introduced cells.

Moreover, it has been shown that the Robo1/ slit signaling interacts with notch to affect proliferation. Therefore, cell non-autonomous mechanisms cannot be excluded. Does the manipulation change the number of proliferating cells in the non-transfected population?

Given the reviewer's comment, we estimated the effect of *Robo1* overexpression in electroporated (GFP+) and non-electroporated (GFP-) cells. At the *Robo1*-electroporated brain region, the ratio of PH3-positive cells was significantly reduced in TBR2+/GFP+ cells but not in TBR2+/GFP- cells (**Fig. 5A**). This suggests that ROBO1 functions cell autonomously but not in a cell non-autonomous manner.

11) Are levels of Robo protein also changed in *Auts2* KO homo and heterozygous?

To evaluate ROBO1 protein levels, we first tried to perform a Western blot analysis on the FACS-sorted EGFP-positive cells from E15.5 brains of control, heterozygotes or homozygotes (*Eomes*^{T2A-d2EGFP/+}; *Auts2*^{fl(ex8)/+}, *Eomes*^{T2A-d2EGFP/+}; *Emx1*^{Cre/+}; *Auts2*^{fl(ex8)/+} or *Auts2*^{fl(ex8)/fl(ex8)}). However, because the number of cells obtained from a single individual was minimal, it was not possible to detect ROBO1 protein by Western blotting.

Then, we performed cell immunocytochemistry for ROBO1 to the sorted EGFP+ cells and estimated the intensity of ROBO1. As a result, we found that the immunofluorescent intensities of ROBO1 were significantly stronger in heterozygous and homozygous EGFP+ cells than in control cells (**Appendix Fig. S6A**), consistent with the result of the RT-qPCR experiment (Fig. 4H).

Why to focus on Robo1 and not also on its ligand Slit1 whose expression, both in RGC and IPCs, is even more affected by the mutation than Robo1?

We performed the overexpression experiment according to the valuable suggestion (**Appendix Fig. S6I**). We found that *Slit1* overexpression also reduced the proportion of PH3+ cells in TBR2+/GFP+ cells, suggesting that upregulated SLIT1-ROBO1 signaling may downregulate IPC division.

12) What is the profile of *Auts2* binding in Cut and Tag analysis of IPCs isolated from heterozygous mutant mice? Are repressive marks altered at the *Robo1* locus also in this case?

Given the reviewer's suggestion, we have performed the CUT&Tag experiment for FACS-sorted cells from heterozygous mutant mice. We found that AUTS2-binding

signals were reduced in heterozygotes, as was observed in homozygotes (**Appendix Fig. S7A**).

The repressive mark (H3K27me3) levels at the *Robo1* locus were not significantly altered in heterozygotes ($p = 0.384$), which contrasts the result observed for homozygotes. As phenotypes in heterozygotes are generally milder than those in homozygotes, we suspect that the changes in the repressive mark might be under significant detection level. We have to admit that further precise studies would be required in the future. The result for heterozygotes is assessed in the legend for Fig. 7E.

13) The interpretation of the effect on IPCs proliferation illustrated in Figure 8f is again complicated by the fact that the manipulation does not target specifically IPC cells and therefore the effect on the number of mitotic IPCs could be a secondary one.

Given the reviewer's suggestion, we tried to inhibit the function of PRC2 specifically in IPCs (**Appendix Fig. S9E**) using the *Eomes*^{T2A-iCre} KI mice (described above, Appendix Fig. S6D). Because, in this Cre-LoxP system, it was difficult to express a certain shRNA and to express its effects specifically in IPCs very quickly, we conducted experiments to express the dominant negative (DN) mutant of EZH2 (a trimethylation enzyme protein in PRC2) specifically in IPCs (Y728* dominant negative mutant EZH2, Deevy et al. bioRxiv 2023, doi: <https://doi.org/10.1101/2023.06.01.543208>). First, we created the pCAG-LSL-*Ezh2*-DN (*Ezh2*^{DN}) expression vector. Next, we co-electroporated pCAG-LSL-*Ezh2*^{DN} and pCAG-H3GFP vectors into *Eomes*^{T2A-iCre/+} KI embryos at E13.5 and examined the percentage of PH3-positive cells in TBR2+/GFP+ cells at two days after electroporation (Appendix Fig. S9E). Compared to the control, we found that the rate of PH3-positive cells decreased in cells that had been introduced with pCAG-LSL-*Ezh2*^{DN}. However, such effects were restored by co-electroporation of wild-type *Ezh2* (pCAG-LSL-*Ezh2*^{WT}) (Appendix Fig. S9E). This suggests that EZH2, a significant component of PRC2, functions in IPCs and is involved in IPC cell division.

Dear Prof. Hoshino,

Thank you again for submitting your revised manuscript (EMBOJ-2024-117495R) to The EMBO Journal for our consideration. It has now been seen by the two original referees who had previously assessed the first version of your manuscript, and I am glad to say that they are both satisfied with the revision, find all previously raised concerns satisfactorily addressed, and now support publication of the manuscript in The EMBO Journal.

I am thus happy to let you know that your manuscript has in principle been accepted for publication in our journal. Congratulations on an excellent work and thank you for your thorough responses to the referees' comments!

There are a few minor formatting/editorial changes and corrections that we need from you to make in a final version of your manuscript before we can proceed with publication of your paper:

- Please rename the heading "Data and Materials Availability" to "Data Availability".
- Please provide the specific URLs for your deposited datasets with accession numbers PRJDB17034 and PRJDB17038 in your Data Availability statement, and make sure that the data will be publicly available at the time of publication.
- Please include your "Code availability" statement in the new "Data Availability" statement.
- Please rename the heading "Disclosure Statement & Competing Interests" to "Disclosure and competing interests statement".
- The author contributions statement should be removed from the manuscript file. Instead, we use CRediT to specify the contributions of each author in the journal submission system. Please feel free to use the free text box to provide more detailed descriptions during submission. See also our guide to authors for more information:
<https://www.embopress.org/page/journal/14602075/authorguide#authorshipguidelines>.
- Please update callouts for Appendix Fig. 9A to "Appendix Fig. S9A".
- Please remove "Supplementary Material" from the main manuscript file.
- The manuscript section order should be corrected as follows: Title page - Abstract & Keywords - Introduction - Results - Discussion - Methods - Data Availability - Acknowledgements - Disclosure and Competing Interests Statement - References - Figure Legends - Expanded View Figure Legends.
- Please provide the manuscript number (117495) in the general information table at the top of your Author Checklist.
- Please update your Table EV1 and Table EV2 to "Dataset EV1" and "Dataset EV2", respectively. Please change accordingly the file names, the titles, the legends, and all manuscript callouts. Please also remove the legends from the main manuscript and upload them instead as a separate tab/sheet in each Excel file.
- Please rename your previous Table EV3 to "Table EV1", and update accordingly its title, legend, and all manuscript callouts. Please also remove its legend from the main manuscript file and upload it instead as a separate tab/sheet in the Excel file.
- Please remove the instructions from the beginning of your Reagents and Tools table.
- Thank you for providing all requested Source Data. Please also complete and upload the Source Data checklist that our Source Data coordinator has shared with you (please upload it to our online manuscript tracking system as a "Related Manuscript File").
- During our standard Figure checks, our Data Integrity analyst detected a few cases of cell re-use in your Appendix Figures S3 and S4. In particular:
 1. Possible cell reused between Appendix Fig. S3 F&L
 2. Possible cell reused between Appendix Fig. S3 G&M
 3. Possible cell reused within Appendix Fig. S4 FPlease note that re-use in Figures is allowed only if it is necessary (and justified by the used experimental setup). Please double-check these cases of cell re-use, and either revise your Figures to avoid re-use, or explain in your cover letter why re-use is necessary. In the latter case, re-use should be explicitly mentioned in the Figure legends.
- During our routine checks, our data editors have raised the following queries regarding figures, data, and legends. Please make sure that all requests listed below are completely addressed in the final version of your manuscript:
 1. Please provide the exact p values in the legends of Figures 1d-f; 2f-g; 3b; 4c; 5b; 8c.
 2. Please indicate the statistical test used for data analysis in the legends of Figures 4e; 7b.

3. Please note that the box plots need to be defined in terms of minima, maxima, centre, bounds of box and whiskers, and percentile in the legend of Figure 4c.

Please also note that as part of the EMBO publications' Transparent Editorial Process, The EMBO Journal publishes online a Peer Review File along with each accepted manuscript. This File will be published in conjunction with your paper and will include the referee reports, your point-by-point response and all pertinent correspondence relating to the manuscript. You can opt out of this by letting the editorial office know (contact@embojournal.org). If you do opt out, the Peer Review File link will point to the following statement: "No Peer Review File is available with this article, as the authors have chosen not to make the review process public in this case."

We look forward to seeing a final version of your manuscript as soon as possible. Please let us know if you have any questions and use this link to submit your revision: <https://emboj.msubmit.net/cgi-bin/main.plex>.

Best regards,

Ioannis

Referee #1:

The authors have responded satisfactorily to the issues that I had raised. They clarified the information on the mouse models that were used and resolved some ambiguous statements by providing additional data. The findings are now also better integrated with the existing literature on Robo1, cell cycle length and PRC2 in cortical development. Overall, the interesting findings of this study related to Auts2 are now well substantiated and the work is ready to be published in EMBO J.

Referee #2:

I am satisfied that all the issues I raised in the first round have been addressed. I therefore recommend publication of the manuscript.

The authors addressed the minor editorial issues.

Dear Prof. Hoshino,

Thank you for addressing the majority of our formatting and other editorial requests. There is one remaining point that has not been completely addressed, and I would like to bring it to your attention, as we cannot move forward with acceptance and publication of your manuscript unless it is fully resolved in a final version of your manuscript.

As I informed you in my previous decision letter, we detected a few cases of cell re-use in your Appendix Figures S3 and S4, in particular:

- between Appendix Fig. S3 F&L
- between Appendix Fig. S3 G&M
- within Appendix Fig. S4 F

To address these issues, we kindly request you to:

1. Regarding Appendix Fig. S3: detail the cell section reuse between Appendix Fig. S3 F&L and Appendix Fig. S3 G&M in their respective Figure legends.
2. Regarding Appendix Fig. S4: annotate Appendix Figure S4F properly and detail the microscopy channels shown (in an "x axis legend").

We look forward to seeing a final version of your manuscript as soon as possible. Please let us know if you have any questions and use this link to submit your final revision: <https://emboj.msubmit.net/cgi-bin/main.plex>.

Best regards,

Ioannis

All editorial and formatting issues were resolved by the authors.

Dear Prof. Hoshino,

Congratulations on an excellent work! I am very pleased to inform you that it has been accepted for publication in The EMBO Journal. Thank you for your comprehensive responses to the initially raised referees' concerns and for addressing all our editorial and formatting requests.

If you have any questions, please do not hesitate to contact the Editorial Office. Thank you for your contribution to The EMBO Journal. Working with you was a pleasure!

Best regards,

Ioannis
